# Projection Optimization: A General Framework for Multi-Objective and Multi-Group RLHF

**Nuoya Xiong** [1]  **Aarti Singh** [1]

## Abstract

Reinforcement Learning with Human Feedback (RLHF) is a widely used fine-tuning approach that aligns machine learning model, particularly Language Model (LM) with human preferences. There are typically multiple objectives driving the preference, hence humans find it easier to express per-objective comparisons rather than a global preference between two choices. Multi-Objective RLHF aims to use per-objective preference feedback and achieve Pareto optimality among these objectives by aggregating them into a single unified objective for optimization. However, nearly all prior works rely on linear aggregation, which rules out policies that favor specific objectives such as the worst one. The only existing approach using non-linear aggregation is computationally expensive due to its reward-based nature and the need for retraining whenever the aggregation parameters change. In this work, we address this limitation by transforming the non-linear aggregation maximization problem into a series of sub-problems. Each sub-problem involves only linear aggregation, making it computationally efficient to solve. We further extend our framework to handle multi-group scenarios, where each group has distinct weights for the objectives. Our method enables achieving consensus or maximizing the aggregated objective across all groups. Theoretically, we demonstrate that our algorithmic framework achieves sublinear regret and can be easily adapted to a reward-free algorithm. Empirically, leveraging our theoretical insights, we propose a nearly training-free algorithm once the optimal policies for individual objectives are obtained.

[1]Carnegie Mellon University, PA, USA. Correspondence to: Aarti Singh <aarti@cs.cmu.edu>.

*Proceedings of the 42$^{nd}$ International Conference on Machine Learning*, Vancouver, Canada. PMLR 267, 2025. Copyright 2025 by the author(s).

## 1. Introduction

In recent years, there has been considerable effort to fine-tune a machine learning model, particularly Large Language Model (LLM), to perform better on particular tasks. RLHF is a popular fine-tuning approach, which receives the human's preference feedback and aligns the LLM model with human values using fine-tuning. Standard RLHF exploits human preference feedback between two outputs to maximize the expectation of the implicit or explicit reward function.

However, there are two main challenges for the application of RLHF in the real world. First, standard RLHF only maximizes a single reward function. However, people often find it hard to evaluate choices in an overall sense as, in reality, there are often *multiple objectives*. For example, comparing two papers or essays overall is harder than comparing them on specific objectives such as novelty, clarity, correctness etc. Similarly, recommending a city for vacation is harder than comparing cities on food options, nightlife, safety, etc. Each objective has its own implicit or explicit reward function, and the LLM needs to achieve a Pareto optimal trade-off between them by, for example, maximizing an aggregation of these reward function. Second, there are *multiple groups* of users in the real world who may prefer different aggregations of the objectives. For example, groups with different genders, political views, marital status, etc. This requires that the LLM either (a) satisfies the requirements of all the groups simultaneously, or (b) optimizes some aggregation across multiple groups.

**Multi-Objective Problem**   There are some works (Rame et al., 2024; Yang et al., 2024; Shi et al., 2024) that consider balancing the utilities of multiple objectives to get the Pareto optimal point or maximize the average expectation. Some works (Zhong et al., 2024; Park et al., 2024) consider multi-party problem in which each reward represents a group, which can also be regarded as a multi-objective problem. We assume that we have $m$ different objectives, and each objective has its own reward function $r_i(x,y)(1 \leq i \leq m)$. Each reward corresponds to an objective of the response $y$ like safety or helpfulness of the LLM, and the expected rewards $\mathbb{E}_{x\sim\rho, y\sim\pi(\cdot|x)}[r_i(x,y)]$ evaluate the LLM on these objectives respectively, where $\rho$ is the distribution of the prompt. Nearly all of the

previous work consider only linear aggregation, i.e., optimizing $u(\pi) = \sum_{i=1}^{m} \alpha_i \mathbb{E}_{x \sim \rho, y \sim \pi(\cdot|x)}[r_i(x, y)]$, where $\alpha = \{\alpha_i\}_{i \in [m]}$ is the weight of all objectives that is assumed to be known.

However, this kind of aggregation may not lead to an LLM that treats all objectives fairly. For example, the LLM may favor one objective significantly at the expense of another. In social choice theory, certain natural axioms such as monotonicity, symmetry, scale invariance, etc. which apply to multi-objective aggregation as well, lead to a more general function class (Cousins, 2021)

$$u(\pi) = \left( \sum_{i=1}^{m} \alpha_i \mathbb{E}_{x \sim \rho, y \sim \pi(\cdot|x)}[r_i(x, y)]^p \right)^{1/p}, p \leq 1, \tag{1}$$

The general $p$-norm aggregation with $p \leq 1$ promotes fairness across multiple objectives, which is particularly useful when aiming for a machine model that achieves a balanced performance among different objectives. Only one paper (Zhong et al., 2024) addresses the p-norm aggregation setting. In that work, the authors first learn a reward function for each objective, aggregate them into a new reward, and then attempt to optimize this new reward directly. However, this reward-based approach is computationally inefficient compared to the reward-free, DPO-based algorithm (Rafailov et al., 2024). Moreover, it requires retraining the entire policy whenever the aggregation method changes, which becomes even more time-consuming.

To reduce the computational cost of the reward-based RLHF algorithm, the paper (Shi et al., 2024) shows that for $p = 1$, once the optimal policy $\pi_{r_i}$ for each individual objective is obtained, the optimal policy $\pi_r$ for the linear averaged sum can be calculated as $\pi_r(y \mid x) \propto \prod_{i=1}^{m} \pi_{r_i}(y \mid x)^{\alpha_i}$. However, the derivation heavily depends on the linear structure of the aggregated reward $r(x, y)$. When $p \neq 1$, this approach breaks and the optimal policy cannot be written as a simple closed-form of the optimal policies of each objective. That raises the first question:

### *Question 1: Can we derive a computationally efficient MORLHF algorithm with non-linear aggregation?*

In our work, we propose a projection-based algorithm both in offline and online preference data settings, which transforms the nonlinear objective maximization problem into a sequence of subproblems, each involving only a linear maximization problem. Theoretically, we provide a thorough analysis for both offline and online setting, showing that it can converge to the optimal policy with a sublinear regret. Empirically, by leveraging the fact that there is a training-free algorithm for linear aggregation maximization, we derive a training-free algorithm for the generalized reward aggregation, which saves significant training time.

Moreover, previous work typically assumes that the weight for each objective is known. This assumption simplifies the problem and allows for straightforward optimization. However, in real-world applications, the importance weights $\{\alpha_i\}$ for each objective are usually unknown. In our work, we observe that the weight of an objective reflects its importance, which can be learned by how frequently the objective is reported in the human preferences. We propose a learning paradigm where the LLM learns objective weights from collected data, enabling the estimation of $\{\alpha_i\}$ and incorporating them into our theoretical results.

**Multi-Group Problem** Classical RLHF often assumes a single-group setting, ignoring the heterogeneity in human feedback and assuming that the human feedback relies on one unique reward function. However, real-world scenarios involve multiple groups with distinct preferences. Fine-tuning an LLM for each group is computationally expensive, making it essential to fine-tune the LLM to accommodate all groups' preferences simultaneously.

Since previous papers (Zhong et al., 2024; Park et al., 2024) working on multi-group RLHF only consider learning the reward function of each group under a single objective and then aggregating them, we regard them as a special case of the MORLHF. Hence, there is a lack of discussion about the multi-group setting where each group may have different importance for different objectives.

Formally, assume that we have $N$ group and $m$ objectives, and each group $n \in [N]$ has their own weight $\alpha^{(n)} \in \Delta_{m-1}$. The utility of the group $n$ for a LLM $\pi$ is then defined by

$$u^{(n)}(\pi) = \left( \sum_{i=1}^{m} \alpha_i^{(n)} \mathbb{E}_{x \sim \rho, y \sim \pi(\cdot|x)}[r_i(x, y)]^{p^{(n)}} \right)^{1/p^{(n)}},$$

where $p^{(n)} \leq 1$ is the parameter of the aggregation for group $n$. The reward function of each objective, $\{r_i(x, y)\}_{i \in [m]}$, remains fixed across different groups, while the weight $\alpha$ and the parameter $p$ can vary. In other words, the reward of each objective is the inherent value, and the importance weight represents the subjective part of each group. Now we pose the last question:

### *Question 2: Can we formulate and tackle the multi-group problem under MORLHF setting?*

In this paper, we consider two final goals for multi-group problem. Motivated by the poll theory, the first objective is called "consensus", in which LLM needs to meet the requirements of all groups as good as possible simultaneously. Motivated by social choice theory, the second objective is called "aggregation", in which the LLM needs to optimize a general aggregation of the utilities of all groups. We will show that our formulation and algorithmic framework naturally solve these two final goals. In summary, we have the following contributions:

- We reformulate the reward maximization in MORLHF as minimizing the distance between the current reward vector and a target set. This reframing decomposes the aggregated reward maximization into sub-problems, each focusing on minimizing the distance in a specific direction. These sub-problems reduce to linear aggregation and can be efficiently solved using previous approaches. Theoretically, we provide converge guarantees for both offline and online setting. Empirically, we provide a training-free algorithm once the optimal policy and the reward function for each objective is given, making it more computationally efficient.

- We tackle the multi-group problem in two ways: (1) achieving consensus by defining the target set as the intersection of all groups' target sets, and (2) minimizing the malfare function (Cousins, 2021) which aggregates the distance between each group's expected reward vector and its target set. Our framework addresses both problems concisely with theoretical guarantees.

- We establish a learning paradigm where the LMs learn the importance weight from data. We integrate weight estimation into the online setting and provide theoretical guarantees.

## 2. Related Works

**RLHF**  Fine-tuning LLMs with human feedback and RL is known as RLHF. The reward-based RLHF first extracts a reward model with a Bradley-Terry (BT) assumption on human preferences, and then optimizes the reward model (Ouyang et al., 2022; Bai et al., 2022; Touvron et al., 2023; Azar et al., 2024). On the other hand, the reward-free RLHF avoids explicit reward modeling by directly formulating the preference loss as a function of the policy and then using supervised learning (Wang et al., 2023; Rafailov et al., 2024), which is more stable and computation-friendly.

**MORLHF**  Multi-Objective RLHF (MORLHF) aims to align an LLM with human preferences while optimizing for multiple objectives, such as harmlessness, helpfulness, and humor. Most previous works aggregate rewards or models as the weighted sum of individual components. MORLHF (Wu et al., 2023; Bai et al., 2022) directly optimizes the aggregated reward using PPO, while MODPO (Zhou et al., 2023) provides a lightweight reward-free alternative. RS (Rame et al., 2024) combines individual models by averaging them. MOD (Shi et al., 2024) calculates the closed-form solution of the optimal policy for aggregated reward directly and derives a training-free algorithm. Only one work (Zhong et al., 2024) consider non-linear aggregation, and they optimize the aggregated reward function directly. However, this approach is computationally expensive and requires retraining when the aggregation changes. Instead, we pro-

*Table 1.* Comparison of previous work for MORLHF. The parameter $p$ means the exponent in Eq.(1). Algorithm 3 (offline setting) & 4 (online setting) have theoretical guarantees, while Algorithm 5 is the more practical version.

| | Aggergation | Reward Free | Traning Free | Multi-Group |
|---|---|---|---|---|
| MORLHF (Wu et al., 2023) | $p = 1$ | ✗ | ✗ | ✗ |
| RS (Rame et al., 2024) | $p = 1$ | ✓ | ✓ | ✗ |
| MOD (Shi et al., 2024) | $p = 1$ | ✓ | ✓ | ✗ |
| PNB (Zhong et al., 2024) | $p \leq 1$ | ✗ | ✗ | ✗ |
| Algorithm 3 & 4 | $p \leq 1$ | ✓ | ✗ | ✓ |
| Algorithm 5 | $p \leq 1$ | ✗ | ✓ | ✓ |

pose a theoretical framework that can be easily adapted to a reward-free algorithm, along with a training-free empirical algorithm built on the same theoretical framework. Detailed comparisons are shown in Table 1.

**Pluralistic Alignment and Preference Aggregation** There is a growing body of work on aligning machine learning models with diverse preferences, accounting for different values and perspectives. The works (Chakraborty et al., 2024; Ramesh et al., 2024) focus on optimizing the worst-case group loss, ensuring that the model achieves reasonable performance across all groups. (Park et al., 2024; Sorensen et al., 2024; Conitzer et al., 2024) explore how to aggregate preferences using social choice and voting theory, outlining a high-level roadmap for pluralistic AI alignment. (Ge et al., 2024) technically demonstrate that the BTL model fails to satisfy well-known standards in social choice theory and propose a novel rule-based approach for learning reward functions. (Chen et al., 2024) further study the generalization of the BTL model and introduce an ideal point model that better accommodates diverse groups.

## 3. Preliminaries and Notations

Denote the prompt space of the LLM as $\mathcal{X}$ and the response space as $\mathcal{Y}$. The distribution $\rho \in \Delta(\mathcal{X})$ represent the distribution of the prompt. A policy $\pi : \mathcal{X} \to \Delta(\mathcal{Y})$ represents an LLM that generates a response distribution given prompt $x$. In RLHF, we assume that we can get a pre-trained LLM $\pi_{\text{ref}}$ that is usually trained on supervised data. The goal is to fine-tune the pre-trained model to align the model with the human preference on one particular task. To be more specific, given prompt $x \sim \rho$, LLM can generate two responses $y_1, y_2$, then the human gives a preference feedback on the response pairs as either $y_1 \prec y_2$ or $y_1 \succ y_2$. The responses $y_1, y_2$ are labeled as $y_w, y_l$ respectively with probability $\mathbb{P}(y_1 \succ y_2 \mid x)$, and are labeled as $y_l, y_w$ with probability $1 - \mathbb{P}(y_1 \succ y_2 \mid x)$. It is further assumed that the human

preference is modeled by a Bradley-Terry (BT) model with the reward function $r^*(x, y) : \mathcal{X} \times \mathcal{Y} \mapsto [0, B]$:

$$\mathbb{P}(y_1 \succ y_2 \mid x) = \sigma(r^*(x, y_1) - r^*(x, y_2)),$$

where $\sigma(z) = \frac{1}{1+\exp(-z)}$ and $B \geq 1$. Given the reward function $r$, the optimal policy $\pi_r$ maximizes the expected reward function, with an additional KL divergence term that prevents the policy from deviating too much from $\pi_{\mathrm{ref}}$.

$$
\begin{aligned}
\pi_r &= \arg\max_\pi J(\pi) \\
&:= \mathbb{E}_{x \sim \rho} \mathbb{E}_{y \sim \pi(\cdot|x)} \left[ r^*(x, y) - \beta \mathbb{D}_{\mathrm{KL}}(\pi \parallel \pi_{\mathrm{ref}}) \right]. \quad (2)
\end{aligned}
$$

In this paper, we consider both offline and online RLHF. For the offline RLHF setting, the LLM has access to a pre-collected offline data $\mathcal{D}$ consisting of prompts and corresponding winning and losing responses, and the expectation in the optimal policy is calculated on the offline data. For the online setting, at each round LLM can generate two responses $y_1, y_2$ following the policy $\pi$, and then receive the preference feedback by human for data collection.

We assume there are $m$ known representations $\{\phi_i(x, y) \in \mathbb{R}^d\}_{i \in [m]}$ and the corresponding reward function class $\{r_i(x, y) = \theta_i^\top \phi_i(x, y) \in [0, B], \|\phi_i\|_2 \leq 1, \|\theta_i\|_2 \leq B\}$ for each objective $i \in [m]$. The true reward $r_i^*$ for objective $i$ can be written as $r_i^*(x, y) = (\theta_i^*)^\top \phi_i(x, y)$. This assumption is purely theoretical. In practice, the reward can be parameterized as $r^\theta$ using a neural network, and our practical algorithm 5 also does not rely on this assumption.

Since the preference only contains the information of $r_i(x, y_1) - r_i(x, y_2)$ for each objective $i$, rewards are invariant to constant shifts in feedback. Follow (Cen et al., 2024), we can assume there is a known policy $\pi_{\mathrm{base}}$ and constant $C$, such that for each $i \in [m]$, the reward parameter space $\Theta_i$ is defined as

$$\Theta_i = \left\{ \theta \in \mathbb{R}^d : \mathbb{E}_{\pi_{\mathrm{base}}} \langle \theta_i, \phi_i(x, y) \rangle = C \right\}. \quad (3)$$

### 3.1. Multi-Objective Learning

We assume that there are $m$ different objectives, and each objective has reward function $r_i(x, y) \in [0, B]$ for $i \in [m]$. As discussed in the introduction, we apply the definition of social welfare function in social choice theory to multi-objective setting and consider the weighted $p$-norm aggregation across objectives

$$u(\pi) = \left( \sum_{i=1}^m \alpha_i \mathbb{E}_{x \sim \rho, y \sim \pi(\cdot|x)} [r_i(x, y)]^p \right)^{1/p}, p \leq 1,$$

where $\alpha \in \Delta_{m-1}$ are weights of the objectives. Note that for positive rewards, aggregation yields Pareto optimality.

The goal is to find the optimal policy for the aggregated utility function. One natural approach to solving multi-objective RLHF is to first learn a reward model for each individual objective, and then aggregate these models to formulate a new reward. Finally, RL methods like PPO can be applied to optimize this new reward. However, this reward-based approach is significantly more computationally inefficient and unstable compared to reward-free approaches, such as DPO (Rafailov et al., 2024). Additionally, it requires retraining the entire model for all possible reward aggregations, which becomes time-consuming when the aggregation parameters change. In this work, we first provide a theoretical algorithmic framework for multi-objective RLHF, which naturally leads to the derivation of a reward-free algorithm. Based on this theoretical framework, we propose a *nearly training-free* practical algorithm that incurs almost zero computational cost once the optimal policy for each objective is obtained.

Previous techniques cannot be easily applied to this setting. In fact, for the linear aggregation when $p = 1$, the paper (Shi et al., 2024) finds that the optimal policy $\pi_r$ can be written as a closed-form of the optimal policy $\pi_{r_i}$ as $\pi_r(\cdot \mid x) \propto \pi_{\mathrm{ref}}(\cdot \mid x) \cdot \exp\left(\frac{1}{\beta} r(x, \cdot)\right)$, and conduct a decoding algorithm MOD using this derivation. By the linear aggregation $r(x, y) = \sum_{i=1}^m \alpha_i r_i(x, y)$ and $\sum_{i=1}^m \alpha_i = 1$, it is easy to verify that $\pi_r(y \mid x) \propto \prod_{i=1}^m \pi_{r_i}(y \mid x)^{\alpha_i}$. Hence, one natural reward-free algorithm is to first learn the optimal policy $\pi_{r_i}$ for each objective using DPO, then calculate the optimal policy $\pi_r$. It is also a training-free algorithm once the optimal policy for each objective is known. However, when we choose the general aggregation with $p \leq 1$, this derivation will fail due to the non-linear structure of the reward, making the problem much more complicated.

To avoid this technical difficulty, we draw inspiration from RL with Blackwell-approachability (Yu et al., 2021), which focuses on minimizing the distance between the reward vector and a specified target set. This approach makes the problem more tractable since we can incorporate the non-linear aggregation into the definition of the target set. To be more specific, a target set $W \subset \mathbb{R}^m$ is a convex set that is defined by

$$W_{p,c}^\alpha = \left\{ z \in \mathbb{R}_{\geq 0}^m : \left( \sum_{i=1}^m \alpha_i z_i^p \right)^{1/p} \geq c \right\},$$

where $\alpha$ represents the weights assigned to the objectives by humans, $p$ represents the degree of fairness, and $c$ reflects the requirement of humans. In practice, we can learn $\alpha$ and $p$ from supervised and preference data, and the parameter $c$ can be provided by humans or chosen by parameter tuning. The definition of target set implies that the group can be satisfied if the aggregation of the reward function is larger than some pre-defined constant. We also define the expected

reward vector $S(\pi) \in \mathbb{R}^m$ as $(S(\pi))_i = \mathbb{E}_\pi[r_i^*(x,y) - \beta\mathbb{D}_{KL}(\pi\|\pi_{\text{ref}})]$, which is the expected reward following the policy $\pi$ with a regularized term of KL divergence. Now assume $c, p, \alpha$ are all given, we can transfer the aggregation maximization problem to minimizing the distance between the expected reward vector (with some regularizer) and the target set $W$. The goal changes to minimizing the distance between $S(\pi)$ and $W_{p,c}^\alpha$:

$$\pi^* = \arg\min_\pi D(\pi) := d(S(\pi), W_{p,c}^\alpha). \qquad (4)$$

Note that if we choose $c$ as the maximum value that there exists a policy $\pi$ that satisfies $d(S(\pi), W_{p,c}^\alpha)) = 0$, then $\pi$ is one of the optimal policies and

$$\pi = \arg\max_{\pi\in\Pi} \left( \sum_{i=1}^m \alpha_i \mathbb{E}_\pi[r_i^*(x,y) - \beta\mathbb{D}_{\text{KL}}(\pi\|\pi_{\text{ref}})]^p \right)^{1/p}$$

where every $\pi \in \Pi$ satisfies that $\mathbb{E}_\pi[r_i^*(x,y)] - \beta\mathbb{D}_{\text{KL}}(\pi\|\pi_{\text{ref}}) \geq 0$. This statement highlights the connection between the original maximization problem Eq. (2) and our formulation Eq. (4). Therefore, our formulation can be viewed as an alternative metric for measuring the performance of LLMs in achieving multi-objective learning tasks.

Now we demonstrate that more general aggregation methods can enable LLM to accommodate a wider range of objectives by selecting different values of $p$.

*Example* 3.1 ($p = 1$ : Linear Aggregation). Now if we choose $p = 1$, and choose $c \geq \max_\pi \sum_{i=1}^m \alpha_i \mathbb{E}_\pi[r_i^*(x,y)]$, then the goal $D(\pi)$ will become

$$D(\pi) = d(S(\pi), W_{1,c}^\alpha)$$
$$= \frac{c - \sum_{i=1}^m \alpha_i \mathbb{E}_\pi[r_i^*(x,y)] + \beta\mathbb{D}_{\text{KL}}(\pi \| \pi_{\text{ref}})}{\sqrt{\sum_{i=1}^m \alpha_i^2}}.$$

The last equality is because the selection of $c$. From this derivation, we know that it is equivalent to the previous classical MORLHF with linear aggregation.

*Example* 3.2 ($p = -\infty$ : worst-case reward). When $p = -\infty$, the target set becomes

$$W_{-\infty,c}^\alpha = \left\{ z \in \mathbb{R}_{\geq 0}^m : \min_i z_i \geq c \right\},$$

which represents that the human wants to find an LLM with no obvious drawback for any of the objectives, i.e., requiring $\min_i \mathbb{E}_\pi[r_i^*(x,y)] - \mathbb{D}_{\text{KL}}(\pi\|\pi_{\text{ref}})$ larger than some threshold. Now we establish the connection between $p = -\infty$ and the max-min RLHF in (Chakraborty et al., 2024). The proof is provided in Appendix B.1.

*Theorem* 3.3. *Define the max-min value as* $c^* = \max_\pi[\min_i \mathbb{E}_\pi[r_i^*] - \beta\mathbb{D}_{\text{KL}}(\pi\|\pi_{\text{ref}})]$. *Then, if we choose the target set* $W_{-\infty,c}^\alpha$ *such that $c$ is close to $c^*$, the resulting optimal policy also achieves a max-min value that close to $c^*$. To be more specific, we have*

$$\min_i \mathbb{E}_\pi[r_i^*(x,y) - \mathbb{D}_{\text{KL}}(\pi\|\pi_{\text{ref}})] \geq c^* - (\sqrt{m} + 1)|c^* - c|.$$

### 3.2. Multi-Group Learning

Beyond the single group setting, we also study the multi-group setting, where each group has a different aggregation approach (parameterized by $c, p$ and $\alpha$). For each group $n$, we assume there is a target set

$$W^{(n)} = \left\{ z \in \mathbb{R}_{\geq 0}^m : \left( \sum_{i=1}^m \alpha_i^{(n)} z_i^{p^{(n)}} \right)^{\frac{1}{p^{(n)}}} \geq c^{(n)} \right\}$$

representing the aggregation rule across objectives for them. We consider two types of goals that represent the effectiveness of alignment across diverse groups.

**Consensus** The first goal is called "consensus", in which we wants to minimize the distance between the expected reward vector and the intersection of all target sets from diverse groups. Formally, the goal is to choose the optimal policy that minimizes the Euclidean distance

$$\pi^* = \arg\min_\pi d\left( S(\pi), \bigcap_{n=1}^N W^{(n)} \right). \qquad (5)$$

**Malfare Function Minimization** Another goal is to minimize the aggregated malfare function, where the malfare function for each group is the square of the distance between the expected reward vector and the group's target set. Formally, with group weight $\zeta_n > 0$ and $\sum_{n=1}^N \zeta_n = 1$, the goal is to find the optimal policy $\pi^*$ that

$$\pi^* = \arg\min_\pi \left( \sum_{n=1}^N \zeta_n \left( d^2(S(\pi), W^{(n)}) \right)^q \right)^{1/q}, q \geq 1.$$

## 4. Algorithms for Multiple Objectives with Linear Aggregation

In this section, we consider the simplest setting where the reward function is a linear aggregation, i.e. $r(x,y) = \sum_{i=1}^m d_i r_i^*(x,y)$, where $d \in \mathbb{R}^m$ is called the *direction*. In fact, the linear aggregation can be viewed as projecting the reward vector onto a specific direction $d$. As we will show later, this will become an essential sub-problem in our final algorithm for non-linear aggregation.

Given the dataset $\mathcal{D}_i = \{x^j, (y_w^j, y_l^j)\}_{j\in[M]}$ containing $M$ data points for objective $i$, we provide offline and online algorithms to learn the optimal policy with respect to multiple objectives in a consistent way. Now we aim to minimize the negative log-likelihood loss of preference data

$$L_i(\theta_i) = - \sum_{(x,y_w,y_l)\in\mathcal{D}_i} \log(\sigma(r_i^{\theta_i}(x,y_w) - r_i^{\theta_i}(x,y_l)))$$

for each objective $i$. Following (Cen et al., 2024), we can refine our estimation of the reward by adding an

additional exploration term $\max_\pi J(r^\theta, d, \pi) = \max_\pi \mathbb{E}_\pi[\sum_{i=1}^m d_i(r_i^{\theta_i} - \beta\mathbb{D}_{\mathrm{KL}}(\pi\|\pi_{\mathrm{ref}}))]$, which represents the optimism/pessimism principle of the online/offline learning process. To be more specific, for the offline and online setting, LLM learns the $\theta_{\mathrm{offline}}$ and $\theta_{\mathrm{online}}$ respectively by

$$\theta_{\mathrm{offline}} = \arg\max_{\theta_1,\cdots,\theta_m}\left(-\max_\pi J(r^\theta, d, \pi) - \sum_{i=1}^m \eta L_i(\theta_i)\right) \quad (6)$$

$$\theta_{\mathrm{online}} = \arg\max_{\theta_1,\cdots,\theta_m}\left(\max_\pi J(r^\theta, d, \pi) - \sum_{i=1}^m \eta L_i(\theta_i)\right), \quad (7)$$

where we use a single parameter $\theta$ to refer the set $\{\theta_i\}_{i\in[m]}$. The difference lies in the optimism and the pessimism principle. In the offline setting, we subtract the exploration term to avoid over-optimization (Cen et al., 2024; Liu et al., 2024) while in the online setting, we add the exploration term to encourage the model to explore (Cen et al., 2024). Then, the LLM executes the greedy policy $\pi^\theta = \arg\max_\pi J(r^\theta, d, \pi)$ to generate the response and receives the human feedback $(y_w, y_l)$. We called the algorithm **M**ulti-**O**bjective **P**rojection (MOP), and the pseudocode for online setting is shown in Algorithm 1. (There is no Line 4 and the output only has $\theta$ for the offline setting.)

---

**Algorithm 1** MOP-Reward Based (RB)

1: **Input**: Direction $d$, dataset $\{\mathcal{D}_i\}_{i\in[m]}, \eta, \beta$.
2: Calculate $\theta_{\mathrm{offline}}$ by Eq. (6) or $\theta_{\mathrm{online}}$ by Eq. (7).
3: Execute $\pi^\theta = \arg\max_\pi J(r_1^\theta, r_2^\theta, \cdots, r_m^\theta, d, \pi)$.
4: Given the prompt $x$, Generate two responses $y_1, y_2 \sim \pi$, and get a preference $y = (y_w, y_l)$.
5: **Output:** Data point $D = \{x, (y_w, y_l)\}$ and $\theta$.

---

The computational cost of Algorithm 1 mainly lies on Line 2. In fact, it needs to learn multiple reward functions directly, and then get the estimation of the optimal policy, which requires a joint optimization subprocedure. In the following, we consider the reward-free version of Algorithm 1.

**Reward-Free Modification** We now show that Algorithm 1 can be easily adapted to a reward-free version. We mainly consider the online setting since the offline setting is similar. Denote $\pi^\theta = \arg\max_\pi J(r^\theta, d, \pi)$. By the same derivation in (Cen et al., 2024), we can get

$$J(r^\theta, d, \pi) = C - \beta\mathbb{E}_{x\sim\rho, y\sim\pi_{\mathrm{base}}}\left[\log\frac{\pi^\theta(y \mid x)}{\pi_{\mathrm{ref}}(y \mid x)}\right],$$

where $C$ and $\pi_{\mathrm{base}}$ are the constant and the baseline policy in Eq. (3), $\pi_{\theta_i}$ is the policy for objective $i$ and $\pi^\theta \propto \pi_{\mathrm{ref}}(y \mid x) \cdot \prod_{i=1}^m (\pi_{\theta_i}(y \mid x))^{d_i}$ is the optimal policy for linear aggregation. The detailed derivation above will be provided in Appendix E. By the derivation in (Rafailov et al., 2024), you can further get the reward-free version of Eq. (7) as

$$\theta = \arg\min_\theta\left\{\beta\mathbb{E}_{\pi_{\mathrm{base}}}\log\pi^\theta(y \mid x) - \eta\sum_{i=1}^m \ell(\mathcal{D}_i, \theta_i)\right\} \quad (8)$$

where $\ell(\mathcal{D}_i, \theta_i) = \sum_{(x,y_w,y_l)\in\mathcal{D}_i}\log\sigma\Big(\beta\log\frac{\pi_{\theta_i}(y_w|x)}{\pi_{\mathrm{ref}}(y_w|x)} - \beta\log\frac{\pi_{\theta_i}(y_l|x)}{\pi_{\mathrm{ref}}(y_l|x)}\Big)$ is the reward-free loss function, and the expectation $\mathbb{E}_\pi[\cdot]$ means $\mathbb{E}_{x\sim\rho, y\sim\pi(\cdot|x)}[\cdot]$.

---

**Algorithm 2** MOP-Reward Free (RF) (Online Version)

1: **Input**: Direction $d$, dataset $\{\mathcal{D}_i\}_{i\in[m]}, \eta, \beta$.
2: Calculate $\theta_{\mathrm{online}} \in \mathbb{R}^m$ by Eq. (8) and $\pi = \pi^\theta$.
3: Given the prompt $x$, Generate two responses $y_1, y_2 \sim \pi$, and get a preference $y = (y_w, y_l)$.
4: **Output:** Data point $D = \{x, (y_w, y_l)\}$ and $\theta$.

---

The Eq. (8) involves an optimization problem on $\theta$, which is a complicated joint optimization since $\theta$ refers to $m$ parameter $\theta_1, \cdots, \theta_m$. In Appendix E, we further study the computational cost of Eq. (8), showing that the gradient descent update rule can be easily computed once the expectation of the score function is available, which commonly appears in previous RL algorithms such as REINFORCE (Williams, 1992).

# 5. General Algorithm for Preference Aggregation

In this section, we introduce general offline and online algorithms that work for both linear and non-linear preference aggregation, and provide their theoretical guarantees. Both algorithms transform the non-linear aggregation into a series of linear aggregation sub-problem, using Algorithm 1 and 2 as their core sub-procedures.

## 5.1. Offline Algorithm

Now we introduce our algorithm **M**ulti-**O**bjective **P**rojection **O**ptimization (MOPO), which follows from the competitive RL with Blackwell-approachability literature (Yu et al., 2021). We receive the offline data set $\mathcal{D} = \{\mathcal{D}_i\}_{i\in[m]}$ which contains $M$ data points $\mathcal{D}_i$ for each objective $i$. The algorithm learns the reward or optimizes the policy directly from the offline data. Our algorithm contains $T$ iterations. In each iteration $t$, we first project the reward vector on the direction $d^t \in \mathbb{R}^m$ defined in the last iteration, i.e. $r(x, y) = \sum_{i=1}^m d_i^t r_i(x, y)$, and then using the sub-procedure in the previous section to find the estimated parameter $\theta^t$ and determine the corresponding policy $\pi^t$. Finally, we derive the estimated expected reward vector $V^t \in \mathbb{R}^m$ as $(V^t)_i = \mathbb{E}_{\pi^t}[r_i^{\theta^t}(x, y) - \mathbb{D}_{\mathrm{KL}}(\pi^t\|\pi_{\mathrm{ref}})]$, and calculate the averaged reward vector as $\overline{V}^t = \frac{1}{t}\sum_{j=1}^t V^j$. Finally, the direction is updated based on the projection of the estimated point $\overline{V}^t$ onto the target set, guided by either the consensus problem or the malfare function minimization problem. The pseudocode is in Algorithm 3.

**Algorithm 3** MOPO-Offline

1: **Initial**: Dataset $\mathcal{D} = \{\mathcal{D}_i\}_{i \in [m]}, \{W^{(n)}\}_{n \in [N]}, \eta, \beta$.
2: **for** $t = 1, 2, \cdots, T$ **do**
3:   Collect $\theta^t$ by MOP-RB($\overline{d^t}, \mathcal{D}$) or MOP-RF($\overline{d^t}, \mathcal{D}$). Get the corresponding policy $\pi^t = \pi^{\theta^t}$.
4:   Calculate the point $V^t = \mathbb{E}_{\pi^t}[r_i^{\theta^t}(x, y) - \beta \mathbb{D}_{\mathrm{KL}}(\pi^t \| \pi_{\mathrm{ref}})] = C - \beta \mathbb{E}_{y \sim \pi_{\mathrm{base}}}[\log \frac{\pi_i^{\theta^t}(y|x)}{\pi_{\mathrm{ref}}(y|x)}] + \beta \mathbb{E}_{y \sim \pi^t}[\log \frac{\pi_i^{\theta^t}(y|x)}{\pi^t(y|x)}]$, and $\overline{V}^t = \frac{t-1}{t}\overline{V}^{t-1} + V^t$.
5:   Calculate the direction $d^{t+1}$ by Eq. (9) or Eq. (10), and calculate $\overline{d^{t+1}} = \frac{d^{t+1}}{\|d^{t+1}\|_1}$.
6: **end for**
7: **Return** $\tilde{\pi}^T = \frac{1}{T} \sum_{t=1}^T \pi^t$.

The key component of our algorithm is the direction calculation in each iteration. Intuitively, the algorithm aims to optimize the reward to guide the expected reward vector toward the target set as effectively as possible. Suppose the target set is $W$, the direction can be calculated by $d^{t+1} = \mathrm{Proj}(W, \overline{V}^t) = \frac{\Pi_W(V) - V}{\|\Pi_W(V) - V\|}$. For the consensus problem, we can substitute into $W = \bigcap_{n=1}^N W^{(n)}$ and get

$$d^{t+1} = \mathrm{Proj}\left(\bigcap_{n=1}^N W^{(n)}, \overline{V}^t\right). \tag{9}$$

For the malfare function minimization problem, we can first calculate the projection to each target set $W^{(n)}$ and then aggregate them as

$$d^{t+1} = \sum_{n=1}^N \mathrm{Proj}\left(W^{(n)}, \overline{V}^t\right) \cdot \frac{\zeta_n \|W^{(n)} - \overline{V}^t\|_2^{2q-1}}{\left(\sum_{n=1}^N \zeta_n \|W^{(n)} - \overline{V}^t\|_2^{2q}\right)^{\frac{2q-1}{2q}}}. \tag{10}$$

Note that if we apply MOPO with $p = 1$, it reduces to the classical MORLHF algorithm. This is because the direction $d^t = \mathrm{Proj}(V^t, W_{1,c}^\alpha) = \alpha$ for each $t$ as long as $c$ is large. However, for $p \neq 1$, MOPO solves the non-linear aggregation maximization problem by transforming into a series of subproblems, in which each subproblem only contains the linear aggregation and can be easily solved using any previous algorithm. Thus, MOPO serves as a general framework for MORLHF with non-linear aggregation. Moreover, suppose we use MOP-RF for each subproblem, MOPO is also a reward-free algorithm since the current reward vector can be computed as $(V^t)_i = \mathbb{E}_{\pi^t}[r_i^{\theta^t}(x, y) - \beta \mathbb{D}_{\mathrm{KL}}(\pi^t \| \pi_{\mathrm{ref}})] = C - \beta \mathbb{E}_{y \sim \pi_{\mathrm{base}}}[\log \frac{\pi_i^{\theta^t}(y|x)}{\pi_{\mathrm{ref}}(y|x)}] + \beta \mathbb{E}_{y \sim \pi^t}[\log \frac{\pi_i^{\theta^t}(y|x)}{\pi^t(y|x)}]$. (See Appendix E.3 for the derivation.)

Now we provide theoretical guarantee of Algorithm 3. The following result shows that MOP-offline can learn the optimal policy well if the offline dataset $\mathcal{D}$ has sufficient coverage for each objective.

**Theorem 5.1** (Consensus Problem). *Let $\eta = 1/\sqrt{M}$ and $\Sigma_{\mathcal{D}_i} = \frac{1}{M} \sum_{(x, y_w, y_l) \in \mathcal{D}_i} (\phi(x, y_w) - \phi(x, y_l))(\phi(x, y_w) - \phi(x, y_l))^\top$ be the empirical covariance matrix of the data for objective $i$. We consider the consensus problem that $W = \bigcap_{n=1}^N W^{(n)}$ and calculate the direction using Eq. (9). Define $D(\pi) = d(S(\pi), \cap_{n=1}^N W^{(n)})$. For $\delta \in (0, 1)$, with probability at least $1 - \delta$, we have*

$$D(\tilde{\pi}^T) - D(\pi^*)$$
$$\leq \widetilde{\mathcal{O}}\left(\frac{m^{3/2}\sqrt{d}}{\sqrt{M}} \mathrm{poly}\left(e^B, B', \left(\min_i \lambda_{\min}(\Sigma_{\mathcal{D}_i}) + \frac{1}{M}\right)^{-1}\right)\right)$$
$$+ \widetilde{\mathcal{O}}\left(\frac{B\sqrt{m}}{\sqrt{T}}\right).$$

The above theorem shows that the final gap of returned policy depends on the coverage term $\min_i \lambda_{\min}(\Sigma_{\mathcal{D}_i})$ of the offline dataset and the number of iterations $T$. As $T$ increases, we achieve a standard convergence rate of $\widetilde{\mathcal{O}}(1/\sqrt{M})$, which is standard in prior offline RL algorithms (Jin et al., 2021; Liu et al., 2020). We also provide the theoretical guarantee for malfare function minimization.

**Theorem 5.2** (Malfare). *With the same definitions and conditions in Theorem 5.1, we consider the malfare function minimization problem with an integer[1] exponential parameter $q \in \mathbb{N}^+$ and use Eq. (10) for the direction. Define $D_q(\pi) = \sqrt[2q]{\sum_{n=1}^N \zeta_n d^{2q}(S(\pi), W^{(n)})}$. For $\delta \in (0, 1)$, with probability at least $1 - \delta$ we have*

$$D_q(\tilde{\pi}^T) - D_q(\pi^*)$$
$$\leq \frac{Nm^{3/2}\sqrt{d}}{\sqrt{M}} \cdot \widetilde{\mathcal{O}}\left(\mathrm{poly}\left(e^{B'}, \min_i \lambda_{\min}(\Sigma_{\mathcal{D}_i})^{-1},\right.\right.$$
$$\left.\left.\left(\min_{n \in [N]} \zeta_n\right)^{-1/2q}\right)\right) + \widetilde{\mathcal{O}}\left(B\sqrt{m}T^{-1/2q}\right).$$

### 5.2. Online Algorithm

Now we provide the online version of MOPO, which is similar to the offline setting. The main difference is the adoption optimism principle (Eq. (7)) rather than the pessimism principle (Eq. (6)). Additionally, the dataset is collected incrementally online, and we also estimate the importance weight $\alpha$ instead of assuming it is known.

Additionally, rather than assuming the weight is known, we estimate it based on the frequency with which humans report the objective. This method also works offline by using the frequency of related data in the dataset. At each round $t$, given a prompt $x^t \sim \rho$ and two responses $y_1$ and $y_2$, each group $n$ identifies an objective $I^{t,(n)} \in [m]$ showing the greatest difference and provides preference feedback

---

[1]We focus on the integer case to simplify the proof.

**Algorithm 4** VPO-objective-learning-general

1: **Initial**: $\mathcal{D} = \emptyset$. parameter $\{p^{(n)}, c^{(n)}\}_{n \in [N]}, \eta, \beta$.
2: **for** $t = 1, 2, \cdots, T$ **do**
3:     Calculate $\tilde{\theta}_i^t = \arg\min_\theta L_i^t(\theta)$ for all $i \in [m]$.
4:     Estimate $\hat{\alpha}^{t,(n)} = \{\hat{\alpha}_i^{t,(n)}\}_{i \in [m]}$ for each $n \in [N]$ by MLE with $\mathcal{D}$ and $\{\tilde{\theta}_i^t\}_{i \in [m]}$ by Eq. (11)
5:     Calculate $W^{t,(n)} = W_{p^{(n)}, c^{(n)}}^{\alpha^{t,(n)}}$ where $\alpha^{t,(n)} = \frac{t-1}{t}\alpha^{t-1,(n)} + \frac{1}{t}\hat{\alpha}^{t,(n)}$ for each $n \in [N]$.
6:     Collect $D_t, \theta^t$ by MOP-RB($\overline{d^t}, \mathcal{D}$) or MOP-RF($\overline{d^t}, \mathcal{D}$), and update $\mathcal{D} = \mathcal{D} \cup D_t$.
7:     Calculate the point $V^t = \mathbb{E}_{\pi^t}[r_i^{\theta^t}(x,y) - \beta\mathbb{D}_{\mathrm{KL}}(\pi^t\|\pi_{\mathrm{ref}})] = C - \beta\mathbb{E}_{y\sim\pi_{\mathrm{base}}}[\log\frac{\pi_i^{\theta^t}(y|x)}{\pi_{\mathrm{ref}}(y|x)}] + \beta\mathbb{E}_{y\sim\pi^t}[\log\frac{\pi_i^{\theta^t}(y|x)}{\pi^t(y|x)}]$, and $\overline{V}^t = \frac{t-1}{t}\overline{V}^{t-1} + V^t$.
8:     Calculate the direction $d^{t+1}$ by Eq. (9) or Eq. (10), and calculate $\overline{d^{t+1}} = \frac{d^{t+1}}{\|d^{t+1}\|_1}$.
9: **end for**
10: **Return** $\tilde{\pi}^T = \frac{1}{T}\sum_{t=1}^T \pi^t$.

---

$(y_w^{t,(n)}, y_l^{t,(n)})$ on that objective. The model collects the data $(x^t, y_w^{t,(n)}, y_l^{t,(n)}, I^{t,(n)})$ into $\mathcal{D}^{(n)}$ for all group $n$. Next, we model how humans select the objective index. For responses $y_w$ and $y_l$, the gap on objective $i$ is quantified as $|\alpha_i \cdot (r_i(x, y_w) - r_i(x, y_l))|$, with the selection following a softmax distribution:

$$\mathbb{P}(I \mid \alpha, r^*, x, y_w, y_l) \propto \exp(\alpha_i \cdot |r_i^*(x, y_w) - r_i^*(x, y_l)|).$$

Then if we define the likelihood function as

$$\mathbb{L}(\alpha, \mathcal{D}^{(n)}, \theta) = \sum_{(x, y_w, y_l, I) \in \mathcal{D}^{(n)}} \mathbb{P}(I \mid \alpha, x, y_w, y_l, r^\theta),$$

we can estimate the importance weight vector for each group by MLE as

$$\hat{\alpha}^{t,(n)} = \arg\max_{\alpha \in \Delta_{m-1}} \mathbb{L}(\alpha, \mathcal{D}^{(n)}, \tilde{\theta}^t), \tag{11}$$

where we use an estimated reward parameter $\tilde{\theta}^t$ to approximate $\theta^*$. Before we present our results, we assume there is a gap between the reward obtained by following the optimal policy $\pi^*$ and the reference policy $\pi_{\mathrm{ref}}$. This gap is reasonable since the expected reward should be improved after fine-tuning.

**Assumption 5.3.** There exists a constant $\gamma > 0$ such that

$$\min_{i \in [m]} \mathbb{E}_{x\sim\rho, y_1\sim\pi^*, y_2\sim\pi_{\mathrm{ref}}}|r_i^*(x, y_1) - r_i^*(x, y_2)| \geq \gamma.$$

The following theorem shows that Algorithm 4 is a no-regret online algorithm that can converge to the optimal policy for the consensus problem.

**Theorem 5.4** (Consensus). *For the consensus problem, suppose the Assumption 5.3 holds and the group $n$ has parameter $p^{(n)}$ and $c^{(n)}$., then if we use Eq. (9) to calculate the direction, for $\delta \in (0, 1)$ and $\eta = 1/\sqrt{T}$, with probability at least $1 - \delta$ we have*

$$D(\tilde{\pi}^T) - D(\pi^*)$$
$$\leq \gamma^{-1}\mathrm{poly}(\exp(1/\beta), m, N, e^B, d, \log(1/\delta), \kappa,$$
$$(\min_{n\in[N]} p^{(n)})^{-1}), B_1) \cdot \widetilde{\mathcal{O}}(1/\sqrt{T}),$$

*where $\tilde{\pi}^T = \frac{1}{T}\sum_{t=1}^T \pi^t$, and $\kappa = \sup_{x,y}\frac{\pi_{\mathrm{base}}(y|x)}{\pi_{\mathrm{ref}}(y|x)}$, $B_1 = 2\sqrt{m}(B + \max_n c^{(n)})$ are constants.*

For the malfare minimization problem, we can derive online results similar to the offline setting.

**Theorem 5.5** (Malfare). *With the same setting in Theorem 5.4, if we consider the malfare function minimization problem with an integer exponential parameter $q \in \mathbb{N}^+$ and uses Eq. (10) to compute the direction, then for $\delta \in (0, 1)$ and $\eta = 1/\sqrt{T}$, with probability at least $1 - \delta$ we have*

$$D_q(\tilde{\pi}^T) - D_q(\pi^*)$$
$$\leq \gamma^{-1}\mathrm{poly}(\exp(1/\beta), m, N, e^B, d, \log(1/\delta), \kappa, B_1,$$
$$(\min_{n\in[N]} p^{(n)})^{-1}, (\min_{n\in[N]} \zeta_n)^{-1/2q}) \cdot \widetilde{\mathcal{O}}(T^{-1/2q}),$$

*where $\tilde{\pi}^T = \frac{1}{T}\sum_{t=1}^T \pi^t$, and $\kappa = \sup_{x,y}\frac{\pi_{\mathrm{base}}(y|x)}{\pi_{\mathrm{ref}}(y|x)}$, $B_1 = 2\sqrt{m}(B + \max_n c^{(n)})$ are constants.*

## 6. Experiments

We fine-tune a LLAMA2-7B model using Anthropic-HH dataset (Bai et al., 2022) with three different objectives of an LM assistant: Humor, Helpful, and Harmless. We run the offline version of MOPO, and use MOD (Shi et al., 2024) as the sub-procedure to solve the linear aggregation maximization problem at each round. The pseudocode is shown in Algorithm 5.

For $p = 0.5$, we compare the RS algorithm (Rame et al., 2024) and MOD algorithm (Shi et al., 2024) (both of which use linear aggregation), and a baseline AR that directly aggregates the reward using non-linear aggregation. The experimental results show that MOPO performs generally better. For $p = -\infty$, we compare MOPO with max-min RLHF (Chakraborty et al., 2024), showing that we achieve comparable performance. Note that MOPO is an iterate algorithm, thus the computational cost can still be high due to the large number of iterations. In practice, we can mitigate this by either reducing the number of iterations or computing a single gradient update per iteration (Guo et al., 2024). In our experiments, we set the number of iterations to 7, striking a balance between computational efficiency

and performance. To compute the expected reward vector $V^t$, we calculate the expectation by taking the expectation over 100 training samples, and we believe the performance of MOPO can be improved by using more training samples to calculate the expectation.

MOPO is a more general framework and can be applied to multi-group problems. More experiments and details are provided in Appendix A.

*Table 2.* Comparison of previous representative work for MORLHF with $p = 0.5, c = 0.5$ and the objective Harmless and Helpful. The score is the distance between the reward vector and the target set. The smaller one is better.

| $\alpha$ | Ours | RS | MOD | AR |
|---|---|---|---|---|
| (0.1,0.9) | **0.229** | 0.971 | 0.808 | 0.555 |
| (0.3,0.7) | **0.051** | 0.666 | 0.079 | 1.459 |
| (0.5,0.5) | **0.015** | 0.078 | 0.103 | 1.314 |
| (0.7,0.3) | **0.067** | 0.707 | 0.800 | 1.004 |
| (0.9,0.1) | **0.184** | 1.153 | 1.137 | 1.526 |

*Table 3.* Comparison of previous representative work for MORLHF with $p = 0.5$, $c = 1.3$ and the objective Harmless and Humor. The score is the distance between the evaluated reward vector and the target set. The smaller one is better.

| $\alpha$ | Ours | RS | MOD | AR |
|---|---|---|---|---|
| (0.1,0.9) | **0.335** | 0.362 | 0.337 | 1.767 |
| (0.3,0.7) | 0.578 | 0.678 | **0.572** | 2.011 |
| (0.5,0.5) | **0.720** | 0.882 | 0.723 | 1.970 |
| (0.7,0.3) | **0.630** | 0.860 | 0.722 | 2.411 |
| (0.9,0.1) | **0.217** | 0.391 | 0.396 | 2.068 |

*Table 4.* Comparison with max-min RLHF for objectives Humor and Harmless. The number pair represents the reward vector. The pair with the larger minimum value is better.

| | Ours | Max-Min RLHF |
|---|---|---|
| (Harmless, Humor) | (1.097,1.297) | **(1.530, 1.146)** |
| (Harmless, Helpful) | **(0.034,0.497)** | (-0.135, 0.393) |

## 7. Conclusion

In this paper, we study efficient multi-objective and multi-group RLHF problems under non-linear aggregation. By transforming the non-linear aggregation maximization into a series of linear aggregation maximization sub-problems, we find a computationally efficient algorithm that can converge to the optimal policy. Theoretically, we establish a general framework with converge guarantees for both offline and online settings, and the framework is also adaptable to a reward-free version. Empirically, we present a training-free framework given the reward functions and optimal policies for all objectives.

There are many future directions worth exploring. First, one can study how to learn the parameter $p$ in the aggregation function like (Pardeshi et al., 2024) using the preference feedback. Second, one can further study the token-level MORLHF (Zeng et al., 2024) based on our idea. Last, it is interesting to further study the multiple preference aggregation in Stochastic Transitivity model (Fishburn, 1973) instead of BTL model, and further discuss the relationship between them and previous distortion negative results (Anshelevich et al., 2021).

## Impact Statement

The goal of this paper is to advance the field of multi-objective RLHF, which can be applied to many applications in society. Our approach aims to mitigate biases in language models and promote fairness across diverse populations. However, it should be careful for implementation and evaluation to avoid unintended consequences, such as exacerbating inequalities or overlooking underrepresented groups.

## Acknowledgement

This work is supported in part by NSF AI Institute for Societal Decision Making under award IIS2229881 and ONR award N000142212363.

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

# A. Experiment Details

In this section, we provide our practical algorithm. We run the offline version of MOPO, and use MOD (Shi et al., 2024) as the sub-procedure to solve the linear aggregation maximization problem at each round. The pseudocode is shown in Algorithm 5.

---

**Algorithm 5** MOPO(Practical Version)-Offline

---

1: **Initial**: $\overline{d}^0 = (\frac{1}{m}, \cdots, \frac{1}{m})^\top$, dataset $\mathcal{D}_{\text{offline}}$, $W$.
2: Calculate the optimal policy $\pi_i$ for each objective $i \in [m]$ using offline dataset $\mathcal{D}_{\text{offline}}$.
3: **for** $t = 1, 2, \cdots, T$ **do**
4:     Execute $\pi^t = \text{MOD}(\{\pi_i\}_{i \leq m}, \overline{d^{t-1}})$.
5:     Calculate the point $V^t \in \mathbb{R}^m$.
6:     Calculate the direction $d^t = \text{Proj}(W, V^t)$, and get the average direction $\overline{d^t} = \frac{1}{t} \sum_{j=1}^t \frac{d^j}{\|d^j\|_1}$.
7: **end for**

---

Note that the algorithm average the direction instead of averaging the estimated reward vector function, which can lead to a more stable result. To execute the Line 2, following the previous paper (Shi et al., 2024), we first fine-tune the model LLAMA2-7B on the Anthropic-HH dataset (Ouyang et al., 2022) to get the reference policy $\pi_{\text{ref}}$. We then get the optimal policy $\pi_i$ for each objective $i \in \{1, 2, 3\}$ using PPO approach trained on three off-sheld reward model:

- Harmlessness: https://huggingface.co/Ray2333/gpt2-large-harmless-reward_model

- Helpfulness: https://huggingface.co/Ray2333/gpt2-large-helpful-reward_model

- Humor: https://huggingface.co/mohameddhiab/humor-no-humor

**Multi-Group Problem with Multiple Objectives**    We perform the experiments on Harmless and Humor dataset when we have $N = 2$ groups. One group has the target set $W_{0.5,1.3}^\alpha$ and the other has the target set $W_{-\infty,1}^\alpha$. We compare our consensus algorithm with Eq. (9) and a variant of max-min RLHF. In this variant of max-min RLHF, we use $\min\{r_1, r_2, \alpha_1 \cdot (\max\{r_1, 0\})^{0.5} + \alpha_2 \cdot (\max\{r_2, 0\})^{0.5}\}$ as the reward. We also perform experiments on the Harmless and Helpful dataset with the target set $W_{0.5,0.5}^\alpha$ and the target set $W_{-\infty,0}^\alpha$. The following tables show the experiment results. The results show that our algorithms perform relatively stable and better, while this variant of max-min RLHF performs unstable. However, note that this variant of max-min RLHF also needs retraining whenever one group changes the aggregation approach, which is time-consuming for real-world applications.

*Table 5.* Comparison of MOPO and a variant of Max-Min RLHF on multi-group setting. The objectives are Harmless and Humor. The score is the distance between the evaluated reward vector and the target set. The smaller one is better.

| $\alpha$ | Ours | Max-Min RLHF |
|---|---|---|
| (0.1,0.9) | **0.408** | 0.992 |
| (0.3,0.7) | **0.577** | 1.171 |
| (0.5,0.5) | 0.708 | **0.429** |
| (0.7,0.3) | **0.619** | 1.342 |
| (0.9,0.1) | 0.406 | **0.208** |

*Table 6.* Comparison of MOPO and a variant of Max-Min RLHF on multi-group setting. The objectives are Harmless and Helpful. The score is the distance between the evaluated reward vector and the target set. The smaller one is better.

| $\alpha$ | Ours | Max-Min RLHF |
|---|---|---|
| (0.1,0.9) | **0.230** | 1.073 |
| (0.3,0.7) | **0.052** | 0.123 |
| (0.5,0.5) | **0.015** | 0.261 |
| (0.7,0.3) | **0.067** | 0.204 |
| (0.9,0.1) | 0.184 | **0.121** |

# B. Proof of Theorems

## B.1. Proof of Theorem 3.3

*Proof.* Then, suppose the reward vector $S(\pi)$ is $(s_1, \cdots, s_m)^\top$, then by the definition of $D(\pi)$, we have

$$D(\pi) = \sum_{i=1}^{m} \max\{c - s_i, 0\}^2 \leq \sum_{i=1}^{m} \max\{c - s_i^*, 0\}^2,$$

where $s_i^* = (S(\pi^*))_i = \mathbb{E}_{\pi^*}[r_i^*(x, y) - \beta \mathbb{D}_{\mathrm{KL}}(\pi^* \| \pi_{\mathrm{ref}})]$. Hence we have

$$\max\{c - \min_i s_i, 0\}^2 \leq \sum_{i=1}^{m} \max\{c - s_i, 0\}^2$$

$$\leq \sum_{i=1}^{m} \max\{c - s_i^*, 0\}^2$$

$$\leq m \cdot (c - \min_i s_i^*)^2 \leq m(c - c^*)^2,$$

which implies that .

$$c - \min_i s_i \leq \sqrt{m} \cdot |c - c^*|,$$

and

$$c^* - \min_i s_i \leq (\sqrt{m} + 1)|c^* - c|.$$

Thus, if $c$ is selected such that $|c - c^*|$ is small, then we can also find a policy $\pi$, such that

$$\min_i \mathbb{E}_\pi[r_i^*(x, y) - \mathbb{D}_{\mathrm{KL}}(\pi \| \pi_{\mathrm{ref}})] \geq c^* - (\sqrt{m} + 1)|c^* - c|.$$

$\square$

## B.2. Proof of Theorem 5.1

For simplicity, for the following proof, we use $\mathbb{E}_{\pi^t}[\cdot]$ to represent $\mathbb{E}_{x \sim \rho, y \sim \pi^t(\cdot | x)}[\cdot]$. Since we do not assume the target set $W^*$ is approachable, we have the following property for the approachability:

**Lemma B.1.** *For each $\theta \in \mathbb{R}_{\geq 0}^m$ with $\|\theta\|_2 = 1$, we have*

$$\min_{x \in W^*} \langle \theta, x \rangle \leq \mathbb{E}_{\pi^*}\left[\langle \theta, r_i^*(x, y) \rangle - \sum_{i=1}^{m} \theta_i \beta \mathbb{D}_{\mathrm{KL}}(\pi^* \| \pi_{\mathrm{ref}})\right] + D(\pi^*) = \|\theta\|_1 \cdot J\left(r_1^*, \cdots, r_m^*, \frac{\theta}{\|\theta\|_1}, W^*, \pi^*\right) + D(\pi^*)$$

$$\leq \sqrt{m} \cdot J\left(r_1^*, \cdots, r_m^*, \frac{\theta}{\|\theta\|_1}, W^*, \pi^*\right) + D(\pi^*)$$

*Proof.* By the definition of $D(\pi^*) = d(S(\pi^*), W^*)$, we know that there exists a vector $p$ with $S(\pi^*) + p \in W^*$ and $\|p\|_2 = D(\pi^*)$. Then we can have

$$\min_{x \in W^*} \langle \theta, x \rangle \leq \langle \theta, S(\pi^*) + p \rangle \leq \mathbb{E}_{\pi^*}\left[\langle \theta, r_i^*(x, y) \rangle - \sum_{i=1}^{m} \theta_i \beta \mathbb{D}_{\mathrm{KL}}(\pi^* \| \pi_{\mathrm{ref}})\right] + D(\pi^*).$$

The last inequality holds because of $\|\theta\|_1 \leq \sqrt{m}$. $\square$

We can first bound the regret by

$$D(\tilde{\pi}^T) - D(\pi^*)$$
$$= d(W^*, \mathbb{E}_{\tilde{\pi}^T}[r^*(x, y)] - \beta \mathbb{D}_{\mathrm{KL}}(\tilde{\pi}^T \| \pi_{\mathrm{ref}})) - D(\pi^*)$$

$$\leq d\left(W^*, \mathbb{E}_{\bar{\pi}^T}[r^*(x,y)] - \frac{\beta}{T}\sum_{t=1}^{T}\mathbb{D}_{\mathrm{KL}}(\pi^t\|\pi_{\mathrm{ref}})\right) - D(\pi^*)$$

$$= \underbrace{d\left(W^*, \mathbb{E}_{\bar{\pi}^T}[r^*(x,y)] - \frac{\beta}{T}\sum_{t=1}^{T}\mathbb{D}_{\mathrm{KL}}(\pi^t\|\pi_{\mathrm{ref}})\right) - d\left(W^*, \frac{1}{T}\sum_{t=1}^{T}\mathbb{E}_{\pi^t}[\hat{r}^t(x,y)] - \frac{\beta}{T}\sum_{t=1}^{T}\mathbb{D}_{\mathrm{KL}}(\pi^t\|\pi_{\mathrm{ref}})\right)}_{(A)}$$

$$+ d\left(W^*, \frac{1}{T}\sum_{t=1}^{T}\mathbb{E}_{\pi^t}[\hat{r}^t(x,y)] - \frac{\beta}{T}\sum_{t=1}^{T}\mathbb{D}_{\mathrm{KL}}(\pi^t\|\pi_{\mathrm{ref}})\right) - D(\pi^*)$$

$$= (A) + d\left(W^*, \overline{V}^T\right) - D(\pi^*). \tag{12}$$

The inequality uses the fact that

$$\mathbb{D}_{\mathrm{KL}}(\tilde{\pi}\|\pi_{\mathrm{ref}}) \leq \frac{1}{T}\left(\sum_{t=1}^{T}\mathbb{D}_{\mathrm{KL}}(\pi^t\|\pi_{\mathrm{ref}})\right).$$

Recall that

$$D(\pi) = d(W^*, \mathbb{E}_{\pi^t}[r(x,y)] - \beta\mathbb{D}_{\mathrm{KL}}(\pi\|\pi_{\mathrm{ref}}))$$

and $\pi^* = \min_\pi D(\pi^*)$. Now, by Lemma B.1, for each $\theta \in \mathbb{R}^m$ with $\|\theta\|_1 \leq 1$, we have

$$\min_{x\in W^*}\langle\theta, x\rangle \leq \mathbb{E}_{\pi^*}[\langle\theta, r_i^*(x,y)\rangle - \sum_{i=1}^{m}\theta_i\beta\mathbb{D}_{\mathrm{KL}}(\pi^*\|\pi_{\mathrm{ref}})] + D(\pi^*) = J(r_1^*, \cdots, r_m^*, \theta, W^*, \pi^*) + D(\pi^*).$$

Denote $V^t \in \mathbb{R}^m$ with $(V^t)_i = \mathbb{E}_{\pi^t}[\hat{r}_i^t(x,y) - \beta\mathbb{D}_{\mathrm{KL}}(\pi^t\|\pi_{\mathrm{ref}})]$, and $\frac{1}{t}\overline{V}^t = \sum_{i=1}^{t}V^i$. We have

$$d(\overline{V}^T, W^*)^2 = \|\overline{V}^T - \Pi_{W^*}(\overline{V}^T)\|^2$$
$$\leq \|\overline{V}^T - \Pi_{W^*}(\overline{V}^{T-1})\|^2$$
$$= \left(\frac{T-1}{T}\right)^2 d(\overline{V}^{T-1}, W^*)^2 + \frac{1}{T^2}\|V^T - \Pi_{W^*}(\overline{V}^{T-1})\|^2$$
$$+ \frac{2(T-1)}{T^2}(\overline{V}^{T-1} - \Pi_{W^*}(\overline{V}^{T-1})) \cdot (V^T - \Pi_{W^*}(\overline{V}^{T-1})).$$

First, based on the definition of $W^*$, it is easy to show that $d^t \succeq 0$. $\pi^t$ is the optimal policy such that

$$\mathbb{E}_{\pi^t}[\langle d^t, \hat{r}(x,y)\rangle] - \sum_{i=1}^{m}d_i^t\beta\mathbb{D}_{\mathrm{KL}}(\pi^t\|\pi_{\mathrm{ref}})] \geq \mathbb{E}_{\pi_{\mathrm{ref}}}[\langle d^t, \hat{r}(x,y)\rangle] \geq 0,$$

thus $(\sum_{i=1}^{m}d_i^t) \cdot \beta\mathbb{D}_{\mathrm{KL}}(\pi^t\|\pi_{\mathrm{ref}}) \leq \mathbb{E}_{\pi^t}[d^t \cdot \hat{r}(x,y)] \leq B$. Hence, given $d^t \succeq 0$ and $\|d^t\|_2 = 1$,

$$\beta\mathbb{D}_{\mathrm{KL}}(\pi^t\|\pi_{\mathrm{ref}}) \leq \frac{B}{\sum_{i=1}^{m}d_i^t} \leq B.$$

we have $|(V^t)_i| \leq B$ and

$$\|V^T - \Pi_{W^*}(\overline{V}^{T-1})\|^2 \leq B^2 m.$$

Thus by iteration we can have

$$T^2 d(\overline{V}^T, W^*)^2 \leq T \cdot B^2 m + \sum_{t=1}^{T}2(t-1)(\overline{V}^{t-1} - \Pi_{W^*}(\overline{V}^{t-1})) \cdot (V^t - \Pi_{W^*}(\overline{V}^{t-1})).$$

Now, by the definition of $d^t$, we have

$$(\overline{V}^{t-1} - \Pi_{W^*}(\overline{V}^{t-1})) \cdot (V^t - \Pi_{W^*}(\overline{V}^{t-1}) = d(\overline{V}^{t-1}, W^*) \cdot d^t \cdot (\Pi_{W^*}(\overline{V}^{t-1}) - V^t).$$

Then, we prove the following lemma.

**Lemma B.2.** $\min_{x \in W^*} \langle d^t, x \rangle = d^t \cdot \Pi_{W^*}(\overline{V}^{t-1})$.

*Proof.* In fact, we only need to prove that for any $x \in W^*$, $\langle \overline{V}^{t-1} - \Pi_{W^*}(\overline{V}^{t-1}), x - \Pi_{W^*}(\overline{V}^{t-1}) \rangle \leq 0$. Suppose there exists $x \in W^*$ such that $\langle \overline{V}^{t-1} - \Pi_{W^*}(\overline{V}^{t-1}), x - \Pi_{W^*}(\overline{V}^{t-1}) \rangle > 0$, then since $W^*$ is a convex set, for any $\lambda \in (0,1)$, we have $x_\lambda = \lambda x + (1-\lambda)\Pi_{W^*}(\overline{V}^{t-1}) \in W^*$. Consider the line

$$\Pi_{W^*}(\overline{V}^{t-1}) + t\frac{\Pi_{W^*}(\overline{V}^{t-1}) - x}{\|\Pi_{W^*}(\overline{V}^{t-1}) - x\|}, \quad t \in \mathbb{R}.$$

Also, we consider the projection of $\overline{V}^{t-1}$ on this line, and denote it as $p$. Then we can get

$$0 < \langle \overline{V}^{t-1} - \Pi_{W^*}(\overline{V}^{t-1}) - p + p, x - \Pi_{W^*}(\overline{V}^{t-1}) \rangle = \langle p - \Pi_{W^*}(\overline{V}^{t-1}), x - \Pi_{W^*}(\overline{V}^{t-1}) \rangle.$$

Hence when $\lambda \to 0$, $x_\lambda$ is between $p$ and $\Pi_{W^*}(\overline{V}^{t-1})$. Also,

$$\|\overline{V}^{t-1} - x_\lambda\|^2 = \|\overline{V} - p\|^2 + \|p - x_\lambda\|^2 \leq \|\overline{V} - p\|^2 + \|p - \Pi_{W^*}(\overline{V}^{t-1})\|^2 \leq \|\Pi_{W^*}(\overline{V}^{t-1}) - d^t\|^2,$$

which contradicts the selection of $\Pi_{W^*}(\overline{V}^{t-1})$. $\qquad\square$

Now, by Lemma B.1 and Lemma B.2, we can get

$$d^t \cdot \Pi_{W^*}(\overline{V}^{t-1}) \leq J(r_1^*, \cdots, r_m^*, d^t, \pi^*) + D(\pi^*).$$

Then, since we define $\overline{d^t} = d^t/\|d^t\|_1$, we can continue the analysis by

$$(\overline{V}^{t-1} - \Pi_{W^*}(\overline{V}^{t-1})) \cdot (V^t - \Pi_{W^*}(\overline{V}^{t-1}))$$
$$= d(\overline{V}^{t-1}, W^*) \cdot \left( \|d^t\|_1 J(r_1^*, r_2^*, \cdots, r_m^*, \overline{d^t}, \pi^*) + D(\pi^*) - d^t \cdot V^t \right)$$
$$= d(\overline{V}^{t-1}, W^*) \cdot (\|d^t\|_1 \cdot \left( J(\hat{r}_1^*, \cdots, \hat{r}_m^*, \overline{d^t}, \pi^*) - J(\hat{r}_1, \cdots, \hat{r}_m, \overline{d^t}, \pi^t) \right) + D(\pi^*))$$
$$= d(\overline{V}^{t-1}, W^*) \cdot (\|d^t\|_1 \cdot (\eta \sum_{i=1}^m L_i(\theta^t) - \eta \sum_{i=1}^m L_i(\theta^*)) + D(\pi^*)).$$

Thus we can get

$$Td(\overline{V}^T, W^*)^2 \leq B^2 m + \sum_{t=1}^T \frac{2(t-1)}{T} d(\overline{V}^{t-1}, W^*) \cdot (\eta\|d^t\|_1 \sum_{i=1}^m L_i(\theta^t) - \eta\|d^t\|_1 \sum_{i=1}^m L_i(\theta^*) + D(\pi^*)).$$

Now we use induction method to show that

$$d(\overline{V}^t, W^*) \leq D(\pi^*) + \frac{\eta}{T} \sum_{t=1}^T \|d^t\|_1 \sum_{i=1}^m (L_i(\theta^t) - L_i(\theta^*)) + 2Bm/\sqrt{t}.$$

When $t = 1$, the inequality holds by

$$\|d(\overline{V}^1, W^*) - D(\pi^*)\| \leq d(\overline{V}^1, S(\pi^*)) \leq 2B.$$

Denote $A_j = \eta \cdot \|d^j\|_1 \cdot (\sum_{i=1}^m (L_i(\theta^*) - L_i(\theta^j)))$ and $S_t = \sum_{j=1}^t A_j$, then for all $t \in [T-1]$, suppose we have

$$d(\overline{V}^{t-1}, W^*) \leq D(\pi^*) + \frac{1}{t-1} S_{t-1} + 2B\left(\frac{\sqrt{m}}{\sqrt{t-1}}\right).$$

Then we substitute these induction hypothesis into the recursion inequality and get

$$Td(\overline{V}^T, W^*)^2$$

$$\leq B^2 m + \sum_{t=1}^{T} \frac{2(t-1)}{T} \left( D(\pi^*) + \frac{1}{t-1} S_{t-1} + 2B \left( \frac{\sqrt{m}}{\sqrt{t-1}} \right) \right) (A_t + D(\pi^*))$$

$$\leq B^2 m + \sum_{t=1}^{T} \left( \frac{2(t-1)}{T} D(\pi^*) + \frac{1}{T} S_{t-1} + 2B \left( \frac{2\sqrt{m}\sqrt{t-1}}{T} \right) \right) (A_t + D(\pi^*))$$

$$= B^2 m + (T-1)D(\pi^*)^2 + \sum_{t=1}^{T} \frac{1}{T} S_{t-1} A_t + \sum_{t=1}^{T} \left( \frac{1}{T} S_{t-1} + \frac{2(t-1)}{T} A_t \right) D(\pi^*)$$

$$\quad + \sum_{t=1}^{T} 2B \left( \frac{2\sqrt{m}\sqrt{t-1}}{T} \right) (A_t + D(\pi^*))$$

$$\leq B^2 m + (T-1)D(\pi^*)^2 + \frac{1}{T} S_T^2 + \sum_{t=1}^{T} D(\pi^*) \cdot \left( \frac{T+t-1}{T} A_t \right) + 2\sqrt{m} \cdot 2B\sqrt{T}D(\pi^*) + (2\sqrt{m} \cdot 2B/\sqrt{T})S_T$$

$$\leq B^2 m + (T-1)D(\pi^*)^2 + \frac{1}{T} S_T^2 + D(\pi^*) \cdot (2S_T) + 2\sqrt{m} \cdot 2B\sqrt{T}D(\pi^*) + 2\sqrt{m} \cdot 2B/\sqrt{T}S_T$$

$$\leq T \cdot (2B\sqrt{m}/\sqrt{T} + D(\pi^*) + \frac{1}{T} S_T)^2.$$

Thus we have

$$d(\overline{V}^T, W^*) \leq D(\pi^*) + \frac{\eta}{T} \sum_{t=1}^{T} \|d^t\|_1 \sum_{i=1}^{m} (L_i(\theta^t) - L_i(\theta^*)) + \frac{2B\sqrt{m}}{\sqrt{T}}.$$

Now we derive the final regret. By inequality Eq. (12), we can get

$$D(\tilde{\pi}^T) - D(\pi^*) \leq (A) + d(W^*, \overline{V}^T) - D(\pi^*)$$

$$\leq (A) + \underbrace{\frac{\eta}{T} \sum_{t=1}^{T} \|d^t\|_1 \sum_{i=1}^{m} (L_i(\theta^t) - L_i(\theta^*))}_{(B)} + \frac{2B\sqrt{m}}{\sqrt{T}}.$$

Now we consider the error term (A), which represents the approximation error of the reward function. Now we have

$$(A) \leq \frac{1}{T} \sum_{t=1}^{T} \mathbb{E}_{\pi^t} \left[ \sum_{i=1}^{m} |\hat{r}_i^t(x,y) - r_i^*(x,y)| \right]$$

$$= \frac{1}{T} \sum_{t=1}^{T} \sum_{i=1}^{m} \mathbb{E}_{\pi^t} \left[ \|\phi_i(x,y)\|_{(\Sigma_{\mathcal{D}_i} + \lambda I)^{-1}} \|\theta_i^t - \theta_i^*\|_{\Sigma_{\mathcal{D}_i} + \lambda I} \right].$$

Similar to (Cen et al., 2024), since $(r^\theta, \pi^\theta)$ can be formulated as a saddle point of the objective $J(r, d, \pi) + \sum_{i=1}^{m} \eta L_i(\theta_i)$ for any direction $d \in (\mathbb{R}^+)^m$, we have

$$\eta \nabla_{\theta_i} L_i(\theta_i) + d_i \mathbb{E}_{x \sim \rho, y \sim \pi^\theta}[\phi_i(x,y)] + \lambda_1 \mathbb{E}_{x \sim \rho, y \sim \pi_{\text{base}}}[\phi_i(x,y)] = 0.$$

Also, denote $\theta_{\text{MLE}} = \arg\min_{\theta \in \Theta} \sum_{i=1}^{m} \eta L_i(\theta_i)$, we have

$$\eta \nabla_{\theta_i} L_i(\theta_{i,\text{MLE}}) + \lambda_2 \mathbb{E}_{x \sim \rho, y \sim \pi_{\text{base}}}[\phi_i(x,y)] = 0.$$

Follow the same derivation in (Cen et al., 2024), we can get

$$\|\theta_i^t - \theta_{i,\text{MLE}}\|_{\Sigma_{\mathcal{D}_i} + \lambda I} \leq \frac{d_i}{\eta} \cdot \frac{(3 + e^{B'})4(\lambda_{\min}(\Sigma_{\mathcal{D}_i}) + \lambda)^{-1}}{M} + 2\sqrt{\lambda(B')^2}$$

$$\leq \frac{(3 + e^{B'})4(\lambda_{\min}(\Sigma_{\mathcal{D}_i}) + \lambda)^{-1}}{\sqrt{M}} + 2\sqrt{\lambda(B')^2},$$

where $B'$ is the upper bound of norm of $\theta$, i.e. $\max_{\theta \in \Theta} \|\theta\|_2 \leq B'$.

Now we recall the Lemma 3.1 in (Zhu et al., 2023), which bounds the true parameter and the MLE parameter.

**Lemma B.3** (Lemma 3.1 in (Zhu et al., 2023)). $\lambda > 0$ is a positive constant. For $\delta \in (0, 1)$, with probability at least $1 - \delta$, we will have

$$\|\theta_i^* - \theta_{i,\mathrm{MLE}}\|_{\Sigma_{\mathcal{D}_i} + \lambda I} \leq \mathcal{O}\left((3 + e^{B'})\sqrt{\frac{d + \log(1/\delta)}{M}} + \sqrt{\lambda(B')^2}\right).$$

*Also, $L_i(\theta)$ is a convex function. In fact,*

$$\frac{1}{3 + e^{B'}}\Sigma_{\mathcal{D}_i} \preceq \frac{1}{M}\nabla_\theta^2 L_i(\theta) \preceq \frac{1}{4}\Sigma_{\mathcal{D}_i}.$$

Hence, we get

$$
\begin{aligned}
\text{(A)} &\leq \frac{1}{T}\sum_{t=1}^{T}\sum_{i=1}^{m}\mathbb{E}_{\pi^t}\left[\|\phi_i(x,y)\|_{(\Sigma_{\mathcal{D}_i} + \lambda I)^{-1}}\|\theta_i^t - \theta_i^*\|_{\Sigma_{\mathcal{D}_i} + \lambda I}\right] \\
&\leq \frac{1}{T}\sum_{t=1}^{T}\sum_{i=1}^{m}\|\mathbb{E}_{\pi^t}\phi_i(x,y)\|_{(\Sigma_{\mathcal{D}_i} + \lambda I)^{-1}} \cdot \mathcal{O}\left(\frac{(3 + e^{B'})4(\lambda_{\min}(\Sigma_{\mathcal{D}_i}) + \lambda)^{-1}\sqrt{d + \log(1/\delta)}}{\sqrt{M}} + \sqrt{\lambda(B')^2}\right) \\
&\leq \widetilde{\mathcal{O}}\left(\frac{m(3 + e^{B'})4(\lambda_{\min}(\Sigma_{\mathcal{D}_i}) + \lambda)^{-2}\sqrt{d + \log(1/\delta)}}{\sqrt{M}} + \sqrt{\lambda(B')^2}\right).
\end{aligned}
$$

The notation $\widetilde{\mathcal{O}}(\cdot)$ hides all the logarithm term like $\log(1/\delta)$.

Now we consider the term (B). First, based on the convexity of $L_i(\theta)$, we have

$$
\begin{aligned}
L_i(\theta_i^t) - L_i(\theta_{i,\mathrm{MLE}}) &\leq \langle \nabla_\theta L_i(\theta^t), \theta_i^t - \theta_{i,\mathrm{MLE}}\rangle \\
&= \frac{1}{\eta}\langle -d_i\mathbb{E}_{x\sim\rho,y\sim\pi^\theta}[\phi_i(x,y)] - \lambda_1\mathbb{E}_{x\sim\rho,y\sim\pi_{\mathrm{base}}}[\phi_i(x,y)], \theta_i^t - \theta_{i,\mathrm{MLE}}\rangle \\
&= \frac{d_i}{\eta}\langle -\mathbb{E}_{x\sim\rho,y\sim\pi^\theta}[\phi_i(x,y)] - \mathbb{E}_{x\sim\rho,y\sim\pi_{\mathrm{base}}}[\phi_i(x,y)], \theta_i^t - \theta_{i,\mathrm{MLE}}\rangle \\
&\leq \frac{d_i}{\eta}\|\mathbb{E}_{x\sim\rho,y\sim\pi^\theta}[\phi_i(x,y)] - \mathbb{E}_{x\sim\rho,y\sim\pi_{\mathrm{base}}}[\phi_i(x,y)]\|_{(\Sigma_{\mathcal{D}_i} + \lambda I)^{-1}}\|\theta_i^t - \theta_{i,\mathrm{MLE}}\|_{\Sigma_{\mathcal{D}_i} + \lambda I} \\
&\leq \frac{2d_i}{\eta} \cdot (\lambda_{\min}(\Sigma_{\mathcal{D}_i}) + \lambda)^{-1} \cdot \|\theta_i^t - \theta_{i,\mathrm{MLE}}\|_{\Sigma_{\mathcal{D}_i} + \lambda I} \\
&\leq \mathcal{O}\left(\frac{(3 + e^{B'})(\lambda_{\min}(\Sigma_{\mathcal{D}_i}) + \lambda)^{-2}}{\sqrt{M}} + \frac{4}{\eta}\sqrt{\lambda(B')^2} \cdot (\lambda_{\min}(\Sigma_{\mathcal{D}_i}) + \lambda)^{-1}\right).
\end{aligned}
$$

The last inequality uses the fact that $d_i \leq 1$. Also, with probability at least $1 - \delta$, we have

$$L_i(\theta_{i,\mathrm{MLE}}) - L_i(\theta^*) \leq \widetilde{\mathcal{O}}(1).$$

Now sum over $t \in [T]$, we can get

$$
\begin{aligned}
\text{(B)} &\leq \frac{\eta}{T}\sum_{t=1}^{T}\|d^t\|_1\sum_{i=1}^{m}(L_i(\theta_i^t) - L_i(\theta^*)) \\
&\leq \sqrt{m} \cdot \frac{m}{\sqrt{M}}\widetilde{\mathcal{O}}\left(\frac{(3 + e^{B'})(\min_i \lambda_{\min}(\Sigma_{\mathcal{D}_i}) + \lambda)^{-2}}{\sqrt{M}} + \frac{4}{\eta}\sqrt{\lambda(B')^2} \cdot (\min_i \lambda_{\min}(\Sigma_{\mathcal{D}_i}) + \lambda)^{-1} + 1\right),
\end{aligned}
$$

where the last inequality uses the fact that $\eta = 1/\sqrt{M}$ and $\|d^t\|_1 \leq \sqrt{m}$. Hence, we have

$$D(\tilde{\pi}^T) - D(\pi^*)$$

$$\leq \text{(A)} + \text{(B)} + \frac{2Bm}{\sqrt{T}}$$

$$\leq \widetilde{\mathcal{O}}\left( \frac{m^{3/2}(3 + e^{B'})(\min_i \lambda_{\min}(\Sigma_{\mathcal{D}_i} + \lambda)^{-2}\sqrt{d + \log(1/\delta)}}{\sqrt{M}} + \frac{4m^{3/2}}{\eta\sqrt{M}}B'\sqrt{\lambda} \cdot (\min_i \lambda_{\min}(\Sigma_{\mathcal{D}_i}) + \lambda)^{-1} + \frac{m^{3/2}}{\sqrt{M}} + \frac{B\sqrt{m}}{\sqrt{T}} \right)$$

$$\leq \widetilde{\mathcal{O}}\left( \frac{m^{3/2}(3 + e^{B'})(\min_i \lambda_{\min}(\Sigma_{\mathcal{D}_i} + \lambda)^{-2}\sqrt{d + \log(1/\delta)}}{\sqrt{M}} + \frac{4m^{3/2}B'}{\sqrt{M}} \cdot (\min_i \lambda_{\min}(\Sigma_{\mathcal{D}_i}) + \lambda)^{-1} + \frac{m^{3/2}}{\sqrt{M}} + \frac{B\sqrt{m}}{\sqrt{T}} \right)$$

$$= \frac{m}{\sqrt{M}} \cdot \widetilde{\mathcal{O}}\left( \text{poly}\left( e^{B'}, \min_i \lambda_{\min}(\Sigma_{\mathcal{D}_i})^{-1}, \sqrt{d + \log(1/\delta)}, B' \right) \right) + \widetilde{\mathcal{O}}\left( \frac{B\sqrt{m}}{\sqrt{T}} \right).$$

The last step is because $\eta = 1/\sqrt{M}, \lambda = 1/M$. Hence we complete the proof. $\qquad\square$

## B.3. Proof of Theorem 5.2

*Proof.* The main proof framework is similar to Theorem 5.1. The difference lies in the approach to deal with the aggregated $p$-norm of the distance.

$$\sum_{n=1}^{N} \zeta_n d^{2q}(\overline{V}^t, W_n^*) = \sum_{n=1}^{N} \zeta_n \|\overline{V}^T - \Pi_{W_n^*}(\overline{V}^T)\|^{2q} \tag{13}$$

$$\leq \sum_{n=1}^{N} \zeta_n \|\overline{V}^T - \Pi_{W_n^*}(\overline{V}^{T-1})\|^{2q} \tag{14}$$

$$= \sum_{n=1}^{N} \zeta_n \left\| \frac{T-1}{T}(\overline{V}^{T-1} - \Pi_{W_n^*}(\overline{V}^{T-1})) + \frac{1}{T}(V^T - \Pi_{W_n^*}(\overline{V}^{T-1})) \right\|^{2q}. \tag{15}$$

For the vector $x_n, y_n \in \mathbb{R}^m$ with $x_n = (T-1)(\overline{V}^{T-1} - \Pi_{W_n^*}(\overline{V}^{T-1})), y_n = (V^T - \Pi_{W_n^*}(\overline{V}^{T-1}))$ we know $\|x_n\| \leq 2TB\sqrt{m}, \|y_n\| \leq 2B\sqrt{m}$. Hence, since $q > 1$, we have

$$\|x_n + y_n\|^{2q} \leq (\|x_n\|^2 + \|y_n\|^2 + 2\langle x_n, y_n \rangle)^q$$

$$\leq \|x_n\|^{2q} + 2\langle x_n, y_n \rangle \|x_n\|^{2q-2} + 3^q \cdot T^{2q-2}(2B)^{2q}m^q$$

We can further bound the inequality (15) as

$$T^{2q} \sum_{n=1}^{N} \zeta_n d^{2q}(\overline{V}^T, W_n^*) \leq \sum_{n=1}^{N} \zeta_n \|x_n + y_n\|^{2q} \tag{16}$$

$$\leq (T-1)^{2q} \sum_{n=1}^{N} \zeta_n d^{2q}(\overline{V}^{T-1}, W_n^*) + 12^q \cdot T^{2q-2}B^{2q}m^q \tag{17}$$

$$+ 2(T-1) \sum_{n=1}^{N} \zeta_n (\overline{V}^{T-1} - \Pi_{W_n^*}(\overline{V}^{T-1}))(V^T - \Pi_{W_n^*}(\overline{V}^{T-1}))\|x_n\|^{2q-2}. \tag{18}$$

Then since $\|x_n\| = (T-1)d(\overline{V}^{T-1}, W_n^*)$, we can finally get

$$T^{2q} \sum_{n=1}^{N} \zeta_n d^{2q}(\overline{V}^T, W_n^*) \leq (T-1)^{2q} \sum_{n=1}^{N} \zeta_n d^{2q}(\overline{V}^{T-1}, W_n^*) + 12^q T^{2q-2}B^{2q}m^q \tag{19}$$

$$+ 2(T-1)^{2q-1} \sum_{n=1}^{N} \zeta_n (\overline{V}^{T-1} - \Pi_{W_n^*}(\overline{V}^{T-1}))(V^T - \Pi_{W_n^*}(\overline{V}^{T-1}))d^{2q-2}(\overline{V}^{T-1}, W_n^*). \tag{20}$$

Hence by the recursion, we have

$$
T^{2q} \sum_{n=1}^{N} \zeta_n d^{2q}(\overline{V}^T, W_n^*) \le 12^q T^{2q-1} B^{2q} m^q
$$

$$
+ 2(t-1)^{2q-1} \sum_{t=1}^{T} \sum_{n=1}^{N} \zeta_n (\overline{V}^{t-1} - \Pi_{W_n^*}(\overline{V}^{t-1}))(V^t - \Pi_{W_n^*}(\overline{V}^{t-1})) d^{2q-2}(\overline{V}^{t-1}, W_n^*).
$$

Now the last term at the right side can be further bounded by

$$
\sum_{t=1}^{T} \sum_{n=1}^{N} \zeta_n (t-1)^{2q-1} (\overline{V}^{t-1} - \Pi_{W_n^*}(\overline{V}^{t-1}))(V^t - \Pi_{W_n^*}(\overline{V}^{t-1})) d^{2q-2}(\overline{V}^{t-1}, W_n^*)
$$

$$
\le \sum_{t=1}^{T} \sum_{n=1}^{N} \zeta_n (t-1)^{2q-1} d^{2q-1}(\overline{V}^{t-1}, W_n^*) d_n^t \cdot (\Pi_{W_n^*}(\overline{V}^{t-1}) - V^t)
$$

$$
\le \sum_{t=1}^{T} \sum_{n=1}^{N} \zeta_n (t-1)^{2q-1} d^{2q-1}(\overline{V}^{t-1}, W_n^*) \left( \|d_n^t\|_1 J(r_1^*, \cdots, r_m^*, \overline{d_n^t}, \pi^*) + d(S(\pi^*), W_n^*) - \|d_n^t\|_1 J(\hat{r}_1, \cdots, \hat{r}_m, \overline{d_n^t}, \pi^t) \right)
$$

$$
\le \sum_{t=1}^{T} (t-1)^{2q-1} \left( \sum_{n=1}^{N} \zeta_n d^{2q}(\overline{V}^{t-1}, W_n^*) \right)^{\frac{2q-1}{2q}}
$$

$$
\cdot \left( \|d^t\|_1 J(r_1^*, \cdots, r_m^*, \overline{d^t}, \pi^*) + \sqrt[2q]{\sum_{n=1}^{N} \zeta_n d^{2q}(S(\pi^*), W_n^*)} - \|d^t\|_1 J(\hat{r}_1, \cdots, \hat{r}_m, \overline{d^t}, \pi^t) \right) \tag{21}
$$

$$
\le \sum_{t=1}^{T} (t-1)^{2q-1} \left( \sum_{n=1}^{N} \zeta_n d^{2q}(\overline{V}^{t-1}, W_n^*) \right)^{\frac{2q-1}{2q}} \cdot \left( D_q(\pi^*) + \eta \|d^t\|_1 \left( \sum_{i=1}^{m} L_i^t(\theta^*) - \eta \sum_{i=1}^{m} L_i^t(\theta^t) \right) \right). \tag{22}
$$

The inequality Eq. (21) derives from the definition of $d^t$ in Eq. (10) and Cauchy's inequality. Let $S_T = \sqrt[2q]{\sum_{n=1}^{N} \zeta_n d^{2q}(\overline{V}^T, W_n^*)}$, then we can get

$$
T S_T^{2q} \le 12^q \cdot B^{2q} m^q + \sum_{t=1}^{T} \frac{2(t-1)^{2q-1}}{T^{2q-1}} S_{t-1}^{2q-1} \cdot \left( D_q(\pi^*) + \eta \|d^t\|_1 \cdot \left( \sum_{i=1}^{m} L_i^t(\theta^*) - \sum_{i=1}^{m} L_i^t(\theta^t) \right) \right).
$$

Define $A_t = D_q(\pi^*) + \eta \|d^t\|_1 \cdot \left( \sum_{i=1}^{m} L_i^t(\theta^*) - \sum_{i=1}^{m} L_i^t(\theta^t) \right)$, then we use the induction to show that there exists a constant $C_q$ such that

$$
S_t \le \left( \frac{1}{t} \sum_{s=1}^{t} A_s + C_q T^{-1/2q} \right).
$$

In fact, it holds when $t = 1$. Now suppose it holds for $t = 1, 2, \cdots, T-1$, we have

$$
S_T^{2q} \le 12^q \cdot B^{2q} m^q / T + \sum_{t=1}^{T} \frac{2(t-1)^{2q-1}}{T^{2q-1}} S_{t-1}^{2q-1} \cdot \frac{A_t}{T}
$$

$$
\le 12^q \cdot B^{2q} m^q / T + 2 \sum_{t=1}^{T} \left( \frac{1}{T} \sum_{s=1}^{t-1} A_s + C_q T^{-1/2q} \right)^{2q-1} \cdot \frac{A_t}{T}
$$

$$
\le 12^q \cdot B^{2q} m^q / T + 2 \sum_{t=1}^{T} \sum_{k=0}^{2q-1} \binom{2q-1}{k} \frac{1}{T} \left( \sum_{s=1}^{t-1} A_s \right)^{k+1} \cdot \frac{A_t}{T} \cdot (C_q)^{2q-1-k} T^{-\frac{2q-1-k}{2q}}
$$

$$
\le 12^q \cdot B^{2q} m^q / T + \sum_{k=0}^{2q-1} \binom{2q-1}{k} (C_q)^{2q-1-k} T^{-\frac{2q-1-k}{2q}} \left( \frac{1}{T} \sum_{t=1}^{T} A_t \right)^{k+1}.
$$

Now we choose $C_q = (12^q \cdot B^{2q} m^q)^{\frac{1}{2q}} = \sqrt{12B^2 m}$, then we have

$$
\begin{aligned}
S_T^{2q} &\leq \mathcal{O}(C_q^{2q}/T) + \sum_{k=0}^{2q-1} \binom{2q-1}{k} (C_q)^{2q-1-k} T^{-\frac{2q-1-k}{2q}} \left( \frac{1}{T} \sum_{t=1}^{T} A_t \right)^{k+1} \\
&\leq \mathcal{O}(C_q^{2q}/T) + \sum_{k=0}^{2q-1} \binom{2q}{k+1} (C_q)^{2q-1-k} T^{-\frac{2q-1-k}{2q}} \left( \frac{1}{T} \sum_{t=1}^{T} A_t \right)^{k+1} \\
&\leq \left( \frac{1}{T} \sum_{t=1}^{T} A_t + C_q T^{-1/2q} \right)^{2q} .
\end{aligned}
$$

which implies that

$$
S_T \leq \frac{1}{T} \sum_{s=1}^{T} A_s + C_q T^{-1/2q} = \frac{1}{T} \sum_{s=1}^{T} A_s + 4B\sqrt{m} \cdot T^{-1/2q}. \tag{23}
$$

Hence we have

$$
\sqrt[2q]{\sum_{n=1}^{N} \zeta_n d^{2q}(\overline{V}^T, W_n^*)} - D_q(\pi^*) \leq \frac{\eta}{T} \sum_{t=1}^{T} \|d^t\|_1 \sum_{i=1}^{m} (L_i^t(\theta^*) - L_i^t(\theta^t)) + \widetilde{\mathcal{O}}(B\sqrt{m} T^{-1/2q}).
$$

Now we derive the final regret. We can see

$$
D_q(\tilde{\pi}^T) - D_q(\pi^*) \tag{24}
$$

$$
= \sqrt[2q]{\sum_{n=1}^{N} \zeta_n d^{2q}(S(\tilde{\pi}^T), W_n^*)} - D_q(\pi^*)
$$

$$
= \sqrt[2q]{\sum_{n=1}^{N} \zeta_n d^{2q}\left(W_n^*, \mathbb{E}_{\tilde{\pi}^T}[r^*(x,y)] - \frac{\beta}{T} \sum_{t=1}^{T} \mathbb{D}_{\mathrm{KL}}(\pi^t \| \pi_{\mathrm{ref}})\right) \cdot \mathbf{1}^m}
$$

$$
- \sqrt[2q]{\sum_{n=1}^{N} \zeta_n d^{2q}\left(W_n^*, \frac{1}{T} \sum_{t=1}^{T} \mathbb{E}_{\pi^t}[r^{\theta^t}(x,y)] - \frac{\beta}{T} \sum_{t=1}^{T} \mathbb{D}_{\mathrm{KL}}(\pi^t \| \pi_{\mathrm{ref}})\right) \cdot \mathbf{1}^m} + \sqrt[2q]{\sum_{n=1}^{N} \zeta_n d^{2q}(W_n^*, \overline{V}^T)} - D_q(\pi^*)
$$

$$
\leq \underbrace{\sqrt[2q]{\sum_{n=1}^{N} \zeta_n \left( d\left(W_n^*, \mathbb{E}_{\tilde{\pi}^T}[r^*(x,y)] - \frac{\beta}{T} \sum_{t=1}^{T} \mathbb{D}_{\mathrm{KL}}(\pi^t \| \pi_{\mathrm{ref}}) \cdot \mathbf{1}^m\right) - d\left(W_n^*, \frac{1}{T} \sum_{t=1}^{T} \mathbb{E}_{\pi^t}[\hat{r}^t(x,y)] - \frac{\beta}{T} \sum_{t=1}^{T} \mathbb{D}_{\mathrm{KL}}(\pi^t \| \pi_{\mathrm{ref}}) \cdot \mathbf{1}^m\right) \right)^{2q}}}_{(A)}
$$

$$
+ \underbrace{\frac{\eta}{T} \sum_{t=1}^{T} \|d^t\|_1 \sum_{i=1}^{m} (L_i^t(\theta^*) - L_i^t(\theta^t))}_{(B)} + \widetilde{\mathcal{O}}(B\sqrt{m} T^{-1/2q}). \tag{25}
$$

The last inequality uses the triangle inequality for $2q$-norm. Now also note that

$$
d\left(W_n^*, \mathbb{E}_{\tilde{\pi}^T}[r^*(x,y)] - \frac{\beta}{T} \sum_{t=1}^{T} \mathbb{D}_{\mathrm{KL}}(\pi^t \| \pi_{\mathrm{ref}}) \cdot \mathbf{1}^m\right) - d\left(W_n^*, \frac{1}{T} \sum_{t=1}^{T} \mathbb{E}_{\pi^t}[r^*(x,y)] - \frac{\beta}{T} \sum_{t=1}^{T} \mathbb{D}_{\mathrm{KL}}(\pi^t \| \pi_{\mathrm{ref}}) \cdot \mathbf{1}^m\right)
$$

$$
\leq \frac{1}{T} \sum_{t=1}^{T} \sum_{i=1}^{m} \mathbb{E}_{\pi^t} |r_i^*(x,y) - \hat{r}_i^t(x,y)|,
$$

we have

$$
(A) \leq \sqrt[2q]{\left( \sum_{n=1}^{N} \zeta_n \right) \left( \sum_{i=1}^{m} \left| \frac{1}{T} \sum_{t=1}^{T} \mathbb{E}_{\pi^t}[r_i^*(x,y) - \hat{r}_i^t(x,y)] \right| \right)^{2q}} = \sum_{i=1}^{m} \left| \frac{1}{T} \sum_{t=1}^{T} \mathbb{E}_{\pi^t}[r_i^*(x,y) - \hat{r}_i^t(x,y)] \right|
$$

Now follow the same proof as Theorem 5.1,

$$\text{(A)} \le \widetilde{\mathcal{O}}\left(\frac{m(3 + e^{B'})4(\lambda_{\min}(\Sigma_{\mathcal{D}_i}) + \lambda)^{-2}\sqrt{d + \log(1/\delta)}}{\sqrt{M}} + \sqrt{\lambda(B')^2}\right),$$

and

$$\text{(B)} \le \frac{\eta}{T}\sum_{t=1}^{T}\|d^t\|_1 \sum_{i=1}^{m}(L_i^t(\theta^*) - L_i^t(\theta^t))$$

$$\le \frac{N\sqrt{m}\cdot m}{\sqrt{M}}\widetilde{\mathcal{O}}\left(\frac{(3 + e^{B'})(\min_i \lambda_{\min}(\Sigma_{\mathcal{D}_i}) + \lambda)^{-2}}{\sqrt{M}} + \frac{4}{\eta}\sqrt{\lambda(B')^2}\cdot(\min_i \lambda_{\min}(\Sigma_{\mathcal{D}_i}) + \lambda)^{-1} + 1\right),$$

where the last inequality we use the fact that

$$\|d^t\|_1 = \left\|\sum_{n=1}^{N} d_n^t \cdot \frac{\zeta_n\|W^{(n)} - \overline{V}^t\|_2^{2q-1}}{\left(\sum_{n=1}^{N}\zeta_n\|W^{(n)} - \overline{V}^t\|_2^{2q}\right)^{\frac{2q-1}{2q}}}\right\|_1 \le \sum_{n=1}^{N}\|d_n^t\|_1 \cdot \zeta_n^{1/2q} \le N\sqrt{m}.$$

Combining the Eq. (25) and the upper bounds for (A) and (B), substitute into $\eta = 1/\sqrt{M}$ and $\lambda = 1/M$, we can complete the final proof. $\square$

## B.4. Proof of Theorem 5.4

*Proof.* Recall that $V^t \in \mathbb{R}^m$ with $(V^t)_i = \mathbb{E}_{\pi^t}[r_i^{\theta_i^t}(x, y) - \beta\mathbb{D}_{\mathrm{KL}}(\pi^t\|\pi_{\mathrm{ref}})]$, and $\overline{V}^t = \frac{1}{t}\sum_{i=1}^{t}V^i$. We also define $W^0 = \{(0,0)\}$. Since $W^t$ is the estimation of $W^*$ at round $t$, we have

$$d(\overline{V}^T, W^T)^2 = \|\overline{V}^T - \Pi_{W^T}(\overline{V}^T)\|^2$$

$$\le \|\overline{V}^T - \Pi_{W^T}(\overline{V}^{T-1})\|^2$$

$$\le \|\overline{V}^T - \Pi_{W^{T-1}}(\overline{V}^{T-1})\|^2 + \|\Pi_{W^T}(\overline{V}^{T-1}) - \Pi_{W^{T-1}}(\overline{V}^{T-1})\|^2$$

$$+ 2\langle\overline{V}^T - \Pi_{W^{T-1}}(\overline{V}^{T-1}), \Pi_{W^T}(\overline{V}^{T-1}) - \Pi_{W^{T-1}}(\overline{V}^{T-1})\rangle. \tag{26}$$

Now by Lemma C.3, since $\|V^{T-1}\|_\infty \le B$ is bounded, we have

$$\|\Pi_{W^T}(\overline{V}^{T-1}) - \Pi_{W^{T-1}}(\overline{V}^{T-1})\|_2^2 \le 4d(\overline{V}^{T-1}, W^{T-1})d_{B_1}(W^T, W^{T-1}) + 2d_{B_1}^2(W^T, W^{T-1}).$$

Then we can get

$$\|\Pi_{W^T}(\overline{V}^{T-1}) - \Pi_{W^{T-1}}(\overline{V}^{T-1})\|_2 \le 2\sqrt{d(\overline{V}^{T-1}, \Pi_{W^{T-1}}(\overline{V}^{T-1}))d_{B_1}(W^T, W^{T-1})} + \sqrt{2}d_{B_1}(W^T, W^{T-1}).$$

Then the third term on the right side can be bounded by

$$\langle\overline{V}^T - \Pi_{W^{T-1}}(\overline{V}^{T-1}), \Pi_{W^T}(\overline{V}^{T-1}) - \Pi_{W^{T-1}}(\overline{V}^{T-1})\rangle$$

$$\le \langle\overline{V}^{T-1} - \Pi_{W^{T-1}}(\overline{V}^{T-1}), \Pi_{W^T}(\overline{V}^{T-1}) - \Pi_{W^{T-1}}(\overline{V}^{T-1})\rangle$$

$$+ \|\overline{V}^T - \overline{V}^{T-1}\| \cdot \|\Pi_{W^T}(\overline{V}^{T-1}) - \Pi_{W^{T-1}}(\overline{V}^{T-1})\|$$

$$\le d(\overline{V}^{T-1}, W^{T-1}) \cdot \left\langle d^t, \Pi_{W^T}(\overline{V}^{T-1}) - \Pi_{W^{T-1}}(\overline{V}^{T-1})\right\rangle + \frac{1}{T}\|\Pi_{W^T}(\overline{V}^{T-1}) - \Pi_{W^{T-1}}(\overline{V}^{T-1})\|.$$

Now denote $\tilde{d}^t = \frac{\Pi_{W^T}(\overline{V}^{T-1}) - \overline{V}^{T-1}}{\|\Pi_{W^T}(\overline{V}^{T-1}) - \overline{V}^{T-1}\|}$, then by Lemma C.4, we can get

$$d(\overline{V}^{T-1}, W^{T-1}) \cdot \|d^t - \tilde{d}^t\| \le 4\sqrt{d(\overline{V}^{T-1}, W^{T-1})d_{B_1}(W^{T-1}, W^T)} + 2d_{B_1}(W^{T-1}, W^T),$$

Then we can bound the inner product term as

$$d(\overline{V}^{T-1}, W^{T-1}) \cdot \left\langle d^t, \Pi_{W^T}(\overline{V}^{T-1}) - \Pi_{W^{T-1}}(\overline{V}^{T-1}) \right\rangle$$

$$\leq d(\overline{V}^{T-1}, W^{T-1}) \cdot \|d^t - \tilde{d}^t\| \cdot \|\Pi_{W^T}(\overline{V}^{T-1}) - \Pi_{W^{T-1}}(\overline{V}^{T-1})\|$$

$$+ d(\overline{V}^{T-1}, W^{T-1}) \cdot \left\langle \tilde{d}^t, \Pi_{W^T}(\overline{V}^{T-1}) - \Pi_{W^{T-1}}(\overline{V}^{T-1}) \right\rangle.$$

By the definition of $\tilde{d}^t$, we know that

$$\langle \tilde{d}^t, \Pi_{W^T}(\overline{V}^{T-1}) \rangle = \min_{x \in W^T} \langle \tilde{d}^t, x \rangle \leq \langle \tilde{d}^t, \Pi_{W^T}(\Pi_{W^{T-1}}(\overline{V}^{T-1})) \rangle$$

$$\leq d_{B_1}(W^T, W^{T-1}) + \langle \tilde{d}^t, \Pi_{W^{T-1}}(\overline{V}^{T-1}) \rangle.$$

Hence the inner product term can be further bounded by

$$d(\overline{V}^{T-1}, W^{T-1}) \cdot \left\langle d^t, \Pi_{W^T}(\overline{V}^{T-1}) - \Pi_{W^{T-1}}(\overline{V}^{T-1}) \right\rangle$$

$$\leq \left( 4\sqrt{d(\overline{V}^{T-1}, W^{T-1}) d_{B_1}(W^{T-1}, W^T)} + 2d_{B_1}(W^{T-1}, W^T) \right)^2$$

$$+ d(\overline{V}^{T-1}, W^{T-1}) \cdot d_{B_1}(W^T, W^{T-1})$$

$$\leq 33 d(\overline{V}^{T-1}, W^{T-1}) \cdot d_{B_1}(W^T, W^{T-1}) + 8 d_{B_1}^2(W^{T-1}, W^T). \tag{27}$$

Now continue to bound the right side in Eq. (26), we can further get that

$$T^2 d(\overline{V}^T, W^T)^2 \leq T^2 \|\overline{V}^T - \Pi_{W^{T-1}}(\overline{V}^{T-1})\|^2 + 37 T^2 d(\overline{V}^{T-1}, W^{T-1}) \cdot d_{B_1}(W^T, W^{T-1}) + 10 T^2 d_{B_1}^2(W^{T-1}, W^T). \tag{28}$$

Now we can further bound the Eq. (28) by expanding the first term on the right side:

$$T^2 \|\overline{V}^T - \Pi_{W^{T-1}}(\overline{V}^{T-1})\|^2 = (T-1)^2 \|\overline{V}^{T-1} - \Pi_{W^{T-1}}(\overline{V}^{T-1})\|^2 + \|V^T - \Pi_{W^{T-1}}(\overline{V}^{T-1})\|^2$$

$$+ 2(T-1) \left\langle \overline{V}^{T-1} - \Pi_{W^{T-1}}(\overline{V}^{T-1}), V^T - \Pi_{W^{T-1}}(\overline{V}^{T-1}) \right\rangle \tag{29}$$

$$\leq (T-1)^2 \|\overline{V}^{T-1} - \Pi_{W^{T-1}}(\overline{V}^{T-1})\|^2 + (B + B_1)^2 m$$

$$+ 2(T-1) \left\langle \overline{V}^{T-1} - \Pi_{W^{T-1}}(\overline{V}^{T-1}), V^T - \Pi_{W^{T-1}}(\overline{V}^{T-1}) \right\rangle. \tag{30}$$

The inner product term is

$$\left\langle \overline{V}^{T-1} - \Pi_{W^{T-1}}(\overline{V}^{T-1}), V^T - \Pi_{W^{T-1}}(\overline{V}^{T-1}) \right\rangle = d(\overline{V}^{T-1}, W^{T-1}) \cdot \left\langle d^{T-1}, \Pi_{W^{T-1}}(\overline{V}^{T-1}) - V^T \right\rangle.$$

Note that $\langle d^{T-1}, \Pi_{W^{T-1}}(\overline{V}^{T-1}) \rangle = \min_{z \in W^{T-1}} \langle d^{T-1}, z \rangle$. Because $\|\Pi_{W^*}(\overline{V}^{T-1})\| \leq B_1$, there is a $z' \in W^{T-1}$ such that $\|z' - \Pi_{W^*}(\overline{V}^{T-1})\| \leq d_{B_1}(W^*, W^{T-1})$. Hence,

$$\langle d^{T-1}, \Pi_{W^{T-1}}(\overline{V}^{T-1}) \rangle \leq \min_{z \in W^{T-1}} \langle d^{T-1}, z \rangle \leq \langle d^{T-1}, z' \rangle \leq \min_{z \in W^*} \langle d^{T-1}, z \rangle + d_{B_1}(W^*, W^{T-1})$$

$$\leq J(r^*, d^{T-1}, \pi^*) + D(\pi^*) + d_{B_1}(W^*, W^{T-1}).$$

The last inequality holds by Lemma B.1. Now we continue to bound the inner product term. We have

$$\left\langle \overline{V}^{T-1} - \Pi_{W^{T-1}}(\overline{V}^{T-1}), V^T - \Pi_{W^{T-1}}(\overline{V}^{T-1}) \right\rangle$$

$$= d(\overline{V}^{T-1}, W^{T-1}) \cdot \left\langle d^{T-1}, \Pi_{W^{T-1}}(\overline{V}^{T-1}) - V^T \right\rangle$$

$$\leq d(\overline{V}^{T-1}, W^{T-1}) \cdot (\|d^{T-1}\|_1 \cdot J(r_1^*, \cdots, r_m^*, \overline{d^{T-1}}, \pi^*) + D(\pi^*) + d_{B_1}(W^{T-1}, W^*) - J(\hat{r}_1^t, \cdots, \hat{r}_m^t, d^{T-1}, \pi^t))$$

$$\leq d(\overline{V}^{T-1}, W^{T-1}) \cdot \left( \eta \|d^{T-1}\|_1 \cdot \left( \sum_{i=1}^{m} L_i^{T-1}(\theta^*) - \sum_{i=1}^{m} L_i^{T-1}(\theta^{T-1}) \right) + D(\pi^*) + d_{B_1}(W^{T-1}, W^*) \right).$$

Thus the Eq. (28) can be rewritten as

$$T^2 d(\overline{V}^T, W^T)^2$$
$$\leq (T-1)^2 \|\overline{V}^{T-1} - \Pi_{W^{T-1}}(\overline{V}^{T-1})\|^2 + (B + B_1)^2 m + 10 T^2 d_{B_1}^2(W^{T-1}, W^T)$$
$$+ 2(T-1) d(\overline{V}^{T-1}, W^{T-1}) \cdot$$
$$\left( \eta \|d^t\|_1 \cdot \left( \sum_{i=1}^{m} L_i^{T-1}(\theta^*) - \sum_{i=1}^{m} L_i^{T-1}(\theta^t) \right) + D(\pi^*) + d_{B_1}(W^{T-1}, W^*) + 37 T d_{B_1}(W^T, W^{T-1}) \right).$$

Then by the recursion, we can get

$$T d(\overline{V}^T, W^T)^2$$
$$\leq (B + B_1)^2 m + \sum_{t=1}^{T} \frac{10 t^2 d_{B_1}^2(W^{t-1}, W^t)}{T}$$
$$+ \sum_{t=1}^{T} \frac{2(t-1)}{T} d(\overline{V}^{t-1}, W^{t-1}) \cdot \left( \eta \|d^{T-1}\|_1 \cdot \left( \sum_{i=1}^{m} L_i^{T-1}(\theta^*) - \sum_{i=1}^{m} L_i^{T-1}(\theta^{T-1}) \right) \right.$$
$$\left. + D(\pi^*) + d_{B_1}(W^{t-1}, W^*) + 37 t d_{B_1}(W^t, W^{t-1}) \right).$$

By this recursion formula, we can use the induction method to prove that

$$d(\overline{V}^T, W^T) \leq \frac{(B + B_1)^2 m}{\sqrt{T}} + \underbrace{\sum_{t=1}^{T} \frac{10 t^2}{T^{3/2}} d_{B_1}^2(W^{t-1}, W^t)}_{(A)} + D(\pi^*) + \underbrace{\frac{\eta}{T} \sum_{t=1}^{T-1} \|d^t\|_1 \sum_{i=1}^{m} (L_i^t(\theta^*) - L_i^t(\theta^t))}_{(B)}$$
$$+ \underbrace{\frac{1}{T} \sum_{i=1}^{T} d_{B_1}(W^{t-1}, W^*)}_{(C)} + \underbrace{\frac{1}{T} \sum_{t=1}^{T} 37 t d_{B_1}(W^t, W^{t-1})}_{(D)}.$$

Now we bound all four terms. We first prove that term (A), (C) and (D) are all at level $\widetilde{\mathcal{O}}(1/\sqrt{T})$.

**Term (A):** First we consider term (A). Since $W^t = \bigcap_{n=1}^{N} W_{p^{(n)}, c^{(n)}}^{\alpha^{t,(n)}}$, the term $d_{B_1}^2(W^{t-1}, W^t)$ can be bounded by

$$d_{B_1}^2(W^{t-1}, W^t) \leq \left( \sum_{n=1}^{N} d_{B_1} \left( W_{p^{(n)}, c^{(n)}}^{\alpha^{t-1,(n)}}, W_{p^{(n)}, c^{(n)}}^{\alpha^{t,(n)}} \right) \right)^2 \leq N \sum_{n=1}^{N} d_{B_1}^2 \left( W_{p^{(n)}, c^{(n)}}^{\alpha^{t-1,(n)}}, W_{p^{(n)}, c^{(n)}}^{\alpha^{t,(n)}} \right).$$

Since $\alpha^t = \frac{t-1}{t} \alpha^{t-1} + \frac{1}{t} \hat{\alpha}^t$, we can know $\|\alpha^t - \alpha^{t-1}\|_\infty \leq \frac{1}{t} \|\hat{\alpha}^t\|_\infty \leq \frac{1}{t}$. Then, by Lemma C.2, we have

$$d_{B_1}(W_{p^{(n)}, c^{(n)}}^{\alpha^{t-1,(n)}}, W_{p^{(n)}, c^{(n)}}^{\alpha^{t,(n)}}) \leq \frac{m^{3/2} B_1}{|p^{(n)}|} \cdot \frac{1}{t}. \tag{31}$$

Thus by Eq. (31), we know that

$$(A) \leq \frac{10N}{T^{3/2}} \sum_{n=1}^{N} \sum_{t=1}^{T} \frac{m^3 B_1^2}{(p^{(n)})^2}$$
$$\leq \sum_{n=1}^{N} \frac{10 N m^3 B_1^2}{(p^{(n)})^2} \cdot \frac{1}{\sqrt{T}}. \tag{32}$$

**Term (C):** We have

$$(C) \le \frac{B_1}{T} + \frac{1}{T} \cdot \sum_{n=1}^{N} \frac{m^{3/2} B_1}{|p^{(n)}|} \cdot \sum_{t=2}^{T} \|\alpha^{t-1,(n)} - \alpha^*\|_\infty$$

$$\le \frac{1}{T} \cdot \sum_{n=1}^{N} \frac{m^{3/2} B_1}{|p^{(n)}|} \cdot \gamma^{-1} \exp(4/\beta) \cdot \widetilde{\mathcal{O}} \left(\mathrm{poly}(m, e^B, d, \log(1/\delta))\right) \cdot \left(\sum_{t=1}^{T} \frac{1}{\sqrt{t}} + 1\right)$$

$$\le \frac{1}{\sqrt{T}} \cdot \frac{N m^{3/2} B_1}{\min_{n \in [N]} |p^{(n)}|} \cdot \gamma^{-1} \exp(4/\beta) \cdot \widetilde{\mathcal{O}} \left(\mathrm{poly}(m, e^B, d, \log(1/\delta))\right). \tag{33}$$

**Term (D):** First, we have

$$(D) \le \frac{1}{T} \sum_{t=1}^{T} 37t \sum_{n=1}^{N} d_{B_1}(W^{t,(n)}, W^{t-1,(n)}).$$

Then, by Lemma C.2,

$$(D) \le \frac{37 B_1}{T} + \frac{1}{T} \sum_{t=2}^{T} \sum_{n=1}^{N} \frac{37 t m^{3/2} B_1}{|p^{(n)}|} \|\alpha^{t,(n)} - \alpha^{t-1,(n)}\|_\infty$$

$$\le \frac{37 m^{3/2} B_1}{T} \cdot \sum_{n=1}^{N} \frac{1}{|p^{(n)}|} \sum_{t=2}^{T} \left(\left\|\hat{\alpha}^{t,(n)} - \alpha^{t-1,(n)}\right\|_\infty + 1\right)$$

$$\le \frac{37 m^{3/2} B_1}{T} \cdot \sum_{n=1}^{N} \frac{1}{|p^{(n)}|} \left(\sum_{t=2}^{T} \left(\|\hat{\alpha}^{t,(n)} - \alpha^{*,(n)}\|_\infty + \left\|\alpha^{*,(n)} - \alpha^{t-1,(n)}\right\|_\infty\right) + 1\right)$$

$$\le \frac{37 m^{3/2} B_1}{T} \cdot \sum_{n=1}^{N} \frac{1}{|p^{(n)}|} \left(\underbrace{\sum_{t=2}^{T} \|\hat{\alpha}^{t,(n)} - \alpha^{*,(n)}\|_\infty}_{(E)} + \underbrace{\sum_{t=2}^{T} \|\alpha^{t-1,(n)} - \alpha^{*,(n)}\|}_{(F)}\right) + \frac{37 m^{3/2} B_1}{T}. \tag{34}$$

For the term (E), by Eq. (57), we have

$$(E) \le \sum_{t=1}^{T} \|\hat{\alpha}^{t,(n)} - \alpha^{*,(n)}\|_\infty \le \gamma^{-1} \cdot \widetilde{\mathcal{O}} \left(\mathrm{poly}(m, e^B, \exp(1/\beta), d, \log(1/\delta))\right) \cdot \sqrt{T}.$$

Also by Eq. (57),

$$(F) \le \sum_{t=2}^{T} \gamma^{-1} \cdot \widetilde{\mathcal{O}} \left(\mathrm{poly}(m, e^B, \exp(1/\beta) d, \log(1/\delta))\right) \cdot \frac{1}{\sqrt{t}}$$

$$\le \gamma^{-1} \cdot \widetilde{\mathcal{O}} \left(\mathrm{poly}(m, e^B, \exp(1/\beta), d, \log(1/\delta))\right) \cdot \sqrt{T}.$$

By Theorem B.4, the term (E) can be bounded by

$$(E) \le \gamma^{-1} \mathrm{poly}(\exp(1/\beta), m, e^B, d, \log(1/\delta)) \widetilde{\mathcal{O}}(\sqrt{T}).$$

Thus substitute these upper bound to the Eq. (34), we get

$$(D) \le \frac{1}{T} \sum_{t=1}^{T} 37t \sum_{n=1}^{N} d_{B_1}(W^{t,(n)}, W^{t-1,(n)})$$

$$\le \gamma^{-1} \mathrm{poly}(\exp(1/\beta), m, N, e^B, d, \log(1/\delta), B_1, (\min_{n \in [N]} p^{(n)})^{-1}) \cdot \widetilde{\mathcal{O}}(1/\sqrt{T}). \tag{35}$$

**Combine them:** Now we combine the upper bound of (A), (C), (D), i.e., Eq. (32), (33), (35), we can get

$$d(\overline{V}^T, W^T) \leq \frac{(B+B_1)^2 m}{\sqrt{T}} + \gamma^{-1} \text{poly}(\exp(1/\beta), m, e^B, d, \log(1/\delta), \min_{n \in [N]} \frac{1}{p^{(n)}}, B_1) \widetilde{\mathcal{O}}(1/\sqrt{T}) + D(\pi^*) + \text{(B)}. \tag{36}$$

Now we consider the proof of Theorem 5.4.

$$\begin{aligned}
&D(\tilde{\pi}^T) - D(\pi^*) \\
&= d(W^*, \mathbb{E}_{\tilde{\pi}^T}[r^*(x,y)] - \beta \mathbb{D}_{\text{KL}}(\tilde{\pi}^T \| \pi_{\text{ref}})) - D(\pi^*) \\
&\leq d(W^*, \mathbb{E}_{\tilde{\pi}^T}[r^*(x,y)] - \frac{\beta}{T} \sum_{t=1}^{T} \mathbb{D}_{\text{KL}}(\pi^t \| \pi_{\text{ref}})) - D(\pi^*) \\
&= \underbrace{d(W^*, \mathbb{E}_{\tilde{\pi}^T}[r^*(x,y)] - \frac{\beta}{T} \sum_{t=1}^{T} \mathbb{D}_{\text{KL}}(\pi^t \| \pi_{\text{ref}})) - d(W^*, \frac{1}{T} \sum_{t=1}^{T} \mathbb{E}_{\pi^t}[\hat{r}^t(x,y)] - \frac{\beta}{T} \sum_{t=1}^{T} \mathbb{D}_{\text{KL}}(\pi^t \| \pi_{\text{ref}}))}_{(*)} \\
&\quad + d(W^*, \frac{1}{T} \sum_{t=1}^{T} \mathbb{E}_{\pi^t}[\hat{r}^t(x,y)] - \frac{\beta}{T} \sum_{t=1}^{T} \mathbb{D}_{\text{KL}}(\pi^t \| \pi_{\text{ref}})) - D(\pi^*) \\
&= (*) + d\left(W^*, \overline{V}^T\right) - D(\pi^*) \\
&\leq (*) + d(W^*, W^T) + \underbrace{d\left(W^T, \overline{V}^T\right) - D(\pi^*)}_{(**)}.
\end{aligned}$$

**Term (∗):** First, the term (∗) can be bounded by

$$(*) \leq \sum_{i=1}^{m} \left| \frac{1}{T} \sum_{t=1}^{T} \mathbb{E}_{\pi^t} \left[ \hat{r}_i^t(x,y) - r_i^*(x,y) \right] \right|.$$

Now note that

$$\frac{1}{T} \sum_{t=1}^{T} \mathbb{E}_{\pi^t} \left[ \hat{r}_i^t(x,y) - r_i^*(x,y) \right] = \frac{1}{T} \sum_{t=1}^{T} \mathbb{E}_{y_1 \sim \pi^t, y_2 \sim \pi_{\text{base}}} \left[ \left( (\hat{r}_i^t(x,y_1) - r_i^t(x,y_2)) - (r_i^*(x,y_1) - r_i^*(x,y_2)) \right) \right]. \tag{37}$$

Now since the reward contains a linear structure, by Lemma D.3 with $d_{\text{cover}}(1/T) = \widetilde{\mathcal{O}}(d)$, for any $\mu_i > 0$ we can derive that

$$\begin{aligned}
(*) &\leq \sum_{i=1}^{m} \mu_i \cdot \sum_{t=1}^{T} \sum_{j=1}^{t-1} \mathbb{E}_{y_1 \sim \pi^j, y_2 \sim \pi_{\text{base}}} [(r_i^t(x,y_1) - r_i^t(x,y_2) - (r_i^*(x,y_1) - r_i^*(x,y_2)))^2] + \frac{d_{\text{cover}}(1/T)}{4\mu_i} + \widetilde{\mathcal{O}}(Bd) \\
&\leq \sum_{i=1}^{m} \mu_i \exp(4/\beta) \kappa \cdot \sum_{t=1}^{T} \sum_{j=1}^{t-1} \mathbb{E}_{y_1 \sim \pi^j, y_2 \sim \pi^j} [(r_i^t(x,y_1) - r_i^t(x,y_2) - (r_i^*(x,y_1) - r_i^*(x,y_2)))^2] + \frac{d_{\text{cover}}(1/T)}{4\mu_i} + \widetilde{\mathcal{O}}(Bd) \\
&= \sum_{i=1}^{m} \mu_i \exp(4/\beta) \kappa \cdot \sum_{t=1}^{T} \sum_{j=1}^{t-1} \mathbb{E}_{y_1, y_2 \sim \pi^j} [\Delta_i^t(x,y)^2] + \frac{d_{\text{cover}}(1/T)}{4\mu_i} + \widetilde{\mathcal{O}}(Bd),
\end{aligned} \tag{38}$$

The last inequality uses the fact that

$$\sup_{x,y} \frac{\pi_{\text{base}}(y \mid x)}{\pi^j(y \mid x)} \leq \sup_{x,y} \frac{\pi_{\text{base}}(y \mid x)}{\pi_{\text{ref}}(y \mid x)} \cdot \sup_{x,y} \frac{\pi_{\text{ref}}(y \mid x)}{\pi^j(y \mid x)} \leq \exp(4/\beta) \cdot \kappa,$$

where $\kappa = \sup_{x,y} \frac{\pi_{\text{base}}(y|x)}{\pi_{\text{ref}}(y|x)}$ (Cen et al., 2024).

**Term (∗∗):** Now we consider the term (∗∗). By Eq. (36), we know that

$$(**) \leq \frac{(B+B_1)^2 m}{\sqrt{T}} + \gamma^{-1} \text{poly}(\exp(1/\beta), m, e^B, d, \log(1/\delta), \min_{n \in [N]} \frac{1}{p^{(n)}}, B_1) \widetilde{\mathcal{O}}(1/\sqrt{T})$$

$$+ \frac{1}{T} \sum_{t=1}^{T} \sum_{i=1}^{m} \eta \|d^t\|_1 (L_i^t(\theta_i^*) - L_i^t(\theta_i^t)). \tag{39}$$

Now by the MLE loss, there exists a constant $C'$ such that

$$\frac{1}{T} \sum_{t=1}^{T} \sum_{i=1}^{m} \eta \|d^t\|_1 (L_i^t(\theta_i^*) - L_i^t(\theta_i^t))$$

$$\leq 2 \sum_{i=1}^{m} \eta \|d^t\|_1 \log(|\mathcal{R}|/\delta) - \frac{C'}{T} \sum_{t=1}^{T} \sum_{i=1}^{m} \eta \|d^t\|_1 \sum_{j \in \mathcal{D}_i^{t-1}} \mathbb{E}_{y \sim \pi^j} \left[ \Delta_i^t(x,y)^2 \right]$$

$$= \widetilde{\mathcal{O}}(2m\eta\sqrt{m}d) - \frac{C'}{T} \sum_{t=1}^{T} \sum_{i=1}^{m} \eta \|d^t\|_1 \sum_{j \in \mathcal{D}_i^{t-1}} \mathbb{E}_{y \sim \pi^j} \left[ \Delta_i^t(x,y)^2 \right], \tag{40}$$

Now consider the second term in Eq. (40). We can bound it by

$$\sum_{t=1}^{T} \sum_{i=1}^{m} \sum_{j \in \mathcal{D}_i^{t-1}} \mathbb{E}_{y_1, y_2 \sim \pi^j} [\Delta_i^t(x,y)^2]$$

$$= \sum_{t=1}^{T} \sum_{i=1}^{m} \sum_{j=1}^{t-1} \mathbb{E}_{y_1, y_2 \sim \pi^j, I \sim \mathbb{P}(\cdot | \alpha^*, x, y_1, y_2, r^*)} [\Delta_i^t(x,y)^2 \mathbb{I}\{I^j = i\}]$$

$$\geq \kappa_1 \sum_{i=1}^{m} \sum_{j=1}^{T} \sum_{t=j+1}^{T} \mathbb{E}_{y_1, y_2 \sim \pi^j, I=i} [\Delta_i^t(x^j, y^j)^2 \mathbb{I}\{I^j = i\}]$$

$$= \kappa_1 \sum_{i=1}^{m} \sum_{j=1}^{T} \sum_{t=j+1}^{T} \mathbb{E}_{y_1, y_2 \sim \pi^j} [\Delta_i^t(x,y)^2]$$

$$= \kappa_1 \cdot \sum_{i=1}^{m} \sum_{t=1}^{T} \sum_{j=1}^{t-1} \mathbb{E}_{y_1, y_2 \sim \pi^j} [\Delta_i^t(x,y)^2], \tag{41}$$

where the inequality uses the fact that $\inf_{y,x,j,I} \frac{1}{\mathbb{P}(I|\alpha^*, x, y_1, y_2, r^*)} = \kappa_1$ for some constant $\kappa_1$. Since the distribution of index is a bounded softmax distribution, we can derive that $\kappa_1 \geq \frac{e^0}{e^0 + (m-1)e^B} \geq \frac{1}{me^B}$. Thus we can get

$$\sum_{t=1}^{T} \sum_{i=1}^{m} \sum_{j \in \mathcal{D}_i^{t-1}} \mathbb{E}_{y_1, y_2 \sim \pi^j} [\Delta_i^t(x,y)^2] \geq \frac{1}{me^B} \cdot \sum_{i=1}^{m} \sum_{t=1}^{T} \sum_{j=1}^{t-1} \mathbb{E}_{y_1, y_2 \sim \pi^j} [\Delta_i^t(x,y)^2].$$

Hence, the Eq. (40) can be further bounded by

$$\frac{1}{T} \sum_{t=1}^{T} \sum_{i=1}^{m} \eta \|d^t\|_1 (L_i^t(\theta_i^*) - L_i^t(\theta_i^t)) \leq \widetilde{\mathcal{O}}(2m\eta\sqrt{m}d) - \frac{\eta C' \|d^t\|_1}{Tme^B} \sum_{i=1}^{m} \sum_{t=1}^{T} \sum_{j=1}^{t-1} \mathbb{E}_{y_1, y_2 \sim \pi^j} [\Delta_i^t(x,y)^2] \tag{42}$$

$$\leq \widetilde{\mathcal{O}}(2m\eta\sqrt{m}d) - \frac{\eta C'}{Tme^B} \sum_{i=1}^{m} \sum_{t=1}^{T} \sum_{j=1}^{t-1} \mathbb{E}_{y_1, y_2 \sim \pi^j} [\Delta_i^t(x,y)^2]. \tag{43}$$

The last inequality uses the fact that $\|d^t\|_1 \geq 1$. Now combining (∗) (Eq. (38)) and (∗∗) (Eq. (39)), by choosing $\mu_i = \frac{C'}{me^B \exp(4/\beta)\kappa\sqrt{T}}$, we can get

$$D(\tilde{\pi}^T) - D(\pi^*)$$

$$\leq (*) + (**) + d(W^*, W^T)$$

$$\leq \frac{me^B \exp(4/\beta)\kappa d_{\text{cover}}(1/T)}{4C'\sqrt{T}} + \frac{(B + B_1)^2 m}{\sqrt{T}} + \widetilde{\mathcal{O}}(Bd) + \widetilde{\mathcal{O}}\left(\frac{m^{3/2}d}{\sqrt{T}}\right) + d(W^*, W^T). \tag{44}$$

Note that

$$d(W^*, W^T) \leq \sum_{n=1}^{N} d_{B_1}(W_{p^{(n)},c^{(n)}}^{\alpha^{T,(n)}}, W_{p^{(n)},c^{(n)}}^{\alpha^{*,(n)}}) \leq m^{3/2}B_1 \sum_{n=1}^{N} \frac{1}{|p^{(n)}|} \cdot \|\alpha^{T,(n)} - \alpha^{*,(n)}\|_\infty$$

$$\leq \frac{m^{3/2}B_1}{\sqrt{T}} \cdot \left(\sum_{n=1}^{N} \frac{1}{p^{(n)}}\right) \cdot \gamma^{-1}\text{poly}(\exp(1/\beta), m, e^B, d, \log(1/\delta)) \tag{45}$$

$$\leq \frac{m^{3/2}B_1 N}{\sqrt{T}} \cdot (\min_{n\in[N]} p^{(n)})^{-1} \cdot \gamma^{-1}\text{poly}(\exp(1/\beta), m, e^B, d, \log(1/\delta)), \tag{46}$$

where the inequality Eq. (45) holds by Theorem (B.4).

Hence, combining Eq. (44) and Eq. (46), we complete the proof. $\square$

### B.5. Proof of Theorem 5.5

*Proof.* First, note that

$$d(\overline{V}^T, W^{T,(n)})^{2q} = \|\overline{V}^T - \Pi_{W^{T,(n)}}(\overline{V}^T)\|^{2q}$$

$$\leq \|\overline{V}^T - \Pi_{W^{T,(n)}}(\overline{V}^{T-1})\|^{2q}$$

$$\leq \left( \|\overline{V}^T - \Pi_{W^{T-1,(n)}}(\overline{V}^{T-1})\|^2 + \|\Pi_{W^{T,(n)}}(\overline{V}^{T-1}) - \Pi_{W^{T-1,(n)}}(\overline{V}^{T-1})\|^2 \right.$$

$$\left. + 2\langle \overline{V}^T - \Pi_{W^{T-1,(n)}}(\overline{V}^{T-1}), \Pi_{W^{T,(n)}}(\overline{V}^{T-1}) - \Pi_{W^{T-1,(n)}}(\overline{V}^{T-1})\rangle \right)^q. \tag{47}$$

Now by Lemma C.3, since $\|V^{T-1}\|_\infty \leq B$ is bounded, we have

$$\|\Pi_{W^{T,(n)}}(\overline{V}^{T-1}) - \Pi_{W^{T-1,(n)}}(\overline{V}^{T-1})\|_2^2 \leq 4d(\overline{V}^{T-1}, W^{T-1,(n)})d_{B_1}(W^{T,(n)}, W^{T-1,(n)}) + 2d_{B_1}^2(W^{T,(n)}, W^{T-1,(n)}). \tag{48}$$

Also, by Eq. (27), we can also have

$$\langle \overline{V}^T - \Pi_{W^{T-1,(n)}}(\overline{V}^{T-1}), \Pi_{W^{T,(n)}}(\overline{V}^{T-1}) - \Pi_{W^{T-1,(n)}}(\overline{V}^{T-1})\rangle$$

$$\leq 33d(\overline{V}^{T-1}, W^{T-1}) \cdot d_{B_1}(W^{T,(n)}, W^{T-1,(n)}) + 8d_{B_1}^2(W^{T-1,(n)}, W^{T,(n)}).$$

Hence, by Eq. (47), we can get

$$T^{2q}d(\overline{V}^T, W^{T,(n)})^{2q}$$

$$\leq \left( \|\overline{V}^T - \Pi_{W^{T-1,(n)}}(\overline{V}^{T-1})\|^2 + \|\Pi_{W^{T,(n)}}(\overline{V}^{T-1}) - \Pi_{W^{T-1,(n)}}(\overline{V}^{T-1})\|^2 \right.$$

$$\left. + 2\langle \overline{V}^T - \Pi_{W^{T-1,(n)}}(\overline{V}^{T-1}), \Pi_{W^{T,(n)}}(\overline{V}^{T-1}) - \Pi_{W^{T-1,(n)}}(\overline{V}^{T-1})\rangle \right)^q.$$

$$\leq T^{2q}\left( \|\overline{V}^T - \Pi_{W^{T-1,(n)}}(\overline{V}^{T-1})\|^2 + 37d(\overline{V}^{T-1}, W^{T-1}) \cdot d_{B_1}(W^{T,(n)}, W^{T-1,(n)}) + 10d_{B_1}^2(W^{T-1,(n)}, W^{T,(n)})\right)^q. \tag{49}$$

Now, since $d(W^{T,(n)}, W^{T-1,(n)}) \le \frac{m^{3/2}B_1}{|p^{(n)}|} \cdot \|\alpha^{t,(n)} - \alpha^{t-1,(n)}\|_\infty \le \frac{m^{3/2}B_1}{|p^{(n)}|T}$, we know

$$37d(\overline{V}^{T-1}, W^{T-1}) \cdot d_{B_1}(W^{T,(n)}, W^{T-1,(n)}) + 10d_{B_1}^2(W^{T-1,(n)}, W^{T,(n)}) \le \frac{37B_1^2 m^{3/2}}{|p^{(n)}|T} + \frac{10B_1^2 m^3}{|p^{(n)}|^2 T^2}. \tag{50}$$

Hence, the Eq. (49) can be further bounded by

$$T^{2q}d(\overline{V}^T, W^{T,(n)})^{2q}$$
$$\le T^{2q}d^{2q}(\overline{V}^T, W^{T-1,(n)}) + \widetilde{\mathcal{O}}(\text{poly}(B_1^q, m^q, (\min_{n\in[N]} p^{(n)})^{-q})T^{2q-2})$$
$$+ qT^{2q}\|\overline{V}^T - \Pi_{W^{T-1,(n)}}(\overline{V}^{T-1})\|^{2q-2}.$$
$$\left(37d(\overline{V}^{T-1}, W^{T,(n)}) \cdot d_{B_1}(W^{T,(n)}, W^{T-1,(n)}) + 10d_{B_1}^2(W^{T-1,(n)}, W^{T,(n)})\right)$$
$$\le T^{2q}d^{2q}(\overline{V}^T, W^{T-1,(n)}) + \widetilde{\mathcal{O}}(\text{poly}(B_1^q, m^q, (\min_{n\in[N]} p^{(n)})^{-q})T^{2q-2})$$
$$+ 37qT^{2q}d^{2q-1}(\overline{V}^{T-1}, W^{T-1,(n)}) \cdot d_{B_1}(W^{T,(n)}, W^{T-1,(n)}).$$

The last inequality is because $\|P_{T,n}\| = \text{poly}(B_1, m, (\min_{n\in[N]} p^{(n)})^{-1} \cdot \widetilde{\mathcal{O}}(1/T)$, and

$$\|\overline{V}^T - \Pi_{W^{T-1,(n)}}(\overline{V}^{T-1})\|^{2q-2} - d^{2q-2}(\overline{V}^{T-1}, W^{T-1,(n)}) \le \widetilde{\mathcal{O}}(\text{poly}(B_1^q, m^q, (\min_{n\in[N]} p^{(n)})^{-q})T^{2q-3}).$$

Now we further bound the first term $T^{2q}d^{2q}(\overline{V}^T, W^{T-1,(n)})$. Using the same derivation for Eq. (20), we know that

$$T^{2q}\sum_{n=1}^N \zeta_n d^{2q}(\overline{V}^T, W^{T-1,(n)})$$
$$\le (T-1)^{2q}\sum_{n=1}^N \zeta_n d^{2q}(\overline{V}^{T-1}, W^{T-1,(n)}) + 12^q T^{2q-2}B_1^{2q}m^q$$
$$+ 2(T-1)^{2q-1}\sum_{n=1}^N \zeta_n(\overline{V}^{T-1} - \Pi_{W^{T-1,(n)}}(\overline{V}^{T-1}))(V^T - \Pi_{W^{T-1,(n)}}(\overline{V}^{T-1}))d^{2q-2}(\overline{V}^{T-1}, W^{T-1,(n)}).$$

Hence, we can derive

$$T^{2q}\sum_{n=1}^N \zeta_n d(\overline{V}^T, W^{T,(n)})^{2q}$$
$$\le (T-1)^{2q}\sum_{n=1}^N \zeta_n d^{2q}(\overline{V}^{T-1}, W^{T-1,(n)}) + \widetilde{\mathcal{O}}(\text{poly}(B_1^q, m^q, (\min_{n\in[N]} p^{(n)})^{-q})T^{2q-2})$$
$$+ 37qT^{2q}\sum_{n=1}^N \zeta_n d^{2q-1}(\overline{V}^{T-1}, W^{T-1,(n)}) \cdot d_{B_1}(W^{T,(n)}, W^{T-1,(n)})$$
$$+ 2(T-1)^{2q-1}\sum_{n=1}^N \zeta_n(\overline{V}^{T-1} - \Pi_{W^{T-1,(n)}}(\overline{V}^{T-1}))(V^T - \Pi_{W^{T-1,(n)}}(\overline{V}^{T-1}))d^{2q-2}(\overline{V}^{T-1}, W^{T-1,(n)}).$$

Now we consider the last term in the inequation above. Similar to the Eq. (22), we have

$$\sum_{n=1}^N \zeta_n(\overline{V}^{T-1} - \Pi_{W^{T-1,(n)}}(\overline{V}^{T-1}))(V^T - \Pi_{W^{T-1,(n)}}(\overline{V}^{T-1}))d^{2q-2}(\overline{V}^{T-1}, W^{T-1,(n)})$$
$$\le \left(\sum_{n=1}^N \zeta_n d^{2q}(\overline{V}^{T-1}, W^{T-1,(n)})\right)^{\frac{2q-1}{2q}} \cdot \left(D_q(\pi^*) + \eta\|d^t\|_1 \underbrace{\left(\sum_{i=1}^m L_i^t(\theta^*) - \sum_{i=1}^m L_i^t(\theta^t)\right)}_{(*)}\right),$$

then we can get

$$T^{2q} \sum_{n=1}^{N} \zeta_n d(\overline{V}^T, W^{T,(n)})^{2q}$$

$$\leq (T-1)^{2q} \sum_{n=1}^{N} \zeta_n d^{2q}(\overline{V}^{T-1}, W^{T-1,(n)}) + \widetilde{\mathcal{O}}(\mathrm{poly}(B_1^q, m^q, (\min_{n \in [N]} p^{(n)})^{-q})T^{2q-2})$$

$$+ 37q T^{2q} \left( \sum_{n=1}^{N} \zeta_n d^{2q}(\overline{V}^{T-1}, W^{T-1,(n)}) \right)^{\frac{2q-1}{2q}} \cdot \sqrt[2q]{\sum_{n=1}^{N} d_{B_1}^{2q}(W^{T,(n)}, W^{T-1,(n)})}$$

$$2(T-1)^{2q-1} \left( \sum_{n=1}^{N} \zeta_n d^{2q}(\overline{V}^{T-1}, W^{T-1,(n)}) \right)^{\frac{2q-1}{2q}} \cdot (D_q(\pi^*) + (*))$$

$$\leq (T-1)^{2q} \sum_{n=1}^{N} \zeta_n d^{2q}(\overline{V}^{T-1}, W^{T-1,(n)}) + \widetilde{\mathcal{O}}(\mathrm{poly}(B_1^q, m^q, (\min_{n \in [N]} p^{(n)})^{-q})T^{2q-2})$$

$$+ 2(T-1)^{2q-1} \left( \sum_{n=1}^{N} \zeta_n d^{2q}(\overline{V}^{T-1}, W^{T-1,(n)}) \right)^{\frac{2q-1}{2q}}$$

$$\cdot \left( \frac{37q T^{2q}}{(T-1)^{2q-1}} \sqrt[2q]{\sum_{n=1}^{N} 2^q T d_{B_1}^{2q}(W^{T,(n)}, W^{T-1,(n)}) + D_q(\pi^*) + (*)} \right).$$

Hence, by the reduction and the fact that $\frac{T}{T-1} \leq 2$ for $T \geq 2$,, we can further get

$$T^{2q} \sum_{n=1}^{N} \zeta_n d^{2q}(\overline{V}^T, W^{T,(n)})$$

$$\leq \widetilde{\mathcal{O}}(\mathrm{poly}(B_1^q, m^q, (\min_{n \in [N]} p^{(n)})^{-q})T^{2q-1}) + \sum_{t=1}^{T} 2(t-1)^{2q-1} \left( \sum_{n=1}^{N} \zeta_n d^{2q}(\overline{V}^{T-1}, W^{T-1,(n)}) \right)^{\frac{2q-1}{2q}}$$

$$\cdot \left( 37q \cdot 2^q T \cdot \sqrt[2q]{\sum_{n=1}^{N} d_{B_1}^{2q}(W^{T,(n)}, W^{T-1,(n)})} + D_q(\pi^*) + \sqrt[2q]{\sum_{n=1}^{N} d_{B_1}^{2q}(W^{T-1,(n)}, W^*)} + (*) \right).$$

Similar to the Section B.3, we can use the induction method to derive

$$\sqrt[2q]{\sum_{n=1}^{N} \zeta_n d^{2q}(\overline{V}^T, W_n^T)} - D_q(\pi^*)$$

$$\leq \widetilde{\mathcal{O}} \left( \mathrm{poly}(B_1, m, (\min_{n \in [N]} p^{(n)})^{-1})T^{-1/2q} \right) + (*) + \frac{1}{T} \sum_{i=1}^{T} \sqrt[2q]{\sum_{n=1}^{N} \zeta_n d_{B_1}^{2q}(W^{t-1,(n)}, W^*)}$$

$$+ \frac{1}{T} \sum_{t=1}^{T} 37q \cdot 2^q t \sqrt[2q]{\sum_{n=1}^{N} \zeta_n d_{B_1}^{2q}(W^{t,(n)}, W^{t-1,(n)})}.$$

Now note that

$$\sqrt[2q]{\sum_{n=1}^{N} \zeta_n d_{B_1}^{2q}(W^{t-1,(n)}, W^*)} \leq \sqrt[2q]{\sum_{n=1}^{N} \zeta_n \frac{m^{3q} B_1^{2q} \|\hat{\alpha}^{t-1,(n)} - \alpha^{*,(n)}\|_\infty^{2q}}{|p^{(n)}|^{2q}}} \tag{51}$$

$$\leq \frac{\gamma^{-1} \cdot \widetilde{\mathcal{O}}(\text{poly}(m, e^B, \exp(1/\beta), d, \log(1/\delta)))}{\min_{n \in [N]} |p^{(n)}|} \cdot \frac{1}{\sqrt{t}}. \tag{52}$$

Also, by Eq. (35), we have

$$\frac{1}{T} \sum_{t=1}^{T} t \sqrt[2q]{\sum_{n=1}^{N} \zeta_n d_{B_1}^{2q}(W^{t,(n)}, W^{t-1,(n)})} \leq \frac{1}{T} \sum_{t=1}^{T} t \sum_{n=1}^{N} d_{B_1}(W^{t,(n)}, W^{t-1,(n)})$$

$$\leq \gamma^{-1} \text{poly}(\exp(1/\beta), m, N, e^B, d, \log(1/\delta), B_1, (\min_{n \in [N]} p^{(n)})^{-1}) \cdot \widetilde{\mathcal{O}}(1/\sqrt{T}). \tag{53}$$

Hence, combining Eq. (52) and Eq. (53),

$$\sqrt[2q]{\sum_{n=1}^{N} \zeta_n d^{2q}(\overline{V}^T, W_n^T)} - D_q(\pi^*)$$

$$\leq \widetilde{\mathcal{O}}\left(\text{poly}(B_1, m, (\min_{n \in [N]} p^{(n)})^{-1}) T^{-1/2q}\right)$$

$$\qquad + \gamma^{-1} \text{poly}(\exp(1/\beta), m, N, e^B, d, \log(1/\delta), B_1, (\min_{n \in [N]} p^{(n)})^{-1}) \widetilde{\mathcal{O}}(1/\sqrt{T}) + (B)$$

$$\leq \widetilde{\mathcal{O}}\left((\gamma^{-1} \text{poly}(\exp(1/\beta), m, N, e^B, d, \log(1/\delta), B_1, (\min_{n \in [N]} p^{(n)})^{-1}) T^{-1/2q}\right) + (*).$$

Now we derive the proof. First,

$$D_q(\tilde{\pi}^T) - D_q(\pi^*)$$

$$= \sqrt[2q]{\sum_{n=1}^{N} \zeta_n d^{2q}(S(\tilde{\pi}^T), W_n^*)} - \sqrt[2q]{\sum_{n=1}^{N} \zeta_n d^{2q}(S(\tilde{\pi}^T), W_n^T)} + \sqrt[2q]{\sum_{n=1}^{N} \zeta_n d^{2q}(S(\tilde{\pi}^T), W_n^T)} - D_q(\pi^*)$$

$$\leq \sqrt[2q]{\sum_{n=1}^{N} \zeta_n |d(S(\tilde{\pi}^T), W_n^T) - d(S(\tilde{\pi}^T), W_n^*)|^{2q}}$$

$$\quad + \sqrt[2q]{\sum_{n=1}^{N} \zeta_n d^{2q}(W_n^T, \mathbb{E}_{\tilde{\pi}^T}[r^*(x,y)] - \frac{\beta}{T} \sum_{t=1}^{T} \mathbb{D}_{\text{KL}}(\pi^t \| \pi_{\text{ref}}) \cdot \mathbf{1}^m)} - \sqrt[2q]{\sum_{n=1}^{N} \zeta_n d^{2q}(W_n^T, \overline{V}^T)}$$

$$\quad + \sqrt[2q]{\sum_{n=1}^{N} \zeta_n d^{2q}(W_n^T, \overline{V}^T)} - D_q(\pi^*)$$

$$\leq \sum_{n=1}^{N} d(W_n^*, W_n^T)$$

$$\quad + \underbrace{\sqrt[2q]{\sum_{n=1}^{N} \zeta_n \left(d(W_n^T, \mathbb{E}_{\tilde{\pi}^T}[r^*(x,y)] - \frac{\beta}{T} \sum_{t=1}^{T} \mathbb{D}_{\text{KL}}(\pi^t \| \pi_{\text{ref}}) \cdot \mathbf{1}^m) - d(W_n^T, \overline{V}^T)\right)^{2q}}}_{(**)}$$

$$\quad + \widetilde{\mathcal{O}}\left((\gamma^{-1} \text{poly}(\exp(1/\beta), m, e^B, d, \log(1/\delta), \min_{n \in [N]} \frac{1}{p^{(n)}}, B_1) N^{1/2q} T^{-1/2q}\right) + (*).$$

First, for the term $\sum_{n=1}^{N} d(W_n^*, W_n^T)$, we can bound it by

$$\sum_{n=1}^{N} d(W_n^*, W_n^T) \leq \sum_{n=1}^{N} \frac{m^{3/2} B_1}{|p^{(n)}|} \cdot \|\alpha^{*,(n)} - \alpha^{T,(n)}\|_\infty.$$

From the Theorem B.4, we can get

$$\sum_{n=1}^{N} d(W_n^*, W_n^T) \leq \frac{m^{3/2} B_1 N}{\min_{n \in [N]} p^{(n)}} \gamma^{-1} \cdot \widetilde{\mathcal{O}}\left(\text{poly}(m, e^B, \exp(1/\beta), d, \log(1/\delta))\right) \cdot \frac{1}{\sqrt{T}}, \tag{54}$$

Now by Eq. (38), note that

$$\overline{V}^T = \frac{1}{T} \sum_{t=1}^{T} \mathbb{E}_{\pi^t}\left[r^{\theta^t}(x, y)\right] - \frac{\beta}{T} \sum_{t=1}^{T} \mathbb{D}_{\text{KL}}(\pi^t \| \pi_{\text{ref}}) \cdot \mathbf{1}^m$$

we have

$$(**) \leq \sqrt[2q]{\sum_{n=1}^{N} \zeta_n \left(\sum_{i=1}^{m} \left|\frac{1}{T} \sum_{t=1}^{T} \mathbb{E}_{\pi^t}[r_i^*(x, y) - \hat{r}_i^t(x, y)]\right|\right)^{2q}} \leq \left(\sum_{n=1}^{N} \zeta_n\right) \cdot \sum_{i=1}^{m} \left|\frac{1}{T} \sum_{t=1}^{T} \mathbb{E}_{\pi^t}[r_i^*(x, y) - \hat{r}_i^t(x, y)]\right|$$

$$\leq \sum_{i=1}^{m} \mu_i \exp(4/\beta) \kappa \cdot \sum_{t=1}^{T} \sum_{j=1}^{t-1} \mathbb{E}_{y_1, y_2 \sim \pi^j}[\Delta_i^t(x, y)^2] + \frac{d_{\text{cover}}(1/T)}{4\mu_i} + \widetilde{\mathcal{O}}(NBd). \tag{55}$$

Consider the term (B). By Eq. (42), we can get

$$(\text{B}) = \frac{1}{T} \sum_{t=1}^{T} \sum_{i=1}^{m} \eta \|d^t\|_1 (L_i^t(\theta_i^*) - L_i^t(\theta_i^t)) \leq \widetilde{\mathcal{O}}(2m\eta\sqrt{m}d) - \frac{\eta C' \|d^t\|_1}{Tme^B} \sum_{i=1}^{m} \sum_{t=1}^{T} \sum_{j=1}^{t-1} \mathbb{E}_{y_1, y_2 \sim \pi^j}[\Delta_i^t(x, y)^2]$$

$$\leq \widetilde{\mathcal{O}}(2m\eta\sqrt{m}d) - \frac{\eta C' \zeta_n^{1/2q}}{Tme^B N^{\frac{2q-1}{2q}}} \sum_{i=1}^{m} \sum_{t=1}^{T} \sum_{j=1}^{t-1} \mathbb{E}_{y_1, y_2 \sim \pi^j}[\Delta_i^t(x, y)^2]. \tag{56}$$

The last inequality is because, if we choose $n' = \max_{n \in [N]} \zeta_n \|W^{(n)} - \overline{V}^t\|_2^{2q}$, then

$$\frac{\zeta_{n'} \|W^{(n')} - \overline{V}^t\|_2^{2q-1}}{\left(\sum_{n=1}^{N} \zeta_n \|W^{(n)} - \overline{V}^t\|_2^{2q}\right)^{\frac{2q-1}{2q}}} \geq \frac{\zeta_{n'} \|W^{(n')} - \overline{V}^t\|_2^{2q-1}}{N^{\frac{2q-1}{2q}} \cdot \zeta_{n'}^{\frac{2q-1}{2q}} \|W^{(n')} - \overline{V}^t\|_2^{2q-1}} = \frac{\zeta_{n'}^{1/2q}}{N^{\frac{2q-1}{2q}}} \geq \frac{\min_{n \in [N]} \zeta_n^{1/2q}}{N^{\frac{2q-1}{2q}}}.$$

Hence, we have

$$\|d^t\|_1 = \left\|\sum_{n=1}^{N} d_n^t \cdot \frac{\zeta_n \|W^{(n)} - \overline{V}^t\|_2^{2q-1}}{\left(\sum_{n=1}^{N} \zeta_n \|W^{(n)} - \overline{V}^t\|_2^{2q}\right)^{\frac{2q-1}{2q}}}\right\|_1 \geq \|d_{n'}^t\|_1 \cdot \frac{\min_{n \in [N]} \zeta_n^{1/2q}}{N^{\frac{2q-1}{2q}}}.$$

Now we choose $\mu_i = \eta \cdot \frac{C' \min_{n \in [N]} \zeta_n^{1/2q}}{me^B \exp(4/\beta) \kappa N^{\frac{2q-1}{2q}}}$, $\eta = 1/\sqrt{T}$, and use the inequality Eq. (56) for bounding (B), we finally get

$$D_q(\tilde{\pi}^T) - D_q(\pi^*)$$

$$\leq \sum_{n=1}^{N} d(W_n^*, W_n^T) + (\text{A}) + (\text{B}) + \widetilde{\mathcal{O}}\left((\gamma^{-1} \text{poly}(\exp(1/\beta), m, e^B, d, \log(1/\delta), \min_{n \in [N]} \frac{1}{p^{(n)}}, B_1) N^{1/2q} T^{-1/2q}\right)$$

$$\leq \gamma^{-1} \text{poly}(\exp(1/\beta), m, N, e^B, d, \log(1/\delta), \kappa, B_1, (\min_{n \in [N]} p^{(n)})^{-1}, (\min_{n \in [N]} \zeta_n)^{-1/2q}) \cdot \widetilde{\mathcal{O}}(T^{-1/2q}).$$

$$\square$$

## B.6. Estimation of $\alpha$

In this subsection, we give a theoretical upper bound of the estimation error of $\alpha$.

**Theorem B.4** (Estimation of $\alpha$). *Assume that we execute the Algorithm 4 with the Assumption 5.3, then for each $t \in [T]$, with probability at least $1 - \delta$ we have*

$$\|\alpha^* - \alpha^t\|_\infty \leq \frac{1}{t} \sum_{k=1}^{t} \|\alpha^* - \hat{\alpha}^k\|_\infty \leq \gamma^{-1} \cdot \widetilde{\mathcal{O}}\left(\mathrm{poly}(m, e^B, \exp(1/\beta), d, \log(1/\delta))\right) \cdot \frac{1}{\sqrt{t}}. \tag{57}$$

*Proof.* First, for any $k \in [t]$, we estimate $\hat{\alpha}$ with $\tilde{\theta}_1^k, \tilde{\theta}_2^k, \cdots, \tilde{\theta}_m^k$, where $\tilde{\theta}_i^k = \arg\min_\theta L_i^k(\theta)$ only minimizes the log-likelihood loss without optimistic exploration. Define $\delta_i^k(x, y) = \left| r_i^{\tilde{\theta}_i^k}(x, y_1) - r_i^{\tilde{\theta}_i^k}(x, y_2) - (r_i^*(x, y_1) - r_i^*(x, y_2)) \right|$. then, by theorem D.1, with probability $1 - \delta$ we have

$$\sum_{s=1}^{k-1} \mathbb{E}_{x,y \sim \mathcal{D}_s} \left\| \mathrm{Softmax}(\hat{\alpha}_i^k \cdot |r_i^{\tilde{\theta}_i^k}(x, y_1) - r_i^{\tilde{\theta}_i^k}(x, y_2)|) - \mathrm{Softmax}(\alpha_i^* \cdot |r_i^*(x, y_1) - r_i^*(x, y_2)|) \right\|_{\mathrm{TV}}^2$$

$$\leq 2\log(d_{\mathcal{F}}(1/k^2)/\delta) + 1/k,$$

where $\mathcal{F} = \{\mathrm{Softmax}(x_i) \mid 1 \leq i \leq m, x_i \leq 1\}$, and the log of $\varepsilon-$covering number $\log(d_{\mathcal{F}}(1/k^2)) = \widetilde{\mathcal{O}}(m)$.

thus by the Cauchy's inequality, we can get

$$\sqrt{2k \log(d_{\mathcal{F}}(1/k^2)/\delta) + 1}$$

$$\geq \sum_{s=1}^{k-1} \mathbb{E}_{x,y \sim \mathcal{D}_s} \left\| \mathrm{Softmax}(\hat{\alpha}_i^k \cdot |r_i^{\tilde{\theta}_i^k}(x, y_1) - r_i^{\tilde{\theta}_i^k}(x, y_2)|) - \mathrm{Softmax}(\alpha_i^* \cdot |r_i^*(x, y_1) - r_i^*(x, y_2)|) \right\|_{\mathrm{TV}}$$

$$\geq \sum_{s=1}^{k-1} \mathbb{E}_{x,y \sim \mathcal{D}_s} \left\| \mathrm{Softmax}(\hat{\alpha}_i^k \cdot |r_i^*(x, y_1) - r_i^*(x, y_2)|) - \mathrm{Softmax}(\alpha_i^* \cdot |r_i^*(x, y_1) - r_i^*(x, y_2)|) \right\|_{\mathrm{TV}}$$

$$- \sum_{s=1}^{k-1} \mathbb{E}_{x,y \sim \mathcal{D}_s} \left\| \mathrm{Softmax}(\hat{\alpha}_i^k \cdot |r_i^{\tilde{\theta}_i^k}(x, y_1) - r_i^{\tilde{\theta}_i^k}(x, y_2)|) - \mathrm{Softmax}(\hat{\alpha}_i^k \cdot |r_i^*(x, y_1) - r_i^*(x, y_2)|) \right\|_{\mathrm{TV}}. \tag{58}$$

Now we bound the difference of $\alpha$ based on the difference of the softmax distribution.

Fixed $k$, since the upper bound of $0 \leq r_i^{\tilde{\theta}_i^k}(x, y) \leq B$ and $0 \leq r^*(x, y) \leq B$, define $X_i = |r_i^{\tilde{\theta}_i^k}(x, y_1) - r_i^{\tilde{\theta}_i^k}(x, y_2)| \leq B$ and $X_i^* = |r_i^*(x, y_1) - r_i^*(x, y_2)| \leq B$, then

$$\left\| \mathrm{Softmax}(\hat{\alpha}_i^k \cdot |r_i^{\tilde{\theta}_i^k}(x, y_1) - r_i^{\tilde{\theta}_i^k}(x, y_2)|) - \mathrm{Softmax}(\hat{\alpha}_i^k \cdot |r_i^*(x, y_1) - r_i^*(x, y_2)|) \right\|_{\mathrm{TV}}$$

$$= \sum_i \left| \frac{e^{X_i \cdot \hat{\alpha}_i^k}}{\sum_j e^{X_i \cdot \hat{\alpha}_j^k}} - \frac{e^{X_j^* \cdot \hat{\alpha}_i^k}}{\sum_j e^{X_j^* \cdot \hat{\alpha}_j^k}} \right|$$

$$= \sum_i \left| \frac{\sum_{j \neq i} e^{X_j^* \cdot \hat{\alpha}_j^k + X_i \hat{\alpha}_i^k} - e^{X_j \cdot \hat{\alpha}_j^k + X_i^* \hat{\alpha}_i^k}}{(\sum_j e^{X_j \cdot \hat{\alpha}_j^k})(\sum_j e^{X_j^* \cdot \hat{\alpha}_j^k})} \right|$$

$$\leq \sum_i \left| \frac{\sum_{j \neq i} e^{X_j^* \hat{\alpha}_j^k + X_i \hat{\alpha}_i^k}(e^{\delta_j^k \hat{\alpha}_j^k + \delta_i^k \hat{\alpha}_i^k} - 1)}{m^2} \right|,$$

where the last inequality uses the fact that $\sum_j e^{X_j \cdot \hat{\alpha}_i^k} \geq m$ and $\sum_j e^{X_j^* \hat{\alpha}_j^k} \geq m$. Now since $e^{X_j^* \hat{\alpha}_j^k + X_i \hat{\alpha}_i^k} \leq e^{B(\hat{\alpha}_i^k + \hat{\alpha}_j^k)} \leq e^B$, and $e^a - 1 \leq e^B \cdot a$ for every $0 \leq a \leq B$, we can have

$$\leq \sum_i \left| \frac{\sum_{j \neq i} e^{2B}(\delta_j^k \hat{\alpha}_j^k + \delta_i^k \hat{\alpha}_i^k)}{m^2} \right|$$

$$\leq \frac{e^{2B}}{m^2} \sum_i \sum_{j \neq i} (\delta_j^k \hat{\alpha}_j^k + \delta_i^k \hat{\alpha}_i^k)$$

$$\leq \frac{e^{2B}}{m} \sum_i \delta_i^k \hat{\alpha}_i^k.$$

Now choose the index $l = \arg\max_{i \in [m]} X_i^* \circ |\alpha_i^* - \hat{\alpha}_i^k|$, and WLOG, assume $\hat{\alpha}_l^k = \alpha_l^* + \varepsilon$, then we can bound

$$\begin{aligned}
&\left\| \mathrm{Softmax}(\hat{\alpha}_l^k \cdot |r_l^*(x, y_1) - r_l^*(x, y_2)|) - \mathrm{Softmax}(\alpha_l^* \cdot |r_l^*(x, y_1) - r_l^*(x, y_2)|) \right\|_{\mathrm{TV}} \\
&\geq \left| \frac{e^{X_l^* \hat{\alpha}_l^k}}{\sum_j e^{X_j^* \hat{\alpha}_j^k}} - \frac{e^{X_l^* \cdot \alpha_l^*}}{\sum_j e^{X_j^* \alpha_j^*}} \right| \\
&= \left| \frac{e^{X_l^*(\alpha_l^* + \varepsilon)}}{e^{X_l^*(\alpha_l^* + \varepsilon)} + \sum_{j \neq l} e^{X_j^* \hat{\alpha}_j^k}} - \frac{e^{X_l^* \cdot \alpha_l^*}}{e^{X_l^* \alpha_l^*} + \sum_{j \neq l} e^{X_j^* \alpha_j^*}} \right| \\
&= \left| \frac{\sum_{j \neq l} e^{X_j^* \alpha_j^* + X_l^*(\alpha_l^* + \varepsilon)} - e^{X_j^* \hat{\alpha}_j^k + X_l^* \alpha_l^*}}{(\sum_j e^{X_j^* \hat{\alpha}_j^k})(\sum_j e^{X_j^* \alpha_j^*})} \right|.
\end{aligned}$$

Now by the selection of the $l$, we can have

$$X_j^* \alpha_j^* + X_l^*(\alpha_l^* + \varepsilon) \geq X_j^* \hat{\alpha}_j^k + X_l^* \alpha_l^*,$$

hence

$$e^{X_j^* \alpha_j^* + X_l^*(\alpha_l^* + \varepsilon)} \geq e^{X_j^* \hat{\alpha}_j^k + X_l^* \alpha_l^*}.$$

Also, since $\sum_i \alpha_i^* = \sum_i \hat{\alpha}_i^k = 1$, and the fact that $\hat{\alpha}_l^k = \alpha_l^* + \varepsilon$, we can further derive

$$\sum_{j \neq l} \alpha_j^* = \sum_{j \neq l} \hat{\alpha}_j^k + \varepsilon.$$

then at least one $j' \neq l$ such that $\alpha_{j'}^* \geq \hat{\alpha}_{j'}^k + \varepsilon/m$. then

$$\begin{aligned}
e^{X_{j'}^* \alpha_{j'}^* + X_l^*(\alpha_l^* + \varepsilon)} - e^{\hat{\alpha}_{j'}^k X_{j'}^* + X_l^* \alpha_l^*} &\geq e^{X_{j'}^* \hat{\alpha}_{j'}^k + X_l^*(\alpha_l^* + \varepsilon)} - e^{\hat{\alpha}_{j'}^k X_{j'}^* + X_l^* \alpha_l^*} \\
&\geq e^{X_l^*(\alpha_l^* + \varepsilon)} - e^{X_l^* \alpha_l^*} \\
&\geq e^{\alpha_l^* X_l^*}(e^{\varepsilon X_l^*} - 1) \\
&\geq e^{\alpha_l^* X_l^*} \cdot \varepsilon X_l^*.
\end{aligned}$$

Thus,

$$\begin{aligned}
&\left\| \mathrm{Softmax}(\hat{\alpha}_l^k \cdot |r_l^*(x, y_1) - r_l^*(x, y_2)|) - \mathrm{Softmax}(\alpha_l^* \cdot |r_l^*(x, y_1) - r_l^*(x, y_2)|) \right\|_{\mathrm{TV}} \\
&\geq \frac{e^{\alpha_l^* X_l^*}}{(\sum_j e^{X_j^* \hat{\alpha}_j^k})(\sum_j e^{X_j^* \alpha_j^*})} \cdot \varepsilon X_l^* \\[2mm]
&\geq \frac{1}{(me^B)^2} \cdot |\hat{\alpha}_l^k - \alpha_l^*| X_l^*.
\end{aligned}$$

Now define $X^* = (X_1^*, X_2^*, \cdots, X_m^*)^\top \in \mathbb{R}^m$ and $|\alpha^* - \hat{\alpha}^k| = (|\alpha_1^* - \hat{\alpha}_1^k|, \cdots, |\alpha_m^* - \hat{\alpha}_m^k|)^\top \in \mathbb{R}^m$. We can get

$$\|X^* \circ |\alpha^* - \hat{\alpha}^k|\|_\infty \leq m^2 e^{2B} \left\| \mathrm{Softmax}(\hat{\alpha}_i^k \cdot |r_i^*(x, y_1) - r_i^*(x, y_2)|) - \mathrm{Softmax}(\alpha_i^* \cdot |r_i^*(x, y_1) - r_i^*(x, y_2)|) \right\|_{\mathrm{TV}},$$

where $X \circ Y$ denotes the Hadamard product. then take the expectation we can get

$$\begin{aligned}
&\mathbb{E}_{x, y \sim \mathcal{D}_s} \|X^* \circ |\alpha^* - \hat{\alpha}^k|\|_\infty \\
&\quad \leq m^2 e^{2B} \mathbb{E}_{x, y \sim \mathcal{D}_s} \left\| \mathrm{Softmax}(\hat{\alpha}_i^k \cdot |r_i^*(x, y_1) - r_i^*(x, y_2)|) - \mathrm{Softmax}(\alpha_i^* \cdot |r_i^*(x, y_1) - r_i^*(x, y_2)|) \right\|_{\mathrm{TV}}.
\end{aligned}$$

Hence, by Eq.(58), we have

$$\sqrt{2k\log(d_{\mathcal{F}}(1/k^2)/\delta)+1}$$

$$\geq \sum_{s=1}^{k-1}\mathbb{E}_{x,y\sim\mathcal{D}_s}\left\|\text{Softmax}(\hat{\alpha}_i^k\cdot|r_i^*(x,y_1)-r_i^*(x,y_2)|)-\text{Softmax}(\alpha_i^*\cdot|r_i^*(x,y_1)-r_i^*(x,y_2)|)\right\|_{\text{TV}}$$

$$-\sum_{s=1}^{k-1}\mathbb{E}_{x,y\sim\mathcal{D}_s}\left\|\text{Softmax}(\hat{\alpha}_i^k\cdot|r_i^{\tilde{\theta}_i^k}(x,y_1)-r_i^{\tilde{\theta}_i^k}(x,y_2)|)-\text{Softmax}(\hat{\alpha}_i^k\cdot|r_i^*(x,y_1)-r_i^*(x,y_2)|)\right\|_{\text{TV}}$$

$$\geq \sum_{s=1}^{k-1}\mathbb{E}_{x,y\sim\mathcal{D}_s}\frac{1}{m^2e^{2B}}\|X^*(x,y)|\alpha^*-\hat{\alpha}^k|\|_{\infty}$$

$$-\sum_{s=1}^{k-1}\mathbb{E}_{x,y\sim\mathcal{D}_s}\left\|\text{Softmax}(\hat{\alpha}_i^k\cdot|r_i^{\tilde{\theta}_i^k}(x,y_1)-r_i^{\tilde{\theta}_i^k}(x,y_2)|)-\text{Softmax}(\hat{\alpha}_i^k\cdot|r_i^*(x,y_1)-r_i^*(x,y_2)|)\right\|_{\text{TV}}$$

$$\geq \sum_{s=1}^{k-1}\mathbb{E}_{x,y\sim\mathcal{D}_s}\frac{1}{m^2e^{2B}}\|X^*(x,y)\cdot|\alpha^*-\hat{\alpha}^k|\|_{\infty}-\frac{e^{2B}}{m}\sum_{s=1}^{k-1}\mathbb{E}_{x,y\sim\mathcal{D}_s}[\delta_i^k(x,y)\hat{\alpha}_i^k].$$

Hence we finally get

$$\sum_{s=1}^{k-1}\mathbb{E}_{x,y\sim\mathcal{D}_s}\|X^*(x,y)\circ|\alpha^*-\hat{\alpha}^k|\|_{\infty}\leq m^2e^{2B}\left(\sqrt{2k\log(d_{\mathcal{F}}(1/k^2)/\delta)+1}+\frac{e^{2B}}{m}\sum_{s=1}^{k-1}\sum_{i=1}^{m}\delta_i^k(x^s,y^s)\hat{\alpha}_i^k\right)$$

$$=\text{poly}(m,\exp(B))\cdot\widetilde{\mathcal{O}}\left(\sqrt{km\log(1/\delta)}+\sum_{s=1}^{k-1}\sum_{i=1}^{m}\mathbb{E}_{y_1,y_2\sim\pi^s}[\delta_i^k(x,y)]\hat{\alpha}_i^k\right).$$

$$(59)$$

the last inequality holds by Azuma-Hoeffding's inequality with probability at least $1-\delta$. Now by Lemma D.2, we can get $\sup_{s,x,y}\frac{\pi^*(y|x)}{\pi^s(y|x)}\leq\exp(4/\beta)$ and $\sup_{x,y}\frac{\pi_{\text{ref}}(y|x)}{\pi^s(y|x)}\leq\exp(4/\beta)$, we can get

$$\gamma(k-1)\|\alpha^*-\hat{\alpha}^k\|_{\infty}\leq (k-1)\mathbb{E}_{y_1\sim\pi^*,y_2\sim\pi_{\text{ref}}}\|X^*(x,y)\circ|\alpha^*-\hat{\alpha}^k|\|_{\infty}$$

$$\leq \exp(8/\beta)\sum_{s=1}^{k-1}\mathbb{E}_{y_1,y_2\sim\pi^s}\|X^*(x,y)\circ|\alpha^*-\hat{\alpha}^s|\|_{\infty}. \quad (60)$$

The first inequality uses the Assumption 5.3 that $\mathbb{E}_{y_1\sim\pi^*,y_2\sim\pi_{\text{ref}}}[X_i^*(x,y)]\geq\gamma$. Now combining Eq.(59) and Eq.(60), we can further get

$$\gamma(k-1)\|\alpha^*-\hat{\alpha}^k\|_{\infty}\leq\exp(8/\beta)\cdot\text{poly}(m,\exp(B))\cdot\widetilde{\mathcal{O}}\left(\sqrt{km\log(1/\delta)}+\sum_{s=1}^{k-1}\sum_{i=1}^{m}\mathbb{E}_{y_1,y_2\sim\pi^s}[\delta_i^k(x,y)]\hat{\alpha}_i^k\right)$$

$$\leq\exp(8/\beta)\cdot\text{poly}(m,\exp(B))\cdot\widetilde{\mathcal{O}}\left(\sqrt{km\log(1/\delta)}+\sum_{s=1}^{k-1}\sum_{i=1}^{m}\mathbb{E}_{y_1,y_2\sim\pi^s}[\delta_i^k(x,y)]\right). \quad (61)$$

Now we further derive the final result. Frist, by $\alpha^t=\frac{1}{t}\sum_{k=1}^{t}\hat{\alpha}^k$, we can get

$$\|\alpha^*-\alpha^t\|_{\infty}\leq\frac{1}{t}\sum_{k=1}^{t}\|\alpha^*-\hat{\alpha}^k\|_{\infty}$$

$$\leq\frac{\gamma^{-1}\exp(8/\beta)\cdot\text{poly}(m,\exp(B))}{t}\cdot\sum_{k=1}^{t}\widetilde{\mathcal{O}}\left(\frac{\sqrt{m\log(1/\delta)}}{\sqrt{k}}+\frac{1}{k}\sum_{s=1}^{k-1}\sum_{i=1}^{m}\mathbb{E}_{y_1,y_2\sim\pi^s}[\delta_i^k(x,y)]\right). \quad (62)$$

Now we derive the final result. First, we can get

$$\delta_i^k(x^s,y^s)=\left|\langle\tilde{\theta}_i^k-\theta_i^*,\phi_i(x^s,y_1^s)-\phi_i(x^s,y_2^s)\rangle\right|$$

$$\leq \|\tilde{\theta}_i^k - \theta_i^*\|_{\Sigma_{\mathcal{D}_i^{k-1}}} \cdot \|\phi_i(x^s, y_1^s) - \phi_i(x^s, y_2^s)\|_{(\Sigma_{\mathcal{D}_i^{k-1}})^{-1}},$$

where $\mathcal{D}_i^{k-1} = \{s \in [k-1] \mid I_s = i\}$ and $\Sigma_{\mathcal{D}_i^{k-1}} = \sum_{s \in \mathcal{D}_i^{k-1}} \phi_i(x^s, y^s) \phi_i(x^s, y^s)^\top$ is the covariance matrix. then by Lemma 3.1 in (Zhu et al., 2023), we can get $\|\tilde{\theta}_i^k - \theta_i^*\|_{\Sigma_{\mathcal{D}_i^{k-1}}} \leq C(d, B, \delta) = \text{poly}(d, B, \log(1/\delta))$ for some constant $C(d, B, \delta)$, and then we can get

$$\delta_i^k(x^s, y^s) \leq C(d, B, \delta) \cdot \|\phi_i(x^s, y_1^s) - \phi_i(x^s, y_2^s)\|_{(\Sigma_{\mathcal{D}_i^{k-1}})^{-1}}.$$

Now apply the same technique in Eq.(41), we can get

$$\sum_{i=1}^m \sum_{k=1}^t \sum_{s=1}^{k-1} \mathbb{E}_{y_1, y_2 \sim \pi^s} \frac{1}{k} [\delta_i^k(x, y) \hat{\alpha}_i^k] \leq m e^B \sum_{k=1}^t \sum_{i=1}^m \sum_{s \in \mathcal{D}_i^{k-1}} \mathbb{E}_{y_1, y_2 \sim \pi^s} \frac{1}{k} [\delta_i^k(x, y) \hat{\alpha}_i^k]$$

$$= m e^B \sum_{i=1}^m \sum_{s=1}^t \mathbb{E}_{y_1, y_2 \sim \pi^s} \sum_{k > s} \left[ \frac{1}{k} \delta_{I_s}^k(x, y) \hat{\alpha}_{I_s}^k \right]$$

$$\leq m e^B \sum_{s=1}^t \mathbb{E}_{y_1, y_2 \sim \pi^s} \sum_{k > s} \left[ \frac{1}{k} \delta_{I_s}^k(x, y) \right].$$

The second line is because that, the summation is over

$$\{(k, i, s) \mid k \in [t], i \in [m], s \in \mathcal{D}_i^{k-1}\} = \{(k, i, s) \mid k \in [t], i \in [m], s \leq k-1, I^s = i\}$$
$$= \{(k, i, s) \mid s \in [t], k > s, i = I^s\}.$$

the last inequality uses the fact that $\hat{\alpha}_{I_s}^k \leq 1$. then we can use the Azuma-Hoeffding's inequality to further get

$$\sum_{i=1}^m \sum_{k=1}^t \sum_{s=1}^{k-1} \mathbb{E}_{y_1, y_2 \sim \pi^s} \frac{1}{k} [\delta_i^k(x, y) \hat{\alpha}_i^k] \leq m e^B \sum_{k=1}^t \sum_{k \geq s} \left[ \frac{1}{k} \delta_{I_s}^k(x^s, y^s) \right] + \mathcal{O}(\sqrt{t} \log(t/\delta))$$

$$\leq m e^B \sum_{k=1}^t \sum_{k \geq s} \left[ \frac{1}{k} C(d, B, \delta) \cdot \|\phi_{I_s}(x^s, y_1^s) - \phi_{I_s}(x^s, y_2^s)\|_{\Sigma_{\mathcal{D}_{I_s}^{k-1}}} \right]$$
$$+ \mathcal{O}(\sqrt{t} \log(t/\delta)) \tag{63}$$

with probability at least $1 - \delta$. Now to present the proof in a simple way, we simplify $\Sigma_{\mathcal{D}_{I_s}^{k-1}}$ as $\Sigma^{k-1, (I_s)}$. We will have

$$m e^B \sum_{s=1}^t \sum_{k > s} \left[ \frac{1}{k} \cdot C(d, B, \delta) \cdot \|\phi_{I_s}(x^s, y_1^s) - \phi_{I_s}(x^s, y_2^s)\|_{(\Sigma^{k-1, (s)})^{-1}} \right]$$

$$\leq m e^B \sum_{s=1}^t \sum_{k > s} \frac{1}{k} \cdot C(d, B, \delta) \cdot \|\phi_{I_s}(x^s, y_1^s) - \phi_{I_s}(x^s, y_2^s)\|_{(\Sigma^{s, (s)})^{-1}}$$

$$\leq m e^B \sum_{s=1}^t C(d, B, \delta) \|\phi_{I_s}(x^s, y_1^s) - \phi_{I_s}(x^s, y_2^s)\|_{(\Sigma^{s, (s)})^{-1}} \sum_{k > s}^t \frac{1}{k}$$

$$\leq \frac{\log t}{\kappa_1} \cdot \sum_{s=1}^t C(d, B, \delta) \|\phi_{I_s}(x^s, y_1^s) - \phi_{I_s}(x^s, y_2^s)\|_{(\Sigma^{s, (s)})^{-1}}. \tag{64}$$

Now, we can decompose $\{1, 2, \cdots, t\}$ into $m$ different set $\mathcal{D}_i = \{s \in [t] : I_s = i\}$. then, we fixed $i$ and denote $M_s = \|\phi_i(x^s, y_1^s) - \phi_i(x^s, y_2^s)\|_{(\Sigma_{\mathcal{D}_{I_s}^s})^{-1}}^2$ with $\|M_s\| \leq B^2$, by Cauchy's inequality,

$$\sum_{s \in \mathcal{D}_i} \|\phi_{I_s}(x^s, y_1^s) - \phi_{I_s}(x^s, y_2^s)\|_{(\Sigma^{s, (s)})^{-1}}$$

$$\leq \sqrt{t}\sqrt{\sum_{s\in\mathcal{D}_i} M_s}$$

$$\leq \sqrt{t}\sqrt{\sum_{s\in\mathcal{D}_i} M_s\mathbb{I}\{M_s \leq 1\}} + \sqrt{t}\sqrt{\sum_{s\in\mathcal{D}_i} M_s\mathbb{I}\{M_s > 1\}}$$

$$\leq \sqrt{t}\cdot\left(\sqrt{\sum_{s\in\mathcal{D}_i} \min\{1, M_s\}} + \sqrt{B^2\sum_{s\in\mathcal{D}_i} \mathbb{I}\{M_s > 1\}}\right)$$

$$\leq \widetilde{\mathcal{O}}(Bd\sqrt{t}).$$

Then, by summing over $i \in [m]$, we can get

$$\frac{\log t}{\kappa_1} \cdot \sum_{s=1}^{t} C(d, B, \delta)\|\phi_{I_s}(x^s, y_1^s) - \phi_{I_s}(x^s, y_2^s)\|_{(\Sigma_{\mathcal{D}_i}^j)^{-1}}$$

$$\leq \frac{\log t}{\kappa_1} \cdot C(d, B, \delta) \cdot m \cdot \widetilde{\mathcal{O}}(Bd\sqrt{t})$$

$$= \widetilde{\mathcal{O}}(m^2 e^B \cdot Bd \cdot C(d, B, \delta)\sqrt{t}). \tag{65}$$

Now combining Eq.(62), Eq.(63), Eq.(64) and Eq.(65), we can finally get

$$\|\alpha^* - \alpha^t\|_\infty \leq \frac{1}{t}\sum_{k=1}^{t}\|\alpha^* - \hat{\alpha}^k\|_\infty \leq \gamma^{-1}\exp(8/\beta)\cdot\widetilde{\mathcal{O}}\left(\text{poly}(m, e^B, d, \log(1/\delta))\right)\cdot\frac{1}{\sqrt{t}}$$

$$= \gamma^{-1}\cdot\widetilde{\mathcal{O}}\left(\text{poly}(m, e^B, \exp(1/\beta), d, \log(1/\delta))\right)\cdot\frac{1}{\sqrt{t}}$$

with probability at least $1 - 3\delta$. By substituting $\delta/3$ with $\delta$, we complete the proof. $\qquad\square$

## C. Error of Estimating the Target Set

First we provide a lemma to show that the projection on $W^*$ is also bounded.

**Lemma C.1.** *Fixed the requirement $p^{(n)}, c^{(n)}$ for all $k \in [K]$. For any importance weight $\{\alpha^{(n)}\}_{k\in[K]}$ such that $\alpha^{(n)} \succeq 0$ and $\|\alpha^{(n)}\|_1 = 1$ for all $k \in [K]$, for $B_1 = 2\sqrt{m}(B + \max_n c^{(n)})$, we have*

$$\|\Pi_{W^*}(x)\|_\infty \leq B_1, \quad W^* = \bigcap_{i=1}^{K} W_{p^{(n)}, c^{(n)}}^{\alpha^{(n)}}$$

*holds for all $\|x\|_\infty \leq B$.*

*Proof.* Suppose we choose any $y \in W^*$, then by the definition of projection, we can get

$$\|\Pi_{W^*}(x)\|_\infty - \sqrt{m}B \leq \|x - \Pi_{W^*}(x)\|_\infty \leq \|x - \Pi_{W^*}(x)\|_2 \leq \|x - y\| \leq \sqrt{m}B + \|y\|,$$

which induces

$$\|\Pi_{W^*}(x)\| \leq 2\sqrt{m}B + \|y\|.$$

Now consider $y = (z, \cdots, z)^\top \in \mathbb{R}^m$, when $z = \max_n c^{(n)}$, for any $\alpha^{(n)}$

$$\left(\sum_{i=1}^{m} \alpha_i^{(n)} y_i^{|p^{(n)}|}\right)^{1/p^{(n)}} = z \cdot \left(\sum_{i=1}^{m} \alpha_i^{(n)}\right)^{1/p^{(n)}} = z \geq c^{(n)}.$$

That means $y \in W_{p^{(n)}, c^{(n)}}^{\alpha^{(n)}}$ and then $y \in W^*$ for any $k \in [K]$. Hence we have

$$\|\Pi_{W^*}(x)\| \leq 2B + \|y\| \leq 2\sqrt{m}(B + \max_n c^{(n)}).$$

We complete the proof of lemma. $\qquad\square$

Now we consider the estimation of the $W^*$. First, we consider the estimation error of $W^\alpha$ when we have an estimation error of $\alpha$. The following lemma tells us the estimation error of parameterized target set.

**Lemma C.2** (Estimation error of parameterized target set). *Suppose we have two different $\alpha, \alpha'$, the distance between $W_{p,c}^\alpha$ and $W_{p,c}^{\alpha'}$ can be bounded by*

$$d_B(W_{p,c}^\alpha, W_{p,c}^{\alpha'}) \leq \frac{m^{3/2}B\|\alpha - \alpha'\|_\infty}{|p|},$$

*where*

$$d_B(S, S') = \max\left\{\max_{x \in S, \|x\|_\infty \leq B} d(x, \Pi_{S'}(x)), \max_{x \in S', \|x\|_\infty \leq B} d(x, \Pi_S(x))\right\}$$

*represents the distance of two sets $S$ and $S'$ restricted to some bounded set.*

*Proof.* Suppose $p \in [0, 1]$ and $x \in W_{p,c}^\alpha$ with $\|x\|_\infty \leq B$, then we have

$$\sum_{i=1}^m \alpha_i x_i^p \geq c^p.$$

First, if $\sum_{i=1}^m \alpha_i' x_i^p \geq c^p$, then $x \in W_{p,c}^{\alpha'}$ and the distance $d(x, \Pi_{W_{p,c}^{\alpha'}}(x)) = 0$. Now we consider the auxillary vector $y \in \mathbb{R}^m$ where $y_i = x_i^p$ for $i \in [m]$. Then $\sum_{i=1}^m \alpha_i y_i \geq c^p$. By the formula of the distance between one point to a line, the distance between $y$ and $W_{p,c}^{\alpha'} = \{y : \sum_{i=1}^m \alpha_i y_i \geq c^p, y_i \succeq 0\}$ can have the following upper bound:

$$d(y, \Pi_{W_{p,c}^{\alpha'}}(y)) = \frac{\max\{c^p - \sum_{i=1}^m \alpha_i' y_i, 0\}}{\sqrt{\sum_{i=1}^m (\alpha_i')^2}} \leq \frac{\max\{\sum_{i=1}^m (\alpha_i - \alpha_i') y_i, 0\}}{\sqrt{\sum_{i=1}^m (\alpha_i')^2}} \leq \frac{\|\alpha - \alpha'\|_\infty m B^p}{\sqrt{\sum_{i=1}^m (\alpha_i')^2}}.$$

Now consider $p < 0$ we have $\sum_{i=1}^m \alpha_i x_i^p \leq c^p$. If $\sum_{i=1}^m \alpha_i' y_i \leq c^p$, then $x \in W_{p,c}^{\alpha'}$ and the distance $d(x, \Pi_{W_{p,c}^{\alpha'}}(x)) = 0$. Otherwise, note that we can rewrite $W_{p,c}^\alpha = \{y : \sum_{i=1}^m \alpha_i y_i \leq c^p, y \succeq 0\}$. We have

$$d(y, \Pi_{W_{p,c}^{\alpha'}}(y)) = \frac{\sum_{i=1}^m \alpha_i' y_i - c^p}{\sqrt{\sum_{i=1}^m (\alpha_i')^2}} \leq \frac{\|\alpha - \alpha'\|_\infty \sum_{i=1}^m y_i}{\sqrt{\sum_{i=1}^m (\alpha_i')^2}} \leq \frac{\|\alpha - \alpha'\|_\infty \cdot m B^p}{\sqrt{\sum_{i=1}^m (\alpha_i')^2}}.$$

So in both cases, we can find

$$d(y, \Pi_{W_{p,c}^{\alpha'}}(y)) \leq \frac{\|\alpha - \alpha'\|_\infty \cdot m B^p}{\sqrt{\sum_{i=1}^m (\alpha_i')^2}} \leq \frac{\|\alpha - \alpha'\|_\infty \cdot m B^p}{1/\sqrt{m}} = m^{3/2} B^p \cdot \|\alpha - \alpha'\|_\infty.$$

Now since by Langarian mean value theorem we have $|x^p - y^p| \geq |pB^{p-1}||x - y|$, the distance between $x$ can be bounded by

$$d(x, \Pi_{W_{p,c}^{\alpha'}}(x)) \leq \frac{1}{|pB^{p-1}|} d(y, \Pi_{W_{p,c}^{\alpha'}}(y)) \leq \frac{m^{3/2}B^p \cdot \|\alpha - \alpha'\|_\infty}{|p|B^{p-1}} = \frac{m^{3/2}B\|\alpha - \alpha'\|_\infty}{|p|}.$$

$\square$

The second lemma shows that the distance between the projection of one point on different convex set.

**Lemma C.3** (Distance of Projections). *Fixed a point $x$ with $\|x\|_\infty \leq B$. Suppose we have two convex sets $A_1, A_2$, then the distance of two projections can be bounded by*

$$\|\Pi_{A_1}(x) - \Pi_{A_2}(x)\|_2^2 \leq 4d(x, A_1)d_{B_1}(A_1, A_2) + 2d_{B_1}(A_1, A_2)^2.$$

*Proof.* WLOG, we can assume $d(x, A_1) \leq d(x, A_2)$. First, we consider $\Pi_{A_2}(\Pi_{A_1}(x)) \in A_2$ and $d(\Pi_{A_2}(\Pi_{A_1}(x)), \Pi_{A_1}(x)) \leq d_{B_1}(A_1, A_2)$, where $B_1$ is from the bounded assumption of the target set. Now we only need to consider $d(\Pi_{A_2}(\Pi_{A_1}(x)), \Pi_{A_2}(x))$. Since $A_2$ is a convex set and $\Pi_{A_2}(\Pi_{A_1}(x)) \in A_2$, we can have

$$\langle x - \Pi_{A_2}(x), \Pi_{A_2}(x) - \Pi_{A_2}(\Pi_{A_1}(x)) \rangle \geq 0,$$

then it is easy to get

$$d(\Pi_{A_2}(\Pi_{A_1}(x)), x)^2 \geq d(x, A_2)^2 + d(\Pi_{A_2}(\Pi_{A_1}(x)), \Pi_{A_2}(x))^2.$$

Also, by the triangle inequality, we can derive

$$d(\Pi_{A_2}(\Pi_{A_1}(x)), x) \leq d(x, A_1) + d(\Pi_{A_1}(x), \Pi_{A_2}(\Pi_{A_1}(x))) \leq d(x, A_1) + d_{B_1}(A_1, A_2).$$

By combining these two inequality we can get

$$d(\Pi_{A_2}(\Pi_{A_1}(x)), \Pi_{A_2}(x))^2 \leq 2d(x, A_1)d_{B_1}(A_1, A_2) + d_{B_1}(A_1, A_2)^2.$$

Hence we can finally get

$$\|\Pi_{A_1}(x) - \Pi_{A_2}(x)\|_2^2 \leq 2d(\Pi_{A_2}(\Pi_{A_1}(x)), \Pi_{A_2}(x))^2 + 2d(\Pi_{A_2}(\Pi_{A_1}(x)), \Pi_{A_1}(x))^2$$
$$\leq 4d(x, A_1)d_{B_1}(A_1, A_2) + 2d_{B_1}(A_1, A_2)^2.$$

$\square$

Now we derive the difference between the direction.

**Lemma C.4.** *If the angle between the direction $\frac{\Pi_{A_1}(x) - x}{d(x, A_1)}$ and $\frac{\Pi_{A_2}(x) - x}{d(x, A_2)}$ is less than $\pi/2$, then the difference between them can be bounded by*

$$\frac{\Pi_{A_1}(x) - x}{d(x, A_1)} - \frac{\Pi_{A_2}(x) - x}{d(x, A_2)} \leq \frac{4\sqrt{d(x, A_1)d_{B_1}(A_1, A_2)} + 2d_{B_1}(A_1, A_2)}{\max\{d(x, A_1), d(x, A_2)\}}.$$

*Proof.* Denote the angle as $\Delta$ Consider the triangle $(x, \Pi_{A_1}(x), \Pi_{A_2}(x))$. By the law of sines, we can get

$$\sin \Delta \leq \frac{d(\Pi_{A_1}(x), d(\Pi_{A_2}(x)))}{\max\{d(x, A_1), d(x, A_2)\}}.$$

By Lemma C.3, we can get

$$\sin \Delta \leq \frac{2\sqrt{d(x, A_1)d_{B_1}(A_1, A_2)} + \sqrt{2}d_{B_1}(A_1, A_2)}{\max\{d(x, A_1), d(x, A_2)\}}.$$

Now since $\Delta \leq \pi/2$ and the direction can be bounded by

$$\frac{\Pi_{A_1}(x) - x}{d(x, A_1)} - \frac{\Pi_{A_2}(x) - x}{d(x, A_2)} \leq \frac{\sin \Delta}{\sin(\frac{\pi - \Delta}{2})} \leq \sqrt{2}\sin \Delta \leq \frac{4\sqrt{d(x, A_1)d_{B_1}(A_1, A_2)} + 2d_{B_1}(A_1, A_2)}{\max\{d(x, A_1), d(x, A_2)\}}.$$

$\square$

## D. Auxiliary Lemmas

**Lemma D.1** (MLE Lemma). *We are given a dataset $D := \{(x_i, y_i)\}$, where $x_i \sim \mathcal{D}_i = \mathcal{D}_i(x_{1:i-1}, y_{1:i-1})$ and $y_i \sim p(\cdot \mid x_i) = f^*(x_i, \cdot)$. Now if we calculate the MLE by*

$$\hat{f} = \arg\max_{f \in \mathcal{F}} \sum_{i=1}^n \log f(x_i, y_i),$$

*then fixed $\delta \in (0, 1)$, assume $|\mathcal{F}| < \infty$ and $f^* \in \mathcal{F}$, then with probability at least $1 - \delta$, we have*

$$\sum_{i=1}^n \mathbb{E}_{x \in \mathcal{D}_i} \left\| \hat{f}(x, \cdot) - f^*(x, \cdot) \right\|_{TV}^2 \leq 2\log(|\mathcal{F}|/\delta).$$

**Lemma D.2.** *For any $\pi, \pi' \in \{\pi^1, \cdots, \pi^t, \pi^*, \pi_{\mathrm{ref}}\}$, we can have*

$$\sup_{x,y} \frac{\pi(y \mid x)}{\pi'(y \mid x)} \leq \exp(4/\beta).$$

*Proof.* First, note that $\pi$ and $\pi'$ are both optimal policy with respect to some reward $\hat{r}$, then $\pi$ can be rewritten as

$$\pi(y \mid x) \propto \pi_{\mathrm{ref}}(y \mid x) \exp(\langle \hat{\alpha}, \hat{r} \rangle / \beta).$$

Thus by the Appendix A.2 in (Cen et al., 2022), then for any $y$ and $x$, we have

$$|\log \pi(y \mid x) - \log \pi_{\mathrm{ref}}(y \mid x)| \leq 2B/\beta.$$

Then

$$\sup_{x,y} \frac{\pi(y \mid x)}{\pi_{\mathrm{ref}}(y \mid x)} \leq \exp(2B/\beta), \quad \sup_{x,y} \frac{\pi_{\mathrm{ref}}(y \mid x)}{\pi(y \mid x)} \leq \exp(2B/\beta).$$

Now from the two inequalities following

$$\sup_{x,y} \frac{\pi(y \mid x)}{\pi_{\mathrm{ref}}(y \mid x)} \leq \exp(2B/\beta),$$

$$\sup_{x,y} \frac{\pi_{\mathrm{ref}}(y \mid x)}{\pi'(y \mid x)} \leq \exp(2B/\beta).$$

we can multiply them and get

$$\sup_{x,y} \frac{\pi(y \mid x)}{\pi'(y \mid x)} \leq \exp(4B/\beta).$$

$\square$

**Lemma D.3** (Linear Structure). *Suppose that we have reward sequence $\{r^t(x)\}_{t \in [T]}$ with $r^t(x) = \langle \theta^t, \phi(x) \rangle$ with $\|\theta\| \leq 1, \|\phi(x)\| \leq B$, then for any policy $\{\pi^t\}_{t \in [T]}$ for any $\mu > 0$, we can have*

$$\sum_{t=1}^{T} \mathbb{E}_{x \sim \pi^t}[r^t(x)] \leq \mu \cdot \sum_{t=1}^{T} \sum_{j=1}^{t-1} \mathbb{E}_{x \sim \pi^j}[(r^t(x))^2 + \widetilde{\mathcal{O}}(Bd) + \frac{d_{\mathrm{cover}}(1/T)}{4\mu}.$$

*Proof.* First, denote $X^t = \mathbb{E}_{x \sim pi^t}[\phi(x)]$, then

$$\sum_{t=1}^{T} \mathbb{E}_{x \sim \pi^t}[r^t(x)] = \sum_{t=1}^{T} \mathbb{E}_{x \sim \pi^t}[\langle \theta^t, \phi(x) \rangle]$$

$$= \sum_{t=1}^{T} \langle \theta^t, X^t \rangle.$$

Now define $\Sigma_t = \varepsilon I + \sum_{i=1}^{t-1} X^i (X^i)^\top$, then we can decompose the term above as

$$\sum_{t=1}^{T} \langle \theta^t, X^t \rangle = \underbrace{\sum_{t=1}^{T} \langle \theta^t, X^t \rangle \mathbb{I}\{\|X^t\|_{\Sigma_t^{-1}} \leq 1\}}_{(A)} + \underbrace{\sum_{t=1}^{T} \langle \theta^t, X^t \rangle \mathbb{I}\{\|X^t\|_{\Sigma_t^{-1}} > 1\}}_{(B)}.$$

The term (A) can be bounded as

$$(A) = \sum_{t=1}^{T} \|\theta^t\|_{\Sigma_t} \|X^t\|_{\Sigma_t^{-1}} \mathbb{I}\{\|X^t\|_{\Sigma_t^{-1}} \leq 1\}$$

$$\leq \sum_{t=1}^{T} \|\theta^t\|_{\Sigma_t} \min\{1, \|X^t\|^2_{\Sigma_t^{-1}}\}^{1/2}$$

$$\leq \sum_{t=1}^{T} \left[\varepsilon\|\theta^t\|^2 + \sum_{i=1}^{t-1}\langle\theta^t, X^i\rangle^2\right]^{1/2} \min\{1, \|X^t\|^2_{\Sigma_t^{-1}}\}^{1/2}$$

$$\leq \sqrt{\left[\sum_{t=1}^{T}\left(\varepsilon\|\theta^t\|^2 + \sum_{i=1}^{t-1}\langle\theta^t, X^i\rangle^2\right)\right] \cdot \left[\sum_{t=1}^{T}\min\{1, \|X^t\|^2_{\Sigma_t^{-1}}\}\right]},$$

where the last inequality uses the Cauchy's inequality.

Now we recall the elliptical potential lemma in (Abbasi-Yadkori et al., 2011), we can get

$$\sum_{t=1}^{T}\min\{1, \|X^t\|^2_{\Sigma_t^{-1}}\} \leq d(\varepsilon) = \widetilde{\mathcal{O}}(d\log(1/\varepsilon)). \tag{66}$$

Thus substitute it into the the inequality for (A), we can get

$$(A) \leq \sqrt{d(\varepsilon) \cdot \left[\sum_{t=1}^{T}\left(\varepsilon\|\theta^t\|^2 + \sum_{i=1}^{t-1}\langle\theta^t, X^i\rangle^2\right)\right]}.$$

Now by the inequality that $\sqrt{a+b} \leq \sqrt{a} + \sqrt{b}$, we can get

$$(A) \leq \sqrt{d(\varepsilon) \cdot \left[\sum_{t=1}^{T}\left(\varepsilon\|\theta^t\|^2 + \sum_{i=1}^{t-1}\langle\theta^t, X^i\rangle^2\right)\right]}$$

$$\leq \sqrt{d(\varepsilon)\varepsilon T} + \sqrt{d(\varepsilon)\cdot\sum_{t=1}^{T}\sum_{t=1}^{t-1}\langle\theta^t, X^i\rangle^2}$$

$$\leq \sqrt{d(\varepsilon)\varepsilon T} + \frac{d(\varepsilon)}{4\mu} + \mu\cdot\sum_{t=1}^{T}\sum_{i=1}^{t-1}\langle\theta^t, X^i\rangle^2$$

$$= \sqrt{d(\varepsilon)\varepsilon T} + \frac{d(\varepsilon)}{4\mu} + \mu\cdot\sum_{t=1}^{T}\sum_{j=1}^{t-1}(\mathbb{E}_{\pi^i}[r^t(x)])^2.$$

Now if we choose $\varepsilon = 1/T$, then $d(\varepsilon) = \widetilde{\mathcal{O}}(d)$, and the upper bound of (A) becomes

$$(A) \leq \sqrt{d_{\text{cover}}(1/T)} + \frac{d_{\text{cover}}(1/T)}{4\mu} + \mu\cdot\sum_{t=1}^{T}\sum_{j=1}^{t-1}(\mathbb{E}_{\pi^i}[r^t(x)])^2.$$

Now we derive the upper bound of (B).

$$(B) = \sum_{t=1}^{T}\langle\theta^t, X^t\rangle\mathbb{I}\{\|X^t\|_{\Sigma_t^{-1}} > 1\}$$

$$\leq B\cdot\sum_{t=1}^{T}\mathbb{I}\{\|X^t\|_{\Sigma_t^{-1}} > 1\}$$

$$\leq B\sum_{t=1}^{T}\min\{1, \|X^t\|^2_{\Sigma_t^{-1}}\}$$

$$\leq Bd_{\text{cover}}(1/T) = \widetilde{\mathcal{O}}(Bd).$$

So by adding (A) and (B), we can finally get

$$\sum_{t=1}^{T}\langle\theta^t, X^t\rangle \le Bd_{\text{cover}}(1/T) + \sqrt{d_{\text{cover}}(1/T)} + \frac{d_{\text{cover}}(1/T)}{4\mu} + \mu\cdot\sum_{t=1}^{T}\sum_{j=1}^{t-1}(\mathbb{E}_{\pi^i}[r^t(x)])^2$$

$$\le \widetilde{\mathcal{O}}(Bd) + \frac{d_{\text{cover}}(1/T)}{4\mu} + \mu\cdot\sum_{t=1}^{T}\sum_{j=1}^{t-1}(\mathbb{E}_{\pi^i}[r^t(x)])^2.$$

$\square$

# E. Some Derivations in Section 4 and Section 5

## E.1. Derivation of Reward-free Modification

Now we derive the equation

$$J(r_1^{\theta_1}, r_2^{\theta_2}, \cdots, r_m^{\theta_m}, \alpha, \pi^\theta) - \sum_{i=1}^{m}\eta L_i(\theta_i) = C - \beta\mathbb{E}_{x\sim\rho, y\sim\pi_{\text{base}}}\left[\log\frac{\pi^\theta(y\mid x)}{\pi_{\text{ref}}(y\mid x)}\right] - \eta\sum_{i=1}^{m}L_i(\theta_i).$$

In fact, since

$$J(r_1^{\theta_1}, r_2^{\theta_2}, \cdots, r_m^{\theta_m}, \alpha, \pi) = \mathbb{E}_{y\sim\pi^\theta(\cdot\mid x)}\left[\sum_{i=1}^{m}\alpha_i r_i^{\theta_i}(x, y) - \beta\cdot\sum_{i=1}^{m}\alpha_i\cdot(\log\pi^\theta(y\mid x) - \log\pi_{\text{ref}}(y\mid x))\right]$$

$$= \mathbb{E}_{y\sim\pi^\theta(\cdot\mid x)}\left[\sum_{i=1}^{m}\alpha_i r(x, y) - \beta\cdot\sum_{i=1}^{m}\alpha_i\cdot(\log\pi^\theta(y\mid x) - \log\pi_{\text{ref}}(y\mid x))\right]$$

$$= \mathbb{E}_{y\sim\pi^\theta(\cdot\mid x)}\left[\log Z(r, x)\right],$$

where $Z(r, x) = \sum_{y\in\mathcal{Y}}\pi_{\text{ref}}(y\mid x)\exp(r(x, y)/\beta)$ is a normalization factor independent with $y$ (Rafailov et al., 2024). Now, since $Z(r, x)$ is independent with $y$, we can get

$$J(r_1^{\theta_1}, r_2^{\theta_2}, \cdots, r_m^{\theta_m}, \alpha, \pi) = \mathbb{E}_{y\sim\pi^\theta(\cdot\mid x)}\left[\log Z(r, x)\right]$$

$$= \mathbb{E}_{y\sim\pi_{\text{base}}(\cdot\mid x)}\left[\log Z(r, x)\right]$$

$$= \mathbb{E}_{y\sim\pi_{\text{base}}(\cdot\mid x)}\left[r(x, y) - \beta(\log\pi^\theta(y\mid x) - \log\pi_{\text{ref}}(y\mid x))\right]$$

$$= C - \beta\mathbb{E}_{y\sim\pi_{\text{base}}(\cdot\mid x)}\left[\log\frac{\pi^\theta(y\mid x)}{\pi_{\text{ref}}(y\mid x)}\right].$$

We complete the derivation.

## E.2. Update Rule of Gradient Descent

In this section, we show that the computational cost of Eq. (8) can be easily computed once the expectation of the score function can be derived.

In fact,

$$\nabla_{\theta_1}\left(-\beta\mathbb{E}_{x\sim\rho, y\sim\pi_{\text{base}}}[\log\pi^\theta(y\mid x)]\right) - \eta\nabla_{\theta_1}\sum_{i=1}^{m}\ell(\mathcal{D}_i, \theta_i)$$

$$= -\beta\underbrace{\mathbb{E}_{x\sim\rho, y\sim\pi_{\text{base}}}[\nabla_{\theta_1}\log\pi^\theta(y\mid x)]}_{(a)} - \eta\underbrace{\nabla_{\theta_1}\ell(\mathcal{D}_1, \theta_1)}_{(b)}.$$

Term (b) in the last line is the gradient of log-likelihood loss that appears in classical reward-free algorithm like DPO. For term (a), note that if $\|d\|_1 = 1$, we have

$$\pi^\theta \propto \pi_{\text{ref}}(y\mid x)\cdot\exp\left(\sum_{i=1}^{m}\beta d_i r_i^{\theta_i}(x, \cdot)\right) = \prod_{i=1}^{m}(\pi^{\theta_i}(y\mid x))^{d_i}.$$

Hence, denote $s(\theta, \pi^*) = \mathbb{E}_{\pi^*}[\nabla_\theta \log \pi^\theta(y \mid x)]$ is the expectation of the score function, we can then derive that

$$(\text{a}) = \beta d_1 \left( s(\theta_1, \pi_{\text{base}}) - s(\theta_1, \pi^\theta) \right).$$

Hence, the update rule can be efficiently computed as long as the score function is available, which commonly appears in previous RL algorithms such as REINFORCE.

Thus, if the learning rate is $\xi > 0$, the gradient descent update rule of $\theta_1$ is

$$\theta_1^t = \theta_1^{t-1} - \xi \left( \beta d_1(s(\theta_1, \pi_{\text{base}}) - s(\theta_1, \pi^\theta)) - \eta \nabla_{\theta_1^{t-1}} L_1^t(\theta_1^{t-1}) \right).$$

Also, for the reward-free version, we can change the term $L_1^t(\theta_1^{t-1})$ to

$$\sum_{(x, y_w, y_l) \in \mathcal{D}_1} \log \sigma \left( \beta \cdot \left( \log \frac{\pi^{\theta_1}(y_w \mid x)}{\pi_{\text{ref}}(y_w \mid x)} - \log \frac{\pi^{\theta_1}(y_l \mid x)}{\pi_{\text{ref}}(y_l \mid x)} \right) \right).$$

### E.3. Derivation of the reward-free equation of expected reward vector

We now prove that

$$(V_i^t) = \mathbb{E}_{\pi^t}[r_i^{\theta_i^t}(x, y) - \beta \mathbb{D}_{\text{KL}}(\pi^t \| \pi_{\text{ref}})] = C - \beta \mathbb{E}_{y \sim \pi_{\text{base}}} \left[ \log \frac{\pi^{\theta_i^t}(y \mid x)}{\pi_{\text{ref}}(y \mid x)} \right] - \beta \mathbb{E}_{y \sim \pi^t} \left[ \log \frac{\pi^{\theta_i^t}(y \mid x)}{\pi^t(y \mid x)} \right].$$

*Proof.* We note that

$$
\begin{aligned}
\mathbb{E}_{\pi^t}[r_i^{\theta_i^t}(x, y) - \beta \mathbb{D}_{\text{KL}}(\pi^t \| \pi_{\text{ref}})] &= \mathbb{E}_{\pi^t} \left[ r_i^{\theta_i^t}(x, y) - \beta \left( \log \frac{\pi^t(y \mid x)}{\pi_{\text{ref}}(y \mid x)} \right) \right] \\
&= \mathbb{E}_{\pi^t} \left[ Z(r_i^{\theta_i^t}, x) + \beta \left( \log \frac{\pi^{\theta_i^t}(y \mid x)}{\pi_{\text{ref}}(y \mid x)} \right) - \beta \left( \log \frac{\pi^t(y \mid x)}{\pi_{\text{ref}}(y \mid x)} \right) \right] \\
&= \mathbb{E}_{\pi^t}[Z(r_i^{\theta_i^t}, x)] + \beta \mathbb{E}_{\pi^t} \left[ \left( \log \frac{\pi^{\theta_i^t}(y \mid x)}{\pi^t(y \mid x)} \right) \right].
\end{aligned}
$$

Now note that $Z(r_i^{\theta_i^t}, x)$ is independent on $y$, hence

$$
\begin{aligned}
\mathbb{E}_{\pi^t}[Z(r_i^{\theta_i^t}, x)] &= \mathbb{E}_{\pi_{\text{base}}}[Z(r_i, x)] \\
&= \mathbb{E}_{\pi_{\text{base}}} \left[ r_i^{\theta_i^t}(x, y) - \beta(\log \pi^{\theta_i^t}(y \mid x) - \log \pi_{\text{ref}}(y \mid x)) \right] \\
&= C - \beta \mathbb{E}_{y \sim \pi_{\text{base}}} \left[ \log \frac{\pi^{\theta_i^t}(y \mid x)}{\pi_{\text{ref}}(y \mid x)} \right].
\end{aligned}
$$

$\square$

