# OpenReview forum: "Projection Optimization: A General Framework for Multi-Objective and Multi-Group RLHF"
_ICML.cc/2025/Conference — ICML 2025 poster_

### Official Review · Reviewer_uhy5 · 2025-03-10

**Overall Recommendation:** 3

**Summary:**

This paper primarily focuses on aligning large language models (LLMs) to multiple objectives using per-objective preference data. Prior works on this topic primarily aim to achieve Pareto optimal alignment across all objectives by linearly aggregating all objectives into a single unified form and optimizing the unified objective. In this work, they aim to propose an efficient strategy to optimize for the non-linear aggregation of the multiple objectives. Moreover, they consider a multi-group setting wherein each group has a different level of importances over the objectives. They propose to use the multi-objective p-norm aggregation framework from the social choice theory that leads to general function classes (non-linear) and also account for the group preferences in this aggregation framework.

Instead of training a policy that optimizes for this aggregated objective, the authors propose an alternative approach: training a policy that minimizes the distance between its expected reward vector and a target set $W$. This target set consists of all vectors whose aggregated p-norm value exceeds a certain constant $c$. Moreover, they also consider the case where each group has a different target set $W$.  Then the policy optimization aims to minimize either (i) the distance between the expected reward vector incurred by the policy and the intersection of all target sets across groups (consensus setting) or (ii) minimize a weighted average of the distances between the expected reward vector incurred by the policy and the target set of each group (malfare setting).

The proposed algorithm consists of two key steps. First, it computes the projection direction between the expected reward vector and the target set. Then, using this direction vector as weights over the objectives, it optimizes a weighted loss using two approaches: reward-free multi-objective optimization and reward-based multi-objective optimization. The algorithm iteratively alternates between computing the direction vector and updating the policy. The authors also provide convergence guarantees for both the consensus and malfare settings.

Finally, they propose an online variant of the algorithm, where responses are collected using the current policy, and preferences are gathered based on the objective with the greatest difference. The importance weights over objectives are estimated online using a likelihood-based approach. For experiments, they fine-tune the Llama2-7B model on Anthropic-HH data using the offline version of the proposed approach for $p=0.5$ and $p=-\infty$ and compare the performance against relevant baselines.

**Claims And Evidence:**

The paper primarily builds upon the target set approach from [Yu et al., 2021] which aims to minimize the distance between the expected reward vector $ S(\pi) $ incurred by the policy and the set of acceptable reward vector values whose aggregation exceeds a predefined constant. However, [Yu et al., 2021] was designed for a reinforcement learning (RL) setting where the initial state is fixed (see Page 4 - Interaction Protocol), resulting in a single $ S(\pi) $ that can be directly compared with the target set.

In contrast, in this RLHF setting, optimization is typically performed over a distribution of prompts, where the objective is to maximize the expected reward across prompts. This key aspect is not addressed in the paper. Furthermore, according to the objective defined in Section 3.1, where the expectation over prompts is taken outside the aggregation, the aggregated reward for each prompt should either satisfy the target set constraint or be close to it. However, the feasibility of this approach and whether it is actually implemented in the experiments remain unaddressed.

To the best of my knowledge, this presents a fundamental issue that requires clarification from the authors.


## update after rebuttal
My concerns regarding having multiple algorithms are addressed. Essentially, there are only 2, as the authors point out, and only one of them (the offline version) has been evaluated. I still believe the paper can be restructured quite a bit to improve the clarity. Kindly recommend the authors include additional experiments and more experimental discussions in the paper. Also, improve the introduction such that $ S(\pi) $ is well defined, as pointed out.

**Essential References Not Discussed:**

The following works related to group preference and multi-objective alignment are missing in related works:

1. Zhao, Siyan, John Dang, and Aditya Grover. "Group preference optimization: Few-shot alignment of large language models." arXiv preprint arXiv:2310.11523 (2023).

2. Ramesh, Shyam Sundhar, Yifan Hu, Iason Chaimalas, Viraj Mehta, Pier Giuseppe Sessa, Haitham Bou Ammar, and Ilija Bogunovic. "Group robust preference optimization in reward-free rlhf." Advances in Neural Information Processing Systems 37 (2024): 37100-37137.

3. Guo, Yiju, Ganqu Cui, Lifan Yuan, Ning Ding, Zexu Sun, Bowen Sun, Huimin Chen et al. "Controllable preference optimization: Toward controllable multi-objective alignment." arXiv preprint arXiv:2402.19085 (2024).

4. Mukherjee, Subhojyoti, Anusha Lalitha, Sailik Sengupta, Aniket Deshmukh, and Branislav Kveton. "Multi-Objective Alignment of Large Language Models Through Hypervolume Maximization." arXiv preprint arXiv:2412.05469 (2024).

**Experimental Designs Or Analyses:**

The experimental evaluation is quite limited. Only one of the proposed algorithms has been evaluated, with minimal discussion of the results. Furthermore, the online version of the algorithm lacks any experimental evaluation, and details regarding the AR baseline are not discussed.

**Methods And Evaluation Criteria:**

The experimental evaluation is quite limited. Only one of the proposed algorithms has been evaluated, with minimal discussion of the results. Furthermore, the online version of the algorithm lacks any experimental evaluation.

**Other Comments Or Suggestions:**

Should it be $\theta_i$ instead of $\theta$ in (3)?

What is $\Pi_{W}(V)$?

**Other Strengths And Weaknesses:**

Strengths:

1. This paper focuses on aligning large language models (LLMs) to multiple objectives using per-objective preference data. It proposes an efficient strategy for optimizing the non-linear aggregation of multiple objectives. Moreover, it considers a multi-group setting where each group assigns different levels of importance to the objectives. To the best of my knowledge, the proposed methodology of minimizing the reward vector's distance to the target set represents a novel framework in this area of research.

2. Multiple algorithms have been proposed for both offline and online settings, with theoretical analysis establishing their convergence guarantees.

Weaknesses:

1. See comments above regarding the theoretical claims and experimental evaluation/design.

2. The paper is overly dense and difficult to follow in many places. It introduces numerous algorithms and concepts, but not all have been experimentally validated.

**Questions For Authors:**

Kindly comment on the issues in the theoretical claims detailed above.

**Relation To Broader Scientific Literature:**

This paper primarily focuses on aligning large language models (LLMs) to multiple objectives using per-objective preference data. Prior work in this area mainly seeks to achieve Pareto-optimal alignment by linearly aggregating all objectives into a single unified form and optimizing the resulting objective. In contrast, this work proposes an efficient strategy to optimize for the non-linear aggregation of multiple objectives.

Moreover, the authors consider a multi-group setting where each group assigns different levels of importance to the objectives. They adopt the multi-objective $ p $-norm aggregation framework from social choice theory, which generalizes to non-linear function classes and incorporates group preferences into the aggregation process.

To the best of my knowledge, the proposed methodology of minimizing the reward vector's distance to the target set represents a novel framework in this area of research.

**Theoretical Claims:**

In the theoretical proofs, $ S(\pi) $ is treated as the expectation over both prompts and policy, with the bounds based on the distance between the expected reward vector over all prompts and the target set. However, optimizing the expected aggregated reward, as detailed in Section 3.1, is not equivalent to ensuring that the aggregation of the expected reward vector satisfies the constraints (or is close to the target set of vectors that satisfies them). This is equivalent to interchanging expectation and aggregation and requires proper comments from the authors discussing the validity of such a maneuver.

---

> ### Author Rebuttal · Authors · 2025-04-01
>
> Thanks for your detailed review! We are happy to address your questions as follows.
>
>
> > 1. The discussion of the expectation over prompt
>
> Thanks for the good question! Our goal is to find the model $\pi$ that maximize the total expected reward $u(\pi)=\sum_{i=1}^m(\mathbb{E}\_{x\sim \rho, y\sim \pi}[r_i(x,y)])^p, $ where the aggregation is applied to the **expected reward $u_i(\pi) = \mathbb{E}\_{x\sim \rho, y\sim \pi}[r_i(x,y)]$**, rather than the reward for each prompt. In practice, the utility function (i.e., performance) of the language model is always measured over the entire prompt distribution, and our aggregation is applied to this overall utility.
>
> We acknowledge that our introduction to multi-objective learning (namely, Eq.(1), (2) and the beginning of Section 3.1) may have been unclear and misrepresented our motivation, potentially leading to this misunderstanding. We can clarify it as follows: For a language model $\pi$ (and single group), the utility for the model and the objective $i$ and the group $n$ is defined as $u_i(\pi) = \mathbb{E}\_{x\sim \rho, y\sim \pi}[r_i(x,y)]$, and the model aims to maximize the aggregated social welfare $(\sum_{i=1}^m \alpha_i (u_i(\pi))^p)^{1/p}$. To make this problem more tractable, we try to minimize the distance between target set and current expected reward vector $S(\pi) = (u_1(\pi), u_2(\pi), \cdots, u_m(\pi))$ (ignore the regularizer for clarity), which is a reasonable alternative. (See Example 3.1 and 3.2 in our paper).
>
> >2. The experimental evaluation and discussion
>
> We conduct our experiments on the Anthropic-HH dataset using the objectives Helpful, Harmless, and Humor, which are popular choices in previous MORLHF papers. We provide comprehensive results across different weights and objectives in both single-group and multi-group settings. Additionally, we include further experiments in the additional experiment part. We also provide an analysis for hyperparameter selection (the selection of $c$) in response to Reviewer MMQx.
>
> The experimental results demonstrate that our approach achieves a smaller distance between the target set and the expected reward vector, and also **achieves balance between diverse objectives**. For example, with the weight (0.7,0.3), objectives 'helpful' and 'harmless', and the aggregation parameter $p=0.5$, MOPO get the utility $u_{MOPO}=(0.123,0.412)$, while $u_{RS}=(1.203,-0.707), u_{MOD}=(1.295, -0.8)$. This shows that MOPO not only achieves a smaller distance to the target set but also better balances multiple objectives, aligning with our theoretical motivation of striking a balance between linear aggregation and max-min RLHF.
>
> Since the offline RLHF is more common, practical, and convenient (we can access offline dataset and reward-model easily), we only perform our method in offline RLHF. We believe that the online MOPO can also achieve better expected utility and better performance compared to linear aggregation, and we leave it as future work.
>
> The AR baseline means that we directly aggregate the reward to $r(x,y) = (\sum_{i=1}^n \alpha_i r_i^p(x,y))^{1/p}$, and train the reward directly using PPO. Since it requires aggregating the individual reward, it cannot be applied to the negative reward setting, while our algorithm can also work for the negative reward. In the experiment, we perform $r_i(x,y) = \max\{r_i(x,y),0\}$ to change the reward into non-negative.
>
> >3. The paper is overly dense in many places.
>
> Thanks for the reminder. We will improve our writing and proof to make it easier to follow in our final version.
>
> >4. Additional experiments.
>
> We also perform experiment results on the Summarize-from-Feedback dataset. We fine-tune ALPACA-7B for objectives Summary and Faithful and get two policies $\pi_1, \pi_2$. Then, we perform experiments on these two policies. We use $\alpha=(1/2,1/2)$ and $p=0.5$ for the aggregation, choose $c$ adaptively by $c=u(\pi^t) + 0.01$ where $u(\pi^t)$ is the expected utility for $\pi^t$ at round $t$ (See Response for Reviewer MMQx for the selection of $c$). The results are shown in the following table, which shows that our MOPO performs better than RS and MOD.
>
> | $\alpha$  | MOPO  | RS    | MOD   |
> |-----------|-------|-------|-------|
> | (0.3,0.7) | 0.521 | 0.515 | 0.507 |
> | (0.5,0.5) | 0.522 | 0.517 | 0.502 |
> | (0.7,0.3) | 0.549 | 0.552 | 0.530 |
>
> >5. Typos and definitions
> Thanks for the reminder. Eq. (3) should be $\theta_i$ instead of $\theta$, and $\Pi_W(V)$ means the projection of the point $V$ on the set $W$, which can be written as $\Pi_W(V) = \arg \min_{x \in W} \|\|V-x\|\|_2^2$.
>
> >6. Missing references
>
> Thank you for providing additional relevant references! We will include a detailed discussion in our next version.
>
> Thanks again for your time and effort! We will be happy to answer any further questions you may have.

---

> > ### Comment · Reviewer_uhy5 · 2025-04-02
> >
> > Thank you for running additional experiments and clarifying the theoretical questions. I also appreciate the further discussion on experiments.
> >
> > However, my concern is still that 4 algorithms have been introduced/proposed in the paper and only one has been evaluated. If the online algorithms are not the primary focus and are not evaluated, they could have been moved to the appendix. Instead, further experimental details and discussion could have been included in the main paper. As it stands, the paper has only half a page of experimental discussion, which is very uncommon in ML papers in the area of LLMs.
> >
> > Further, as the authors themselves agree, the motivation and introduction to their approach need major revision.
> >
> > For the above reasons, I believe the paper needs major restructuring, and I will maintain my score.

---

> > > ### Author Response · Authors · 2025-04-09
> > >
> > > Thank you so much for the response and we are happy to further clarify the concerns.
> > >
> > > Since the primary contribution of our paper lies in the theoretical analysis, we believe the online setting—particularly the incorporation of importance weight estimation into the theoretical framework—is crucial for addressing the challenge of an unknown $\alpha$. Also, although our paper includes several algorithmic components, Algorithms 1 and 2 are merely subprocedures within the overall MOPO framework. Actually, we only have two algorithms: offline MOPO and online MOPO. We chose to evaluate offline MOPO because it can be easily compared to the previous work and show the effectiveness of our projection-based optimization approach, while the online MORLHF has been limitedly studied in previous work.
> > >
> > > If you consider MOP-RB and MOP-RF in Algorithm 3 as two distinct algorithms, it’s important to note that, in practice, we typically ignore the exploration term. Then, the key difference between MOP-RB and MOP-RF only lies in whether we derive the optimal policy for each objective using reward-based algorithms like PPO or DPO. However, this derivation is not the central focus of our work. The most important aspect of our paper is the combination of the optimal policies, rather than the process of deriving them. In fact, we conduct experiments across different datasets, multiple objectives, and various LLMs to demonstrate the effectiveness of the proposed MOPO algorithm.
> > >
> > > Regarding the motivation and introduction, we believe the misunderstanding can be resolved through improved writing, rather than requiring major revisions. Our motivation is to apply a more general form of aggregation, rather than relying solely on linear aggregation, in the MORLHF setting. It is natural to aggregate the expected reward $u_i(\pi)$, as it is common for evaluating an LLM's performance. We will improve our writing to avoid this confusion in the next version.
> > >
> > > We hope our response has helped further address your concerns, and we would appreciate it if our response could be considered in your final recommendation.

---

### Official Review · Reviewer_3eRZ · 2025-03-12

**Overall Recommendation:** 4

**Summary:**

This paper introduces a novel Multi-Objective RLHF (MORLHF) framework that leverages per-objective preference feedback to achieve Pareto optimality by aggregating multiple objectives into a single unified optimization target. Unlike existing approaches that rely on linear aggregation, this work overcomes their limitations by reformulating the non-linear aggregation maximization problem into a series of sub-problems, each involving only linear aggregation. This decomposition enhances computational efficiency. The framework is further extended to handle multi-group settings and scenarios with unknown objective weights. Experimental results demonstrate a clear advantage of the proposed method over benchmark approaches.

## update after rebuttal: Thanks for the rebuttal. I will keep my original rating.

**Claims And Evidence:**

Yes. The claim is supported by evidence in theorem and experiments.

**Essential References Not Discussed:**

See my questions.

**Experimental Designs Or Analyses:**

Yes, the experiments are good.

**Methods And Evaluation Criteria:**

Yes

**Other Comments Or Suggestions:**

See my questions.

**Other Strengths And Weaknesses:**

See my questions.

**Questions For Authors:**

1. The multi-group problem considered in this paper is closely related to reinforcement learning from human feedback (RLHF) with heterogeneous feedback, where individuals may provide varying feedback based on their expertise or background. However, the paper assumes that the type/group of each human is known in advance and that the reward of each objective remains invariant across groups. In practice, this group information may be unknown. Lee et al. (2024) address this challenge by learning a latent representation of heterogeneous human feedback. Could the authors discuss potential ways to relax this assumption?

Lee et al. (2024), *Low-Rank Contextual Reinforcement Learning from Heterogeneous Human Feedback*, https://arxiv.org/abs/2412.19436.

2. Given that there are $m$ different objectives and $N$ distinct human groups, does the framework assume that each human provides pairwise feedback for all $m$ objectives? If so, this may be a strong assumption. In practical online interaction settings, when asking a human to compare two responses, it may be impractical to request comparisons across all specified objectives.

3. Building on the previous question, the assumption that $m$ objectives are explicitly available may not always hold in practice. For example, when soliciting human feedback on two responses to a prompt, we do not necessarily know which specific objectives a human considers when making their judgment. How does the proposed framework address this challenge?

4. Assumption 5.3 requires a constant gap between the reward functions of two provided responses, which appears to be a strong assumption. In practice, this gap may decrease over time as the learning process progresses—i.e., as training advances, both responses may improve, leading to a diminishing difference in rewards. Would it be possible to model this gap as a decaying function of the learning horizon $T$ or other parameters?

**Relation To Broader Scientific Literature:**

The Multi-Objective RLHF is an interesting topic and has received increasing attentions recently.

**Theoretical Claims:**

See my questions.

---

> ### Author Rebuttal · Authors · 2025-04-01
>
> Thanks for your positive response and time in reviewing our paper! We will address your questions as follows.
>
> >1. The paper assumes that the type/group of each human is known in advance.
>
> In this paper, we assume the group information is known. However, if the group information is unknown, we can use EM or other clustering algorithms to first get an estimation of the group indexes. This approach is also used in previous work [1].
>
> >2. Does the paper assume that human provides pairwise feedback for all objectives?
>
> Yes. We assume that humans can give feedback for all objectives. However, this is not a strong assumption.
> In fact,  it is a common assumption that we have feedback or the reward model for all objectives in previous MORLHF works [2,3,4,5]. Also, in our online setting, we do not need the human to give all objectives for one response. Instead, we only need users to give pairwise feedback on the objectives they consider most important. Specifically, we ask them to first select the most important objective and provide a comparison between them. The selection of the objective is modeled by a softmax distribution based on the weighted reward gap. $\alpha_i \cdot (r_i(x,y_1)-r_i(x,y_2))$ (See Line 392 in our paper).
>
> >3. The assumption that objectives are explicitly available may not always hold in practice.
>
> That's a great question! One possible way to address this challenge is to require a "reason" alongside the human's response. This reason could be a property or a short explanation, such as "helpfulness" or "Response 2 is too long". When the model encounters a reason that differs from the previous ones, it can identify this reason as a potentially new factor and add it as a new objective. This is an exciting and meaningful direction for future research, which we leave for future work.
>
> 4. The assumption 5.3 seems too strong.
>
> In fact, the distribution of $y_1$ and $y_2$ will not be close as training advances: $y_1\sim \pi^*(y\mid x)$, which is the optimal policy, while $y_2\sim \pi_{ref}(y\mid x)$ only follows the reference policy (typically the policy after SFT). Therefore, we simply assume that the performance gap between the optimal human-aligned policy and the reference policy remains constant, which is generally a reasonable assumption in practice.
>
>
>
> Thanks again for your time and effort! If you have more questions, we are happy to solve them.
>
>
>
>
> [1]. Chakraborty et al. 2024: MaxMin-RLHF: Alignment with Diverse Human Preferences
>
> [2]. Shi et al. 2024: Decoding-Time Language Model Alignment with Multiple Objectives.
>
> [3]. Yang et al. 2024: Rewards-in-Context: Multi-objective Alignment of Foundation Models with Dynamic Preference Adjustment.
>
> [4]. Wang et al. 2024: Interpretable Preferences via Multi-Objective Reward Modeling and Mixture-of-Experts
>
> [5]. Mukherjee et al. 2024: Multi-Objective Alignment of Large Language Models Through Hypervolume Maximization

---

### Official Review · Reviewer_MMQx · 2025-03-13

**Overall Recommendation:** 4

**Summary:**

This paper introduces a projection-based optimization framework for Multi-Objective Reinforcement Learning with Human Feedback (MORLHF). The approach reformulates non-linear reward aggregation as a series of linear sub-problems, enabling computationally efficient Pareto-optimal solutions. The framework is extended to multi-group settings, allowing for consensus or aggregated utility optimization across diverse preference groups.

**Claims And Evidence:**

The authors argue that existing MORLHF methods, which rely on linear aggregation, fail to address fairness and multi-group preferences effectively. These claims are supported by theoretical sublinear regret guarantees for both offline and online settings and empirical results demonstrating improved fairness and competitive performance against baselines like RS, MOD, and max-min RLHF.

**Essential References Not Discussed:**

No

**Experimental Designs Or Analyses:**

The experimental design in this paper is not comprehensive enough.
1. The lack of analytical experiments on multiple LLMs may limit the universality of the findings, and the experimental verification of the framework for solving multi-objective problems is limited to only three objectives, which raises uncertainty about its potential extension or wider application.
2. While the experiments validate the framework’s effectiveness, they lack an analysis of hyperparameter impacts.
3. Although the authors emphasize computational efficiency, they fail to provide a direct cost comparison with state-of-the-art LLM models or nonlinear aggregation baselines.

**Methods And Evaluation Criteria:**

Yes, it makes sense.

**Other Comments Or Suggestions:**

Providing a code library would assist other researchers in validating and extending the work.

**Other Strengths And Weaknesses:**

Strengths:
1. Efficient solution to address Multi-Objective and Multi-Group RLHF Problems Under Non-Linear Aggregation.
2. The theoretical derivation is comprehensive.

Weaknesses:
1. While the experimental results demonstrate strong performance, the scalability of the proposed method remains uncertain, as it was solely tested on one LLM and three objectives.
2. Lack of guidance on choosing hyperparameters for practitioners.

**Questions For Authors:**

No

**Relation To Broader Scientific Literature:**

The paper situates itself within RLHF and multi-objective RL literature, and proposes multi-objective and multigroup RLHF problems under non-linear aggregation. It also connects to social choice theory and Blackwell approachability.

**Theoretical Claims:**

This paper provides a relatively comprehensive proof for its theoretical statements.

---

> ### Author Rebuttal · Authors · 2025-04-01
>
> Thanks for your positive response and time in reviewing our paper! We will address your questions as follows.
>
>
>
> 1. Analysis of hyperparameter impacts and selection.
>
> Since $\alpha$ and $p$ is assumed to be given in the experiment, the only hyperparameter specifically included in our algorithm is the parameter $c$, which represents the requirement by human.
>
> If $c$ is too small, the target set is easily achieved, causing MOPO to lose the motivation to change direction. (In this case, the induced LLM is enough for achieving the human's requirement.) On the other hand, if $c$ is too large, the target set becomes distant, and the projection direction will tend to balance $[1/m,1/m,\cdots,1/m]$, which is also not conducive to maximizing the weighted p-norm expected utility. Hence, a value that is slightly larger than the expected utility could be suitable for $c$, for maximizing the expected utility function.
>
> In practice, it can be selected in different ways:
> (a) Since it represents a requirement set by the human, it can be provided directly by the human. Note that in this case, if LLM already satisfied the requirement, it will have no motivation to further enhance the expected utility.
> (b) If the goal is to maximize the expected utility function $u(\pi)=\sum_{i=1}^m(\mathbb{E}\_{x\sim \rho, y\sim \pi}[r_i(x,y)])^p,$ $c$ can be adaptively chosen to be slightly larger than the current expected utility, such as $c = u(\pi^t) + 0.01$ after getting the policy $\pi^t$ at each round $t$. Then, the model will tend to maximize the expected utility by approaching the target set.
>
> In experiments in our original paper, we manually choose the parameter $c$ that is slightly larger than the previously calculated expected utility. We also provide additional experiments using the adaptive approach (b) to validate our statements. See the additional experiment for details.
>
> For other hyperparameters like $\beta$ and $\eta$, these are standard hyperparameters used in other works ($\beta$ in [1], $\eta$ in [4]). We can just use the standard configuration.
>
> >2. The lack of analytical experiments on multiple LLMs may limit the universality of the findings.
>
> We conduct experiments on LLAMA2-7B, which is a commonly used model.  We use three objectives to evaluate our algorithm since they are classical metrics that are used in previous works [1,2,3]. Since the primary contribution of this paper is theoretical, we provide preliminary experiments on popular benchmarks.
>
>  We also perform experiment results on the Summarize-from-Feedback dataset. We fine-tune a different LLM ALPACA-7B for objectives 'Summary' and 'Faithful' and get two policies $\pi_1, \pi_2$. Then, we perform experiments on these two policies. We use $\alpha=(1/2,1/2)$ and $p=0.5$ for the aggregation, and $c=u(\pi^t) + 0.01$ for round $t$. The results are shown in the following table, which shows that our MOPO performs better than RS and MOD, with adaptively selected parameter $c$.
>
> | $\alpha$  | MOPO  | RS    | MOD   |
> |-----------|-------|-------|-------|
> | (0.3,0.7) | 0.521 | 0.515 | 0.507 |
> | (0.5,0.5) | 0.522 | 0.517 | 0.502 |
> | (0.7,0.3) | 0.549 | 0.552 | 0.530 |
>
> >3. Computational efficiency compared to non-linear aggregation baseline.
>
> The nonlinear aggregation baseline in our paper is Aggregated Reward (AR), which means that we directly aggregate the reward to $r(x,y) = (\sum_{i=1}^n \alpha_i r_i^p(x,y))^{1/p}$, and training the reward directly using PPO. It cannot be directly applied to the negative reward setting because of p-norm aggregation. To address this, we transform the rewards into non-negative values by applying $r_i(x,y) = \max\\{r_i(x,y),0\\}$.
>
> For computational efficiency, AR needs retraining each time we change the aggregation method or the weights, which takes more than 7 hours in our experiment and 30 minutes evaluation times per different weight. In contrast, our training-free algorithm does not require retraining. Instead, we use pre-calculated optimal policies for each objective, which only requires less than 30 minutes for evaluation to get the expected reward for each weight.
>
>
> We will release our code later in our revised version. Thanks again for your time and effort! If you have more questions, we are happy to solve them.
>
> [1]. Shi et al. 2024: Decoding-Time Language Model Alignment with Multiple Objectives.
>
> [2]. Zhou et al. 2024: Beyond One-Preference-Fits-All Alignment: Multi-Objective Direct Preference Optimization.
>
> [3]. Yang et al. 2024: Rewards-in-Context: Multi-objective Alignment of Foundation Models with Dynamic Preference Adjustment.
>
> [4]. Cen et al. 2024: Value-Incentivized Preference Optimization: A Unified Approach to Online and Offline RLHF.

---

> > ### Comment · Reviewer_MMQx · 2025-04-02
> >
> > Thanks for the authors’ response. I have no further questions.

---

### Official Review · Reviewer_Ekmw · 2025-03-21

**Overall Recommendation:** 3

**Summary:**

The paper proposes a general framework for multi-objective multi-group RLHF. The authors creatively draw inspiration from
RL with Blackwell-approachability to resolve handle the non-linear structure of the aggregated reward.

**Claims And Evidence:**

The authors provided theoretical and empirical guarantees for the proposed offline and online algorithms. Empirical evidence on anthropic HH supports these claims, with metrics showing performance gains over simpler baselines.

**Essential References Not Discussed:**

N/A

**Experimental Designs Or Analyses:**

While the results on anthropic HH illustrate the method’s viability, this dataset is somewhat outdated. Testing on newer datasets, such as HelpSteer2 with higher-quality multi-objective labels and more categories, would better demonstrate robustness and broad applicability.

**Methods And Evaluation Criteria:**

The proposed method is a good addition to the literature. I have some slight concerns regarding the theoretical and empirical analysis, discussed in the two sections below.

**Other Comments Or Suggestions:**

Please see above

**Other Strengths And Weaknesses:**

Please see above

**Questions For Authors:**

Please see above

**Relation To Broader Scientific Literature:**

The paper extends the literature of multi-objective and multi-group RLHF and provides novel algorithms.

**Theoretical Claims:**

- The methods developed are built on the assumption that the reward is non-negative and bounded. However,  the Bradley–Terry–derived reward may not be strictly in [0,B]. If negative values occur, the paper should discuss how p-norm aggregation (and potentially other proposed methods) might be affected or require rescaling.
- The minimax formulation in Equations (6) and (7) is a central theoretical contribution. However, solving it might be computationally challenging; the paper would benefit from a more concrete explanation of how one can efficiently compute or approximate the solution in large-scale settings.

---

> ### Author Rebuttal · Authors · 2025-03-31
>
> Thanks for your positive response and meaningful review! We will address your question as follows.
>
> >1. If negative values occur, the paper should discuss how p-norm aggregation applies.
>
> Both in theory and practice, our algorithm can handle both the negative reward and the positive reward. In fact, by shifting the objective from maximizing the aggregated expected reward to minimizing the distance between the target set and the expected reward vector, we can effectively handle situations where the expected reward is negative (as the expected reward vector can be negative while the target set remains in the positive range). In this case, the goal is to make the estimated reward vector (which may be negative) closely align with the target set. For the theoretical part, note that the p-norm aggregation appears in the definition of the target set, and the expected reward vector $S(\pi)$ can be negative. In practice,  MOPO performs well in experiments where the reward is negative, whereas truncating rewards to ensure positivity and directly aggregating them leads to failure (AR in our table).
>
> Moreover, the boundedness of the reward function is a common assumption and holds for most reward models.
>
> >2. The minimax formulation in Equations (6) and (7) is a central theoretical contribution. However, solving it might be computationally challenging; the paper would benefit from a more concrete explanation of how one can efficiently compute or approximate the solution in large-scale settings.
>
> This is an important question. We have two approaches to make it practical.  The first way is to use a reward-free DPO algorithm, as in Eq.(8) in our paper.  We can further simplify the Eq.(8) by replace $\log \pi_\theta(y\mid x)$ to $\alpha_i \log \pi_{\theta_i}(y\mid x)$, and Eq.(8) becomes decentralized optimization, where  $\\theta_i = \arg\min_{\theta}\beta \alpha_i\mathbb{E}\_{\pi_{base}} [\log \pi_{\theta}(y\mid x) - \eta \ell(D_i, \theta_i)]$ for each objective $i$, and we can perform VPO [1] for estimating $\theta_i$.
>
> Second, one can ignore the first exploration term, and Eq.(6) and Eq.(7) become traditional MLE for estimating the reward model. Also, one can use a pre-existing reward model if it is available. In our practical MOPO algorithm, we use a pre-existing reward model and use PPO to get the optimal policy $\pi_{\theta_i}$ for each objective. Then, Line 2 and Line 3 in Algorithm 1 can be implemented by previous linear MORLHF algorithms like MOD and RS. We will discuss it in detail in our next version.
>
> >3. While the results on anthropic HH illustrate the method’s viability, this dataset is somewhat outdated. Testing on newer datasets, such as HelpSteer2, with higher-quality multi-objective labels and more categories would better demonstrate robustness and broad applicability.
>
> We also perform experiment results on the Summarize-from-Feedback dataset [3]. We fine-tune a different LLM ALPACA-7B for objectives 'Summary' and 'Faithful' and get two policies $\pi_1, \pi_2$. Then, we perform experiments on these two policies. We use $\alpha=(1/2,1/2)$ and $p=0.5$ for the aggregation and choose $c$ adaptively by $c=u(\pi^t) + 0.01$ where $u(\pi^t)$ is the expected utility for $\pi^t$ at round $t$ (See Response for Reviewer MMQx for the selection of $c$ and Response for reviewer uhy5 for the definition of $u(\pi)$).
>
>  The following table shows the aggregated expected utility, which shows that our MOPO performs better than RS and MOD. Note that we do not select the HelpSteer dataset since the variance of its reward model is too small (see Appendix F.4 in [2]), making it less suitable for demonstrating the differences between linear and non-linear aggregation.
>
> | $\alpha$  | MOPO  | RS    | MOD   |
> |-----------|-------|-------|-------|
> | (0.3,0.7) | 0.521 | 0.515 | 0.507 |
> | (0.5,0.5) | 0.522 | 0.517 | 0.502 |
> | (0.7,0.3) | 0.549 | 0.552 | 0.530 |
>
> Thanks again for your time and effort! We will be happy to answer any further questions you may have.
>
>
> [1]. Cen et al. 2024: Value-Incentivized Preference Optimization: A Unified Approach to Online and Offline RLHF.
>
> [2]. Shi et al. 2024: Decoding-Time Language Model Alignment with Multiple Objectives.
>
> [3]. Stiennon et al. 2024: Learning to summarize from human feedback.

---

> > ### Comment · Reviewer_Ekmw · 2025-04-05
> >
> > Thank you for the response. I'll keep my score here.

---

### Decision · Program_Chairs · 2025-05-01

**Decision:**

Accept (poster)

**Comment:**

This paper proposes MOPO, a general framework for aligning large language models (LLMs) with human feedback across multiple objectives and user groups. Unlike most existing works that use linear aggregation of objectives, MOPO enables efficient optimization with non-linear aggregation through projection-based reformulations. The framework supports both offline and online RLHF and includes theoretical convergence guarantees. Empirical results on standard datasets and multiple model/objective setups demonstrate that MOPO improves over baseline approaches.

The paper tackles an increasingly relevant problem in LLM alignment—how to balance multiple competing objectives (e.g., helpfulness, harmlessness, humor) and diverse user preferences—using a principled and computationally efficient method. The projection-based approach enables support for non-linear utility aggregation and offers provable guarantees. The authors propose both offline and online variants and demonstrate the framework’s flexibility across single- and multi-group settings. Reviewers consistently highlighted the paper's strong theoretical depth and well-motivated methodology. The rebuttal further clarified confusion around prompt-level expectations and validated robustness via new experiments, including on the Summarize-from-Feedback dataset. Some concerns were raised about assumptions around known group identities, reward structures, and feedback availability—though these were addressed reasonably well in the rebuttal. Further refinement of the writing and presentation could improve accessibility.

Based on the common consensus, the paper is recommended for acceptance once the authors incorporate all the reviewer's feedback in the final version.